# A Unified View of Double-Weighting
# for Marginal Distribution Shift

**José I. Segovia-Martín**                                                    *jsegovia@bcamath.org*
*Basque Center for Applied Mathematics (BCAM)*
*Bilbao, Spain*

**Santiago Mazuelas**                                                        *smazuelas@bcamath.org*
*Basque Center for Applied Mathematics (BCAM)*
*IKERBASQUE-Basque Foundation for Science*
*Bilbao, Spain*

**Anqi Liu**                                                                 *aliu@cs.jhu.edu*
*CS department, Whiting School of Engineering*
*Johns Hopkins University*
*Baltimore, Maryland, USA*

**Reviewed on OpenReview:** *https://openreview.net/forum?id=aPyJilTiIb*

## Abstract

Supervised classification traditionally assumes that training and testing samples are drawn from the same underlying distribution. However, practical scenarios are often affected by distribution shifts, such as covariate and label shifts. Most existing techniques for correcting distribution shifts are based on a reweighted approach that weights training samples, assigning lower relevance to the samples that are unlikely at testing. However, these methods may achieve poor performance when the weights obtained take large values at certain training samples. In addition, in multi-source cases, existing methods do not exploit complementary information among sources, and equally combine sources for all instances. In this paper, we establish a unified learning framework for distribution shift adaptation. We present a double-weighting approach to deal with distribution shifts, considering weight functions associated with both training and testing samples. For the multi-source case, the presented methods assign source-dependent weights for training and testing samples, where weights are obtained jointly using information from all sources. We also present generalization bounds for the proposed methods that show a significant increase in the effective sample size compared with existing approaches. Empirically, the proposed methods achieve enhanced classification performance in both synthetic and empirical experiments.

## 1 Introduction

Supervised classification traditionally assumes that training and testing samples are independently and identically distributed (i.i.d.) drawn from the same underlying distribution. However, practical scenarios are often affected by distribution shifts, such as covariate shift and label shift. In covariate shift, the marginal distribution over the instances (covariates $x$) differs while the label conditional distribution remains the same. In label shift, the marginal distribution over the labels (classes $y$) differs while the instance conditional distribution remains the same. Additionally, in multi-source scenarios, the training data is obtained from multiple sources, each of which has different probability distributions.

Distribution shifts are common in many practical applications, including electronic health record data analysis (Singh et al., 2022). For example, a model may be trained to learn a patient's disease severity using historical patients' data, but there may exist shifts between training and testing populations due to the challenges in obtaining data from patients within the same population (Humbert-Droz et al., 2022). Moreover,

we may also need to learn from multiple datasets collected from different hospitals due to health record fragmentation.

Multiple techniques have been developed for correcting different types of distribution shifts (Quinonero-Candela et al., 2008; Sugiyama & Kawanabe, 2012; Zhang et al., 2013; Lipton et al., 2018; Sun et al., 2011; Schweikert et al., 2008; Zhang et al., 2015). Most existing distribution shift correction techniques are based on a reweighted approach. Reweighted techniques weight training samples assigning lower relevance to the training samples that are unlikely at testing. However, these methods require certain assumptions about the supports of the distributions and may achieve poor performance when the weights obtained take large values at certain training samples (Cortes & Mohri, 2014; Martino et al., 2018) (see Fig. 1). Additionally, existing methods for multi-source distribution shift adaptation inherit the problems of single-source reweighting methods. Most methods define classification rules as a linear combination of the classifiers learned independently on each source (Zhang et al., 2015; Shui et al., 2021; Wang et al., 2023) (see Fig. 2). Theoretical work (Mansour et al., 2008) has shown that it is more effective to have sample-dependent coefficients to allow different combinations of classification rules for each instance. Further details on the prior work are provided in Appendix A.

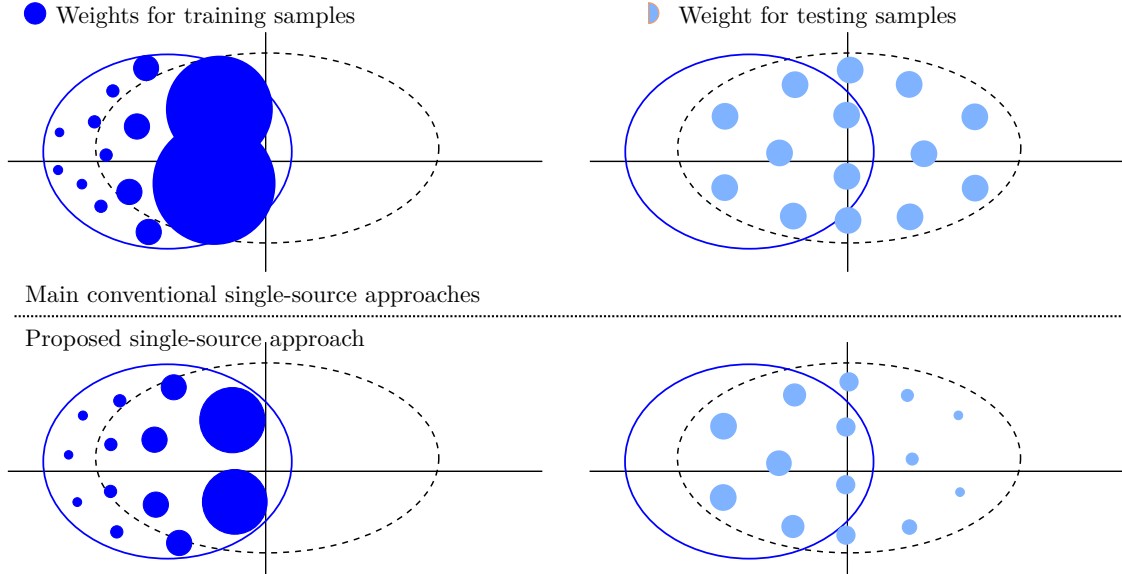

Figure 1: Different approaches for single-source distribution shift (the training and testing samples follow Gaussian distributions with probability mass concentrated in the blue and black circles, resp.). Existing methods obtain weights that take large values at certain training samples and do not consider weights at testing (i.e., weights at testing are constant). The proposed approach reduces the large values of training weights by utilizing weights at testing.

Recently, the double-weighting approach has been proposed to correct for single-source covariate shift (Segovia-Martín et al., 2023), addressing the limitations of covariate shift reweighted methods by assigning weights for both training and testing instances. The double-weighting approach does not require assumptions for supports and can avoid large values by adjusting the weights given the relation of weights (see Fig. 1). However, it remains unclear how assigning weights to both training and testing data points can help more general marginal distribution shift, like label shift and multi-source distribution shift adaptation. In the latter case, weight estimation and weight employment in both training and testing require the consideration of multiple training sources.

This paper establishes a unified learning framework for distribution shift adaptation using a double-weighting approach, considering weight functions that depend on the covariates and the labels. The usage of the double-weighting approach enables us to overcome the limitations of existing reweighted methods. Our framework can be reduced to dealing with covariate shift, label shift, and multi-source cases. In particular, in the

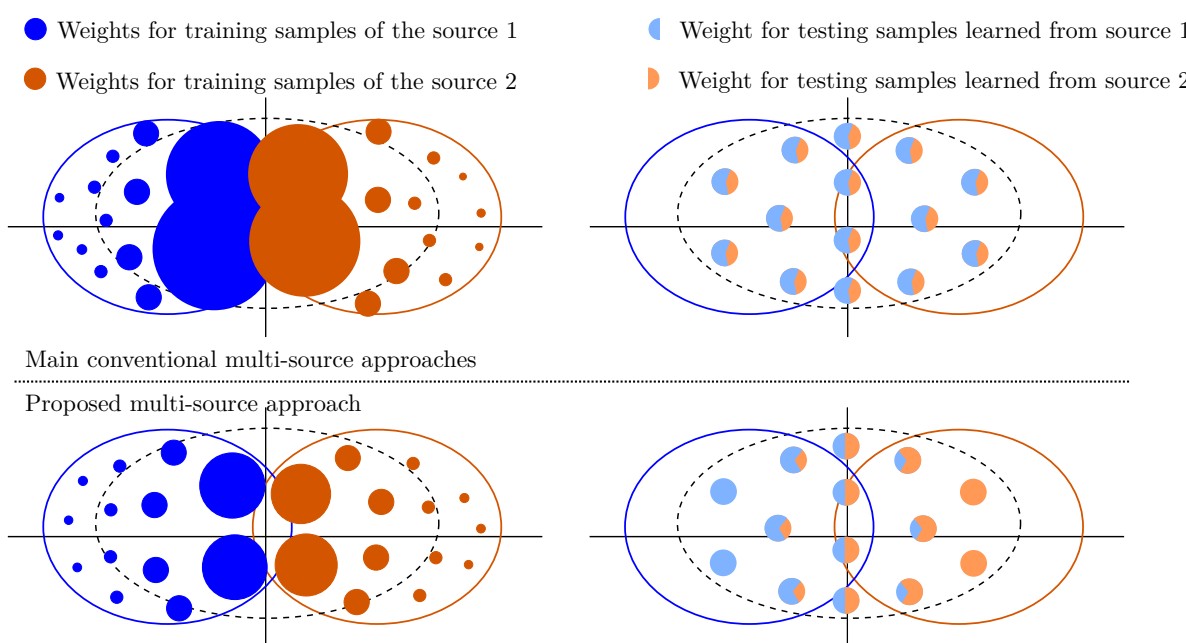

Figure 2: Different approaches for multi-source distribution shift (the two training sources and testing samples follow Gaussian distributions with probability mass concentrated in the blue, orange and black circles, resp.). Existing methods obtain weights that take large values at certain training samples and classify each testing instance using linear combinations of feature mappings. The proposed approach reduces the large values of training weights by also utilizing weights at testing and considers classification rules that involve sample-dependent combinations of feature mappings.

covariate shift and label shift cases, our approach alleviates the problem of extreme weights by also assigning weights for testing samples (see Fig. 1). In the multi-source settings, our proposed method leverages the rich complementary information among sources (see Fig. 2), inducing classification rules that involve sample-dependent weighted combinations of feature mappings.

The main contributions in the paper are as follows.

1. We establish a unified double-weighting learning framework to deal with general distribution shifts. Our framework can be reduced to dealing with covariate shift, label shift, and multi-source cases. In general, the double-weighting framework alleviates problems of reweighted techniques, avoiding extreme weights considering weights associated with the training and testing samples.

2. In the multi-source distribution shift cases, the proposed double-weighting method assigns source-dependent weights for training and testing samples to better utilize the information from multiple sources jointly. In addition, our multi-source methodology obtains classification rules that involve sample-dependent weighted combinations of feature mappings.

3. We develop generalization bounds for the proposed methods. Our analysis shows that double-weighting techniques significantly increase the effective sample size for different types of distribution shifts adaptation compared with reweighted approaches.

4. We demonstrate that the proposed techniques achieve a significant performance improvement in multiple distribution shift scenarios with experiments on both synthetic and real-world datasets. Our results show that in cases of significant shifts between training and testing distributions, most existing techniques result in a negative transfer of information among sources, while the proposed approach achieves improved performance.

**Notation.** Calligraphic upper case letters represent sets; bold lower and upper case letters represent vectors and matrices, respectively; for a vector $\mathbf{v}$, $\mathbf{v}^{\mathrm{T}}$ denotes its transpose, $v^{(i)}$ denotes its $i$-th component, $|\mathbf{v}|$ denotes its component-wise absolute value; $\mathbf{1}$ denotes a vector with all components equal to 1; $||\cdot||_1$, $||\cdot||_\infty$,

and $||\cdot||_{\mathcal{H}}$ denote the 1-norm, the infinity, and the Hilbert space norm of its argument, respectively; $\preceq$ and $\succeq$ represent vector component-wise inequalities; $N(\mathbf{x}; \mathbf{m}, \boldsymbol{\Sigma})$ denotes the pdf of a Gaussian r.v. $\mathbf{x}$ with mean $\mathbf{m}$ and covariance matrix $\boldsymbol{\Sigma}$; and $\mathbb{E}_{\mathrm{p}}\{\cdot\}$ denotes the expectation of its argument w.r.t distribution p.

## 2 Preliminaries

This section describes the problem setup and the framework for minimax risk classifiers (MRCs), a supervised classification learning framework that allows us to correct distribution shifts using the double-weighting methodology.

### 2.1 Problem setup

Let $\mathcal{X}$ be the set of instances and $\mathcal{Y}$ be the set of labels represented by the set $\{1, 2, \ldots, |\mathcal{Y}|\}$. We denote by $\Delta(\mathcal{X} \times \mathcal{Y})$ the set of probability distributions over $\mathcal{X}$ and $\mathcal{Y}$, and by $\mathrm{T}(\mathcal{X}, \mathcal{Y})$ the set of all classification rules from instances $\mathcal{X}$ to labels $\mathcal{Y}$. For $\mathrm{h} \in \mathrm{T}(\mathcal{X}, \mathcal{Y})$, we denote by $\mathrm{h}(y|x)$ the probability of assigning label $y \in \mathcal{Y}$ to instance $x \in \mathcal{X}$. With a slight abuse of notation, we denote by $\mathrm{h}(x)$ the label assignment provided by the classification rule h for the instance $x$. We use the notation $\mathrm{p}_{\mathrm{te}}$ for the underlying distribution at test, $(x_1, y_1), (x_2, y_2) \ldots, (x_n, y_n)$ for the training samples, and $x_{n+1}, x_{n+2}, \ldots, x_{n+t}$ for the testing instances. For the multi-source case, we use the notation $S$ for the number of training sources, $[S] = \{1, 2, \ldots, S\}$ for the set of sources, and $(x_{s,1}, y_{s,1}), (x_{s,2}, y_{s,2}) \ldots, (x_{s,n_s}, y_{s,n_s})$ for the training samples that belong to the source $s \in [S]$.

The $\ell$-risk of a classification rule h is its expected classification loss with respect to the true underlying distribution at test $\mathrm{p}_{\mathrm{te}}$, i.e.,

$$R_\ell(\mathrm{h}) = \mathbb{E}_{\mathrm{p}_{\mathrm{te}}} \{\ell(\mathrm{h}, (x, y))\}. \tag{1}$$

Supervised classification techniques use the training samples to find a classification rule h that has small $\ell$-risk $R_\ell(\mathrm{h})$. In this paper, we consider 0-1-loss and log-loss:

$$\ell_{01}(\mathrm{h}, (x, y)) = \mathbb{P}\{\mathrm{h}(x) \neq y\} = 1 - \mathrm{h}(y|x) \tag{2}$$
$$\ell_{\log}(\mathrm{h}, (x, y)) = -\log \mathrm{h}(y|x) \tag{3}$$

although the results presented can be analogously used with general losses, as is done in (Mazuelas et al., 2022).

In the following, we assume that, for single-source distribution shift, $n$ samples from $\mathrm{p}_{\mathrm{tr}}$ and $t$ testing instances from $\mathrm{p}_{\mathrm{te}}$ are available at learning. For the multi-source distribution shift, we assume that $n_s$ samples from $\mathrm{p}_s$ for $s \in [S]$ and $t$ testing instances from $\mathrm{p}_{\mathrm{te}}$ are available at learning.

**Single-source marginal distribution shift (covariate and label shift).** The training samples follow a distribution $\mathrm{p}_{\mathrm{tr}}(x, y)$ such that the marginal distributions of instances (resp. labels) differ, i.e., $\mathrm{p}_{\mathrm{tr}}(x) \neq \mathrm{p}_{\mathrm{te}}(x)$ (resp. $\mathrm{p}_{\mathrm{tr}}(y) \neq \mathrm{p}_{\mathrm{te}}(y)$), but the label conditional (resp. instance conditional) coincide, i.e., $\mathrm{p}_{\mathrm{tr}}(y|x) = \mathrm{p}_{\mathrm{te}}(y|x)$ (resp. $\mathrm{p}_{\mathrm{tr}}(x|y) = \mathrm{p}_{\mathrm{te}}(x|y)$).

**Multi-source marginal distribution shift (multi-source covariate and label shift).** The training samples from each of the $S$ training sources follow distribution $\mathrm{p}_s(x, y)$ such that the marginal distributions of instances (resp. labels) differ, i.e., $\mathrm{p}_s(x) \neq \mathrm{p}_{s'}(x)$ if $s \neq s' \in [S]$, $\mathrm{p}_s(x) \neq \mathrm{p}_{\mathrm{te}}(x)$ (resp. $\mathrm{p}_s(y) \neq \mathrm{p}_{s'}(y)$ if $s \neq s' \in [S]$, $\mathrm{p}_s(y) \neq \mathrm{p}_{\mathrm{te}}(y)$) for $s \in [S]$; but the label (resp. instance conditional) conditional coincide, i.e., $\mathrm{p}_s(y|x) = \mathrm{p}_{\mathrm{te}}(y|x)$ (resp. $\mathrm{p}_s(x|y) = \mathrm{p}_{\mathrm{te}}(x|y)$) for $s \in [S]$.

### 2.2 Minimax risk classifiers

Similarly to other approaches based on robust risk minimization (RRM) also known as distributionally robust learning (Farnia & Tse, 2016; Fathony et al., 2016), MRC methods (Mazuelas et al., 2022; 2023) do not require that the training and testing samples follow the same distribution. Notice that these methods are referred to as minimax risk techniques because they minimize the worst-case expected loss over distributions in an uncertainty set, but such approach is different to classical minimax analysis of classification methods (Tsybakov, 2009). The uncertainty sets $\mathcal{U}$ are formed by probability distributions that match

data-based constraints for the expectations of a function $\Phi : \mathcal{X} \times \mathcal{Y} \longrightarrow \mathbb{R}^m$ referred to as feature mapping (Mazuelas et al., 2022; 2023). These feature mappings are defined using one-hot encodings of the elements of $\mathcal{Y}$ as $\Phi(x, y) = \boldsymbol{e}_y \otimes \Psi(x)$, where $\boldsymbol{e}_y$ is the $y$-th element of the canonical basis of $\mathbb{R}^{|\mathcal{Y}|}$, $\otimes$ denotes the Kronecker product, and $\Psi : \mathcal{X} \to \mathbb{R}^d$ is a map that represents instances as real vectors of dimension $d$. Given the uncertainty set $\mathcal{U}$, a classification rule $\mathrm{h}^{\mathcal{U}}$ is an $\ell$-MRC for $\mathcal{U}$ if

$$\mathrm{h}^{\mathcal{U}} \in \arg \min_{\mathrm{h} \in \mathrm{T}(\mathcal{X}, \mathcal{Y})} \max_{\mathrm{p} \in \mathcal{U}} \ell(\mathrm{h}, \mathrm{p}) \tag{4}$$

where $\ell(\mathrm{h}, \mathrm{p})$ denotes the expected loss of classification rule h w.r.t. distribution p, and we denote by $R(\mathcal{U})$ the minimax risk against $\mathcal{U}$. The uncertainty set $\mathcal{U}$ is defined as

$$\mathcal{U} = \{\mathrm{p} \in \Delta\left(\mathcal{X} \times \mathcal{Y}\right) : |\mathbb{E}_{\mathrm{p}} \Phi(x, y) - \boldsymbol{\tau}| \preceq \boldsymbol{\lambda} \text{ and } \mathrm{p}(x) = \mathrm{p}_{\mathrm{te}}(x), \forall x \in \mathcal{X}\} \tag{5}$$

where $\boldsymbol{\lambda}$ is the confidence vector that assesses the inaccuracies in expectations estimates, and $\boldsymbol{\tau}$ denotes the mean vector of expectation estimates.

The mean vector $\boldsymbol{\tau}$ in (5) is an estimate of the expectation of the feature mapping $\mathbb{E}_{\mathrm{p}_{\mathrm{te}}} \Phi(x, y)$ with respect to the underlying distribution. In cases without distribution shift, the $n$ training samples $(x_1, y_1), (x_2, y_2), \ldots, (x_n, y_n)$ are drawn from the underlying distribution at test $\mathrm{p}_{\mathrm{te}}$ so that the expectation $\mathbb{E}_{\mathrm{p}_{\mathrm{te}}} \Phi(x, y)$ can be estimated using averages of training samples

$$\boldsymbol{\tau} = \frac{1}{n} \sum_{i=1}^{n} \Phi(x_i, y_i) \tag{6}$$

As shown in the following, the expectation estimate is modified in situations affected by distribution shifts to account for the difference between the training and the testing distribution.

### 2.3 Novelty with respect to double-weighting methods for single-source covariate shift

The results in the paper provide a unified approach for the adaptation to general types of distribution shifts, including label shift and multi-source shifts. On the other hand, the methods in (Segovia-Martín et al., 2023) address only the case of single-source covariate shift. The methods presented in the following address a broader class of distributional shifts that not only include covariate shift but also general shifts, label shifts, and multi-source shifts. The paper presents new algorithms for label shift, multi-source covariate shift, and multi-source label shift. In addition, the experimental results in Section 6 below demonstrate that the proposed methods obtain better performance than existing approaches, particularly in scenarios involving distribution support mismatch, aligning with the theoretical performance guarantees and generalization bounds presented in this paper.

Scenarios with multiple sources impose unique challenges and require different algorithms and classification rules. The analysis of how the weights for different sources interact with each other and how they contribute to the sample complexity is completely different from the single-source case addressed in (Segovia-Martín et al., 2023). Appendix B further discusses the contribution of the techniques presented with respect to those in (Segovia-Martín et al., 2023). In particular, the appendix shows how the new methods presented for multi-source adaptation can leverage the complementary information from different sources.

## 3 Unified Double-Weighting Framework for Single-Source Distribution Shifts

This section describes the unified learning framework for double-weighting for single-source distribution shift adaptation. We also present generalization bounds for the proposed framework in comparison with the reweighted framework.

### 3.1 Double-weighting pairwise distributions

The proposed framework considers weights $\beta(x, y)$ for the training distribution $\mathrm{p}_{\mathrm{tr}}$ and weights $\alpha(x, y)$ for the testing distribution $\mathrm{p}_{\mathrm{te}}$ (see Fig. 1). For any function $f$, we exploit the fact that

$$\mathbb{E}_{\mathrm{p}_{\mathrm{te}}(x, y)} \alpha(x, y) f(x, y) = \mathbb{E}_{\mathrm{p}_{\mathrm{tr}}(x, y)} \beta(x, y) f(x, y) \tag{7}$$

can be achieved if we have

$$p_{te}(x,y)\alpha(x,y) = p_{tr}(x,y)\beta(x,y) \tag{8}$$

that can be attained by multiple choices of the weights $\alpha(x,y)$ and $\beta(x,y)$. For the reweighted approaches, the choice of weights is

$$\alpha(x,y) = 1, \ \ \beta(x,y) = \frac{p_{te}(x,y)}{p_{tr}(x,y)} \tag{9}$$

if $p_{tr}(x,y) > 0 \Rightarrow p_{te}(x,y) > 0$ (Sugiyama & Kawanabe, 2012). However, the equality in (7) can be obtained in multiple ways, for instance the usage of

$$\alpha(x,y) = \min\left(1, \frac{p_{tr}(x,y)}{p_{te}(x,y)}C\right), \ \ \beta(x,y) = \min\left(\frac{p_{te}(x,y)}{p_{tr}(x,y)}, C\right) \tag{10}$$

allows us to satisfy (7) for any $C > 0$. Notice also that weights as those in (10) above can be utilized for general train and test distributions without requiring assumptions for their supports.

As shown in the following subsection, the weight functions $\beta(x,y)$ assign relevance to the training samples, while the weight functions $\alpha(x,y)$ determine the confidence in the predictions for the testing instances. Weights as in (10) can alleviate the limitations of reweighted approaches in distribution shift adaptation. By applying (8), if the ratio $p_{te}(x,y)/p_{tr}(x,y)$ is large, using a small $\alpha(x,y)$ enables having $p_{te}(x,y)\alpha(x,y) = p_{tr}(x,y)\beta(x,y)$ with moderate values of $\beta(x,y)$. Using weights as in (10), we predict with high confidence $(\alpha(x,y) \approx 1)$ by taking large $C$. Given that $\beta(x,y) \leq C$, we can consider small values of $\beta(x,y)$ by sacrificing prediction confidence (decreasing $C$).

The usage of weights $\alpha(x,y)$ and $\beta(x,y)$ tailored to general single-source distribution shift scenarios offers a significant improvement over methods based on the reweighted approach. The proposed approach can handle general single-source distribution shifts. In addition, this approach results in enhanced generalization in comparison with the reweighted approach, as shown in Theorem 3.2.

## 3.2 MRC learning using general double-weighting

This subsection describes the proposed learning methodology for the unified view of double-weighting using weights $\alpha(x,y)$ and $\beta(x,y)$.

**Uncertainty sets.** To address the single-source distribution shift, we construct an uncertainty set $\mathcal{U}$ defined in terms of constraints of the expectation of a weighted feature $\Phi_\alpha(x,y)$. Then, the uncertainty set $\mathcal{U}$ can be defined as

$$\mathcal{U} = \left\{ p \in \Delta\left(\mathcal{X} \times \mathcal{Y}\right) : |\mathbb{E}_p\Phi_\alpha(x,y) - \boldsymbol{\tau}| \preceq \boldsymbol{\lambda} \text{ and } p(x) = p_{te}(x), \forall x \in \mathcal{X} \right\} \tag{11}$$

where the feature mapping $\Phi_\alpha(x,y)$ is defined as $\Phi_\alpha(x,y) = \alpha(x,y)\Phi(x,y)$. The expectation of such feature mapping can be estimated using sample averages of training samples weighted by $\beta(x,y)$ as

$$\boldsymbol{\tau} = \frac{1}{n}\sum_{i=1}^n \Phi_\beta(x_i, y_i), \text{ for } \Phi_\beta(x,y) = \beta(x,y)\Phi(x,y). \tag{12}$$

Using weights $\alpha(x,y)$ and $\beta(x,y)$ that satisfy (8) we can achieve that the estimate $\boldsymbol{\tau}$ above is an unbiased estimator of $\mathbb{E}_{p_{te}(x,y)}\Phi_{\alpha(x,y)}$. In addition, the variance of such estimator can be reduced by using weights $\alpha(x,y)$ that avoid large weights $\beta(x,y)$. The constraints in the proposed uncertainty set $\mathcal{U}$ in (11) characterize weighted feature expectation matching on $\Phi_\alpha(x,y)$ in the training domain with weights $\beta(x,y)$.

**Convex optimization.** MRCs corresponding with the uncertainty set (11) can be learned by solving the convex optimization problem

$$\min_{\boldsymbol{\mu}} -\boldsymbol{\tau}^{\mathrm{T}}\boldsymbol{\mu} + \boldsymbol{\lambda}^{\mathrm{T}}|\boldsymbol{\mu}| + \mathbb{E}_{p_{te}(x)}\varphi_\ell\left(\boldsymbol{\mu}, x, \alpha\right) \tag{13}$$

where $\varphi_\ell$ is a function defined in terms of the loss. For 0-1-loss, we have

$$\varphi_{01}\left(\boldsymbol{\mu}, x, \alpha\right) = 1 + \max_{\mathcal{C} \subseteq \mathcal{Y}} \frac{\sum_{y \in \mathcal{C}} \Phi_\alpha(x,y)^{\mathrm{T}}\boldsymbol{\mu} - 1}{|\mathcal{C}|} \tag{14}$$

and, for log-loss, we have

$$\varphi_{\log}\left(\boldsymbol{\mu}, x, \alpha\right) = \log \sum_{y \in \mathcal{Y}} \exp\left\{\Phi_\alpha(x, y)^{\mathrm{T}}\boldsymbol{\mu}\right\} \tag{15}$$

**Theorem 3.1.** *Let $\boldsymbol{\tau}, \boldsymbol{\lambda} \in \mathbb{R}^m$ be such that the uncertainty set $\mathcal{U}$ in (11) is not the empty set. If $\boldsymbol{\mu}^*$ is a solution of (13) for 0-1-loss, the classification rule*

$$\mathrm{h}^{\mathcal{U}}(y|x) = \left(\Phi_\alpha(x, y)^{T}\boldsymbol{\mu}^* - \max_{\mathcal{C} \subseteq \mathcal{Y}} \frac{\sum_{y' \in \mathcal{C}} \Phi_\alpha(x, y')^{T}\boldsymbol{\mu}^* - 1}{|\mathcal{C}|}\right)_+ \tag{16}$$

*is a 0-1-MRC for $\mathcal{U}$. If $\boldsymbol{\mu}^*$ is a solution of (13) for log-loss, the classification rule*

$$\mathrm{h}^{\mathcal{U}}(y|x) = \frac{\exp\{\Phi_\alpha(x, y)^{T}\boldsymbol{\mu}^*\}}{\sum_{y' \in \mathcal{Y}} \exp\left\{\Phi_\alpha(x, y')^{T}\boldsymbol{\mu}^*\right\}} \tag{17}$$

*is a log-MRC for $\mathcal{U}$. In addition, the minimax risk $R(\mathcal{U})$ is given by*

$$R(\mathcal{U}) = -\boldsymbol{\tau}^{T}\boldsymbol{\mu}^* + \boldsymbol{\lambda}^{T}|\boldsymbol{\mu}^*| + \mathbb{E}_{\mathrm{p}_{te}(x)}\varphi_\ell\left(\boldsymbol{\mu}^*, x, \alpha\right). \tag{18}$$

*Proof.* See Appendix C. $\qquad\qquad\square$

**Remarks.** The convex optimization problem in (13) can be addressed using conventional techniques such as stochastic (sub)gradient method. Specifically, the optimization problem in (13) is an unconstrained convex optimization problem for which stochastic subgradients can be readily obtained using the testing instances $x_{n+1}, x_{n+2}, \ldots, x_{n+t}$. Note that such a subgradient can be efficiently computed even in cases with a sizable number of classes using the greedy approach shown in (Fathony et al., 2016).

**Regularization.** The convex optimization problem (13) implements L1-type regularization, with the regularization parameter represented by the vector $\boldsymbol{\lambda}$. This regularization term in (13) enables to penalize differently each component of the parameter $\boldsymbol{\mu}$, ensuring that feature components with poorly estimated expectations (i.e., components $i$ with large $\lambda^{(i)}$) are strongly penalized.

**Classification rule.** The form of the classification rules allows the adjustment of the confidence of the predictions based on the weight function $\alpha(x, y)$. For very small values of $\alpha(x, y)$ for all $y \in \mathcal{Y}$ and a specific testing instance $x$, the classifier $\mathrm{h}^{\mathcal{U}}$ uniformly assigns labels in the set $\mathcal{Y}$ for both losses, i.e., $\mathrm{h}^{\mathcal{U}}(y|x) = 1/|\mathcal{Y}|$ for all $y \in \mathcal{Y}$.

### 3.3 Generalization bounds

This subsection describes the generalization bounds of the proposed single-source methods. Such bounds are given in terms of the smallest minimax risk, $R^\infty$, that corresponds with the uncertainty set given by the exact expectations, and is defined by

$$R^\infty = \min_{\boldsymbol{\mu}} -\mathbb{E}_{\mathrm{p}_{te}(x,y)}\Phi_\alpha(x, y)^{\mathrm{T}}\boldsymbol{\mu} + \mathbb{E}_{\mathrm{p}_{te}(x)}\varphi_\ell\left(\boldsymbol{\mu}, x, \alpha\right). \tag{19}$$

The MRC corresponding to that smallest minimax risk $R^\infty$ could only be obtained by an exact estimation of the expectations of the feature mapping $\Phi_\alpha$ that in turn would require an infinite amount of training samples. The theorem below provides risk bounds for the proposed MRCs in terms of smallest minimax risks $R^\infty$, showing how the proposed methods can lead to a significant decrease in the estimation error compared to existing methods.

**Theorem 3.2.** *Let $\mathcal{U}$ be a non-empty uncertainty set given by (11) and $\mathrm{h}^{\mathcal{U}}$ be an $\ell$-MRC for $\mathcal{U}$. If weights $\alpha(x, y)$ and $\beta(x, y)$ are given by (10) with $C = B/\sqrt{D}$ for $D \geq 1$, $\boldsymbol{\mu}^*$ and $\boldsymbol{\mu}^\infty$ are solutions to (18) and (19), respectively, and*

$$B = \sup_{x \in \mathcal{X}, y \in \mathcal{Y}} \frac{\mathrm{p}_{te}(x, y)}{\mathrm{p}_{tr}(x, y)}. \tag{20}$$

*Then, with probability at least $1 - \delta$ we have that*

$$R(\mathrm{h}^{\mathcal{U}}) \leq R^{\infty} + \boldsymbol{\lambda}^T (|\boldsymbol{\mu}^{\infty}| - |\boldsymbol{\mu}^*|) + \|\boldsymbol{\mu}^{\infty} - \boldsymbol{\mu}^*\|_1 \|\boldsymbol{\tau} - \mathbb{E}_{\mathrm{p}_{te}} \Phi_\alpha(x,y)\|_{\infty}$$

$$\leq R^{\infty} + \boldsymbol{\lambda}^T (|\boldsymbol{\mu}^{\infty}| - |\boldsymbol{\mu}^*|) + M\|\boldsymbol{\mu}^{\infty} - \boldsymbol{\mu}^*\|_1 \sqrt{\frac{2B^2}{Dn} \log \frac{2m}{\delta}} \tag{21}$$

*where $M$ is a constant satisfying $\|\Phi(x,y)\|_{\infty} \leq M$ for all $x \in \mathcal{X}$, $y \in \mathcal{Y}$.*

*Proof.* See Appendix C. $\qquad\square$

Note that the difference between the risk $R(\mathrm{h}^{\mathcal{U}})$ and the smallest minimax risk $R^{\infty}$ decreases with the estimation error $\|\boldsymbol{\tau} - \mathbb{E}_{\mathrm{p}_{te}} \Phi_\alpha(x,y)\|_{\infty}$. Note that the term $\|\mu^{\infty} - \mu^*\|_1$ can be bounded by the constant term $2\|\mu^{\infty}\|_1$ because the optimization problem (13) carries out an L1 penalization while that in (19) does not have the L1 penalization term and does not depend on the training samples.

**Corollary 3.3.** *If weights $\alpha(x,y)$, $\beta(x,y)$ are given by (10) with $C = B/\sqrt{D}$ for $D \geq 1$, and $B$ as in (20), the estimation error is bounded as*

$$\left\|\boldsymbol{\tau} - \mathbb{E}_{\mathrm{p}_{te}(x,y)} \alpha(y)\Phi(x,y)\right\|_{\infty} \leq M\sqrt{\frac{2\|\beta(x,y)\|_{\infty}^2}{n} \log \frac{2m}{\delta}} \leq M\sqrt{\frac{2B^2}{Dn} \log \frac{2m}{\delta}} \tag{22}$$

*with probability $1 - \delta$, where $M$ is a constant satisfying that $\|\Phi(x,y)\|_{\infty} \leq M$ for all $x \in \mathcal{X}$, $y \in \mathcal{Y}$.*

*Proof.* The proof is straightforward using Hoeffding's inequality. $\qquad\square$

Using a reweighted approach, we have that

$$\|\beta(x,y)\|_{\infty} = B := \sup_{x \in \mathcal{X}, y \in \mathcal{Y}} \frac{\mathrm{p}_{te}(x,y)}{\mathrm{p}_{tr}(x,y)}. \tag{23}$$

If $\mathrm{p}_{te}(x,y)/\mathrm{p}_{tr}(x,y)$ take large values, using weights (9) as in reweighted leads to large values, while if we consider weights as in (10), we can have $\beta(x,y) = C < \mathrm{p}_{te}(x,y)/\mathrm{p}_{tr}(x,y)$. This is achieved at the cost of using values of $\alpha(x,y) = C\mathrm{p}_{tr}(x,y)/\mathrm{p}_{te}(x,y) < 1$. Using the weights in (10) we have that

$$\|\beta(x,y)\|_{\infty} = \frac{1}{\sqrt{D}} \sup_{x \in \mathcal{X}, y \in \mathcal{Y}} \frac{\mathrm{p}_{te}(x,y)}{\mathrm{p}_{tr}(x,y)}. \tag{24}$$

This way, the methods proposed can achieve an effective sample size $D$ times larger using the double-weighting given by (10) with $C = B/\sqrt{D}$. This is achieved at the cost of using classification rules with confidence in a subregion of $\mathcal{X} \times \mathcal{Y}$, since the region where $\alpha(x,y)$ is significantly large shrinks when $C$ decreases (i.e., when $D$ increases). Considering the ratio

$$\frac{\min_{x \in \mathcal{X}, y \in \mathcal{Y}} \alpha(x,y)}{\max_{x \in \mathcal{X}, y \in \mathcal{Y}} \alpha(x,y)} \tag{25}$$

we have that using weights as in (10) the ratio is smaller than one, while if we consider weights given by reweighted in (9) that ratio is one since all testing samples have the same confidence. Ratios in (25) significantly smaller than one correspond to situations where some points are predicted with very low confidence. This reduction in prediction confidence for points $(x,y)$ with $\mathrm{p}_{tr}(x,y) \ll \mathrm{p}_{te}(x,y)$ reflects the scarcity of training samples in such regions of $\mathcal{X} \times \mathcal{Y}$.

### 3.4 Double-weighting for different distribution shifts

This subsection describes how to apply the unified double-weighting framework presented in Section 3 to different types of distribution shifts. Specifically, in this section we describe the learning methodology using double-weighting under single-source covariate and label shift.

### 3.4.1 Single-source covariate shift

In cases where $\mathrm{p}_{\mathrm{te}}(y|x) = \mathrm{p}_{\mathrm{tr}}(y|x)$, the reference weights in (10) simplify to

$$\alpha(x) = \min\left(1, \frac{\mathrm{p}_{\mathrm{tr}}(x)}{\mathrm{p}_{\mathrm{te}}(x)}C\right), \ \ \beta(x) = \min\left(\frac{\mathrm{p}_{\mathrm{te}}(x)}{\mathrm{p}_{\mathrm{tr}}(x)}, C\right) \tag{26}$$

for any $C > 0$, so that the weights depend only on the covariates $x$.

The double-weighting (DW) approach has been recently proposed to deal with extreme weight values in single-source covariate shift adaptation by utilizing weights for both training and testing samples (Segovia-Martín et al., 2023).

### 3.4.2 Single-source label shift

In cases where $\mathrm{p}_{\mathrm{te}}(x|y) = \mathrm{p}_{\mathrm{tr}}(x|y)$, the weights in (10) simplify to

$$\alpha(y) = \min\left(1, \frac{\mathrm{p}_{\mathrm{tr}}(y)}{\mathrm{p}_{\mathrm{te}}(y)}C\right), \ \ \beta(y) = \min\left(\frac{\mathrm{p}_{\mathrm{te}}(y)}{\mathrm{p}_{\mathrm{tr}}(y)}, C\right) \tag{27}$$

for any $C > 0$, so that the weights depend only on the labels $y$.

Weights as in (27) can alleviate the limitations of existing reweighted label shift approaches. Considering both weights $\beta(y)$ and $\alpha(y)$ we can assign low relevance to the training samples with labels that are unlikely at testing, and also assign low confidence prediction to the labels that are underrepresented in the training data.

**Learning MRCs.** We can derive the learning framework for label shift adaptation using the learning methodology presented in Section 3.2 by considering that the weight functions $\alpha(y)$ and $\beta(y)$ only depend on the labels, simplifying the general distribution shift framework to address label shift scenarios.

The classification rules associated to the MRCs corresponding with the uncertainty set $\mathcal{U}$ in (11) with weight functions $\alpha(y)$ and $\beta(y)$ are given by

$$\mathrm{h}^{\mathcal{U}}(y|x) = \left(\alpha(y)\Phi(x,y)^{\mathrm{T}}\boldsymbol{\mu}^* - \max_{\mathcal{C} \subseteq \mathcal{Y}} \frac{\sum_{y' \in \mathcal{C}}\left(\alpha(y')\Phi(x,y')^{\mathrm{T}}\boldsymbol{\mu}^*\right) - 1}{|\mathcal{C}|}\right)_+ \tag{28}$$

for 0-1-loss and

$$\mathrm{h}^{\mathcal{U}}(y|x) = \frac{\exp\{\alpha(y)\Phi(x,y)^{\mathrm{T}}\boldsymbol{\mu}^*\}}{\sum_{y' \in \mathcal{Y}}\exp\left\{\alpha(y')\Phi(x,y')^{\mathrm{T}}\boldsymbol{\mu}^*\right\}} \tag{29}$$

for log-loss, where $\boldsymbol{\mu}^*$ is a solution of (13) considering weigh functions that only depend on the labels.

**Predictive per label confidence.** The values of $\alpha(y)$ adjust the confidence with which each label is classified. Low values of $\alpha(y)$ reflect lower confidence in the classification of certain labels, meaning a poor learning of those labels during training due to their underrepresentation in the training data. The components of the parameter $\boldsymbol{\mu}$ associated to the labels with low value $\alpha(y)$ will be less reliable, since the feature components are poorly estimated. Low values of $\alpha(y)$ imply that the classifier would not assign label $y$ regardless of the instance we want to classify.

In this section, we described the unified learning framework for double-weighting for single-source distribution shift adaptation. In the next section, we will extend the unified learning framework for multi-source distribution shift adaptation.

## 4 Unified Double-Weighting Framework for Multi-Source Distribution Shifts

This section describes the unified learning framework of double-weighting for multi-source distribution shift adaptation. We also present generalization bounds for the proposed framework in comparison with the reweighted approach.

### 4.1 Double-weighting pairwise distributions

The proposed framework extends that presented in Section 3.1 for the case where we have multiple training sources. We match multiple training distributions $p_1, p_2, \ldots, p_S$ with the testing distribution $p_{te}$ (see Fig. 2), exploiting the fact that, for any function $f$, we have that

$$\mathbb{E}_{p_{te}(x,y)} \alpha_s(x,y) f(x,y) = \mathbb{E}_{p_s(x,y)} \beta_s(x,y) f(x,y) \tag{30}$$

for $s \in [S]$, can be attained by multiple choices of the sets of weights $\{\alpha_s(x,y)\}_{s \in [S]}$ and $\{\beta_s(x,y)\}_{s \in [S]}$. In this paper, we propose the usage of weights tailored to general scenarios of multi-source distribution shift as follows

$$\alpha_s(x,y) = \min\left(\frac{p_s(x,y)}{\sum_{s'=1}^{S} p_{s'}(x,y)}, \frac{p_s(x,y)}{p_{te}(x,y)} C\right), \quad \beta_s(x,y) = \min\left(\frac{p_{te}(x,y)}{\sum_{s'=1}^{S} p_{s'}(x,y)}, C\right) \tag{31}$$

for any $C > 0$. Such weights satisfy (30) and also address the limitations of reweighted approaches because if the ratio $p_{te}(x,y)/\sum_{s=1}^{S} p_s(x,y)$ take large values, using small $\alpha_s(x,y)$ can enable having (30) with moderate values of $\beta_s(x)$. In particular, for single-source distribution shift, the double-weighting approach presented in (10) can alleviate the limitations of reweighted approaches by sacrificing confidence in samples such that $p_{te}(x,y) > C p_s(x,y)$. For the multi-source case, the same improvements can be achieved by only sacrificing confidence in the points such that $p_{te}(x,y) > C \sum_{s=1}^{S} p_s(x,y)$ that is a much smaller set because $\sum_{s=1}^{S} p_s(x,y) > p_{s'}(x,y)$ for all $s' \in [S]$. Using the same $C$ as in Section 3.1, we alleviate the existing trade-off between expectation estimates and confidence in the classification rules because we do not need to have significant confidence in each source $s \in [S]$. Instead, we need that $\sum_{s=1}^{S} \alpha_s(x,y) \approx 1$, that is achieved since the weights in (31) take into account the rest of the sources. This allows us to consider smaller sets of weights $\{\beta_s(x)\}_{s \in [S]}$ without losing confidence in the classifiers. Specifically, weights as in (31) can achieve higher confidence predictions in comparison with weights as in (10) for the testing instances such that $p_s(x,y)$ take large values for multiple $s \in [S]$.

The usage of weights $\{\alpha_s(x,y)\}_{s=1}^{S}$ and $\{\beta_s(x,y)\}_{s=1}^{S}$ tailored to multi-source scenarios offers a significant improvement over methods based on the reweighted approach and the double weighting for single-source distribution shift. The proposed approach can handle general multi-source distribution shifts. In addition, this approach results in enhanced generalization, as shown in Theorem 4.2.

### 4.2 MRC learning using general multi-source double-weighting

This subsection describes the proposed learning methodology for the unified view of double-weighting using sets of weights $\{\alpha_s(x,y)\}_{s=1}^{S}$ and $\{\beta_s(x,y)\}_{s=1}^{S}$. To simplify the exposition, we assume that all training sources have an equal number of training samples. In Appendix D, we extend the paper's results for cases with a general number of training samples in each source.

**Uncertainty sets.** To address the multi-source distribution shift, we construct an uncertainty set $\mathcal{U}$ defined in terms of the intersection of $S$ sets of constraints. Each set of constraints bounds the expectation of the weighted feature $\Phi_{\alpha_s}(x,y)$ for each training source $s \in [S]$. Then, the uncertainty set $\mathcal{U}$ can be defined as

$$\mathcal{U} = \left\{ p \in \Delta(\mathcal{X} \times \mathcal{Y}) : |\mathbb{E}_p \Phi_{\alpha_s}(x,y) - \boldsymbol{\tau}_s| \preceq \boldsymbol{\lambda}_s \text{ for } s = 1, 2, \ldots, S \text{ and } p(x) = p_{te}(x), \forall x \in \mathcal{X} \right\} \tag{32}$$

where the expectation of feature mapping $\Phi_{\alpha_s}(x,y) = \alpha_s(x,y)\Phi(x,y)$, for $s \in [S]$, is estimated using sample averages of training samples from source $s$ weighted by $\beta_s(x,y)$ as

$$\boldsymbol{\tau}_s = \frac{1}{n} \sum_{i=1}^{n} \Phi_{\beta_s}(x_{s,i}, y_{s,i}), \text{ for } \Phi_{\beta_s}(x,y) = \beta_s(x,y)\Phi(x,y). \tag{33}$$

The proposed uncertainty set $\mathcal{U}$, defined in (32), is the intersection of $S$ sets of constraints, each of which characterizing weighted feature expectation matching on $\Phi_{\alpha_s}(x,y)$ in each source domain with weights $\alpha_s(x,y)$. This set is contained in each uncertainty set $\mathcal{U}_s$ defined using (11) for each source $s \in [S]$ and is generally significantly smaller.

**Convex optimization.** MRCs corresponding with the uncertainty set (32) can be learned by solving the convex optimization problem

$$\min_{\{\boldsymbol{\mu}_s\}_{s=1}^S} -\sum_{s=1}^{S} \boldsymbol{\tau}_s^{\mathrm{T}} \boldsymbol{\mu}_s + \sum_{s=1}^{S} \boldsymbol{\lambda}_s^{\mathrm{T}} |\boldsymbol{\mu}_s| + \mathbb{E}_{\mathrm{p}_{\mathrm{te}}(x)} \varphi_\ell \left(\{\boldsymbol{\mu}_s\}_{s=1}^S, x, \{\alpha_s\}_{s=1}^S\right) \tag{34}$$

where $\varphi_\ell$ is a function defined in terms of the loss. For 0-1-loss, we have

$$\varphi_{01} \left(\{\boldsymbol{\mu}_s\}_{s=1}^S, x, \{\alpha_s\}_{s=1}^S\right) = 1 + \max_{\mathcal{C} \subseteq \mathcal{Y}} \frac{\sum_{s=1}^{S} \sum_{y \in \mathcal{C}} \Phi_{\alpha_s}(x, y)^{\mathrm{T}} \boldsymbol{\mu}_s - 1}{|\mathcal{C}|} \tag{35}$$

and, for log-loss, we have

$$\varphi_{\log} \left(\{\boldsymbol{\mu}_s\}_{s=1}^S, x, \{\alpha_s\}_{s=1}^S\right) = \log \sum_{y \in \mathcal{Y}} \exp \left\{\sum_{s=1}^{S} \Phi_{\alpha_s}(x, y)^{\mathrm{T}} \boldsymbol{\mu}_s\right\} \tag{36}$$

**Theorem 4.1.** *Let $\boldsymbol{\tau}_s \in \mathbb{R}^m$, $\boldsymbol{\lambda}_s \in \mathbb{R}^m$ for $s \in [S]$ be such that the uncertainty set $\mathcal{U}$ in (32) is not the empty set. If $\{\boldsymbol{\mu}_s^*\}_{s=1}^S$ is a solution of (34) for 0-1-loss, the classification rule*

$$\mathrm{h}^{\mathcal{U}}(y|x) = \left(\sum_{s=1}^{S} \Phi_{\alpha_s}(x, y)^T \boldsymbol{\mu}_s^* - \max_{\mathcal{C} \subseteq \mathcal{Y}} \frac{\sum_{s=1}^{S} \sum_{y' \in \mathcal{C}} \Phi_{\alpha_s}(x, y')^T \boldsymbol{\mu}_s^* - 1}{|\mathcal{C}|}\right)_+ \tag{37}$$

*is a 0-1-MRC for $\mathcal{U}$. If $\{\boldsymbol{\mu}_s^*\}_{s=1}^S$ is a solutions of (34) for log-loss, the classification rule*

$$\mathrm{h}^{\mathcal{U}}(y|x) = \frac{\exp\{\sum_{s=1}^{S} \Phi_{\alpha_s}(x, y)^T \boldsymbol{\mu}_s^*\}}{\sum_{y' \in \mathcal{Y}} \exp\left\{\sum_{s=1}^{S} \Phi_{\alpha_s}(x, y')^T \boldsymbol{\mu}_s^*\right\}} \tag{38}$$

*is a log-MRC for $\mathcal{U}$. In addition, the minimax risk $R(\mathcal{U})$ is given by*

$$-\sum_{s=1}^{S} \boldsymbol{\tau}_s^T \boldsymbol{\mu}_s^* + \sum_{s=1}^{S} \boldsymbol{\lambda}_s^T |\boldsymbol{\mu}_s^*| + \mathbb{E}_{\mathrm{p}_{te}(x)} \varphi_\ell \left(\{\boldsymbol{\mu}_s^*\}_{s=1}^S, x, \{\alpha_s\}_{s=1}^S\right). \tag{39}$$

*Proof.* See Appendix C. □

The theorem above is a generalization of Theorem 3.1 for the multi-source case. The classification rules in (37) for 0-1-loss, and (38) for log-loss involve sample specific weighted combinations of the feature mappings given by $\{\alpha_s(x, y)\}_{s \in [S]}$. This way, we allow different contributions from each source to classify each testing instance, as detailed in the following subsections.

**Source-dependent weights for training and testing.** We obtain the benefits of the double-weighting approach in the previous section by considering both weights $\beta_s(x, y)$ and $\alpha_s(x, y)$ for each pair of training and testing distribution, $\mathrm{p}_s(x, y)$ and $\mathrm{p}_{\mathrm{te}}(x, y)$. We can assign low relevance to training samples that are unlikely at testing and also assign low confidence to the testing instances that are unlikely at training. In addition, since the uncertainty set $\mathcal{U}$ in (32) is contained in each uncertainty set $\mathcal{U}_s$ defined using (11) for each source $s \in [S]$ we have that

$$\min_{\mathrm{h}} \max_{\mathrm{p} \in \mathcal{U}} \ell(\mathrm{h}, \mathrm{p}) \leq \min_{s \in [S]} \min_{\mathrm{h}} \max_{\mathrm{p} \in \mathcal{U}_s} \ell(\mathrm{h}, \mathrm{p}) \tag{40}$$

and $R(\mathcal{U}) \leq \min_{s \in [S]} R(\mathcal{U}_s)$. Therefore, the minimax risk of the proposed multi-source method is smaller than the minimax risk of the single-source method. This fact also differentiates the proposed approach from "summing over the sources," which involves obtaining an MRC associated with the uncertainty set $\mathcal{U}_s$ for each source $s \in [S]$ and then combining the classifiers. Directly applying Section 3.1 and Section 3.2 to each source will not lead to the minimax risk $R(\mathcal{U})$.

**Sample-specific weights for classification rules.** Considering the set of weights $\{\alpha_s(x,y)\}_{s\in[S]}$, the prediction of each testing instance is obtained by combining the relevant feature mappings, leading to classification rules that involve sample-specific combinations of feature mappings given by $\{\alpha_s(x,y)\}_{s\in[S]}$. Since the set $\{\alpha_s(x,y)\}_{s\in[S]}$ depends on $x$ and $y$, we allow different contributions from each source to classify each testing instance. This way, the proposed learning methodology is capable of learning complex classification rules to classify each instance with high confidence. Even if some $\alpha_s(x,y)$ are small for a particular source $s$ and pair $(x,y)$, we may be able to still achieve high confidence using the information from other sources. In particular, we can leverage other $\alpha_{s'}(x,y)$ to classify that instance as long as we have $\sum_{s=1}^{S}\alpha_s(x,y)\approx 1$.

### 4.3 Generalization bounds

This subsection describes the generalization bounds of the proposed methods. Such bounds are given in terms of the smallest minimax risk, $R^\infty$, that corresponds with the uncertainty set given by the exact expectations, and is defined by

$$R^\infty = \min_{\{\boldsymbol{\mu}_s\}_{s=1}^{S}} -\mathbb{E}_{\mathrm{p_{te}}(x,y)}\sum_{s=1}^{S}\Phi_{\alpha_s}(x,y)^{\mathrm{T}}\boldsymbol{\mu}_s + \mathbb{E}_{\mathrm{p_{te}}(x)}\varphi_\ell\left(\{\boldsymbol{\mu}_s\}_{s=1}^{S}, x, \{\alpha_s\}_{s=1}^{S}\right). \tag{41}$$

The MRC corresponding to that smallest minimax risk $R^\infty$ could only be obtained by an exact estimation of the expectations of the feature mapping $\Phi_{\alpha_s}$, for $s\in[S]$, that in turn would require an infinite amount of training samples from each source $s\in[S]$. The theorem below provides risk bounds for the proposed MRCs in terms of smallest minimax risks $R^\infty$, showing how the proposed methods can lead to a significant decrease in the estimation error compared to existing methods.

**Theorem 4.2.** *Let $\mathcal{U}$ be a non-empty uncertainty set given by (32) and $\mathrm{h}^{\mathcal{U}}$ be an $\ell$-MRC for $\mathcal{U}$. If weights $\{\alpha_s(x,y)\}_{s\in[S]}$ and $\{\beta_s(x,y)\}_{s\in[S]}$ are given by (31) with $C = B/\sqrt{D}$ for $D \geq 1$ and*

$$B = \sup_{x\in\mathcal{X},y\in\mathcal{Y}} \frac{\mathrm{p}_{te}(x,y)}{\sum_{i=1}^{S}\mathrm{p}_i(x,y)}. \tag{42}$$

*Then, with probability at least $1-\delta$ we have that*

$$R(\mathrm{h}^{\mathcal{U}}) \leq R^\infty + \sum_{s=1}^{S}\boldsymbol{\lambda}_s^T\left(|\boldsymbol{\mu}_s^\infty| - |\boldsymbol{\mu}_s^*|\right) + \|\boldsymbol{\mu}^\infty - \boldsymbol{\mu}^*\|_1 \max_{s\in[S]}\|\boldsymbol{\tau}_s - \mathbb{E}_{\mathrm{p_{te}}}\Phi_{\alpha_s}(x,y)\|_\infty$$

$$\leq R^\infty + \sum_{s=1}^{S}\boldsymbol{\lambda}_s^T\left(|\boldsymbol{\mu}_s^\infty| - |\boldsymbol{\mu}_s^*|\right) + M\|\boldsymbol{\mu}^\infty - \boldsymbol{\mu}^*\|_1\sqrt{\frac{2B^2}{Dn}\log\frac{2m}{\delta}} \tag{43}$$

*where $\boldsymbol{\mu}^\infty = [\boldsymbol{\mu}_1^\infty, \dots, \boldsymbol{\mu}_S^\infty]$, $\boldsymbol{\mu}^* = [\boldsymbol{\mu}_1^*, \dots, \boldsymbol{\mu}_S^*]$ are solutions to (39) and (41), respectively, $n$ is the number of training samples available in each domain $s \in [S]$, and $M$ is a constant satisfying $\|\Phi(x,y)\|_\infty \leq M$ for all $x \in \mathcal{X}$, $y \in \mathcal{Y}$.*

*Proof.* See Appendix C. $\qquad\qquad\square$

Note that, similarly to Theorem 3.2, the difference between the risk $R(\mathrm{h}^{\mathcal{U}})$ and the smallest minimax risk $R^\infty$ decreases as the maximum of the estimation errors over the training sources $s \in [S]$, $\max_{s\in[S]}\|\boldsymbol{\tau}_s - \mathbb{E}_{\mathrm{p_{te}}}\Phi_\alpha(x,y)\|_\infty$, decreases. The set of weights $\{\alpha_s(x,y)\}_{s\in[S]}$ and $\{\beta_s(x,y)\}_{s\in[S]}$ allow us to reduce the estimation error in $\|\boldsymbol{\tau}_s - \mathbb{E}_{\mathrm{p_{te}}}\Phi_{\alpha_s}(x,y)\|_\infty$, as detailed in the following section. We also detail how weights for multi-source in (31) further reduce the maximum value of the weights $\|\beta_s(x,y)\|$ for each source $s \in [S]$ in comparison with weights for single-source in (10).

### 4.4 Leveraging complementary information from multiple sources

In this subsection, we explain how the usage of complementary information among different sources leads to improved effective sample size. The bound in Theorem 4.2 depends on the estimate of $\mathbb{E}_{\mathrm{p_{te}}}\Phi_{\alpha_s}(x,y)$ and, as shown in Corollary 3.3, the error in the estimate depends on the values of the weights $\beta_s(x,y)$.

In Section 3.3 we show that the proposed single-source double-weighting can achieve an effective sample size $D$ times larger than reweighted approach considering weights given in (10). This was achieved at the cost of using classification rules with significant confidence in a subregion of $\mathcal{X}$ that shrinks when $D$ increases.

In multi-source cases we can improve the trade-off presented in Section 3.3 between the error in the estimation of $\mathbb{E}_{\mathrm{P_{te}}} \Phi_{\alpha_s}(x, y)$ and the confidence of the classification rules. In particular, using weights as in (31) with $C = B/\sqrt{D}$ and $B = \sup_{x \in \mathcal{X}, y \in \mathcal{Y}} \mathrm{p_{te}}(x, y) / \sum_{i=1}^{S} \mathrm{p}_i(x, y)$, we have that

$$\|\beta_s(x, y)\|_\infty = \frac{B}{\sqrt{D}} = \frac{1}{\sqrt{D}} \sup_{x \in \mathcal{X}, y \in \mathcal{Y}} \frac{\mathrm{p_{te}}(x, y)}{\sum_{i=1}^{S} \mathrm{p}_i(x, y)}. \tag{44}$$

In the following, we will compare the reference solution for multi-source double-weighting in (31) and the reference solution for single-source double-weighting in (10). If we consider weights $\beta_s(x, y)$ as in (10) with $\|\beta_s(x, y)\|_\infty = B_s/\sqrt{D}$ and $B_s$ defined in (20), we have that for any source $s$

$$B = \sup_{x \in \mathcal{X}, y \in \mathcal{Y}} \frac{\mathrm{p_{te}}(x, y)}{\sum_{i=1}^{S} \mathrm{p}_i(x, y)} \leq \sup_{x \in \mathcal{X}, y \in \mathcal{Y}} \frac{\mathrm{p_{te}}(x, y)}{\mathrm{p}_s(x, y)} = B_s \tag{45}$$

and often $B \ll B_s$ as long as the supports of the training sources significantly overlap ($\mathrm{p}_s(x, y) \gg 0$ for multiple $s \in [S]$). When we decrease $\|\beta_s(x, y)\|_\infty$, i.e., when we consider larger $D$, the region where the classifier has significant confidence also decreases. Similarly as in Section 3.3, the ratio

$$\frac{\min_{x \in \mathcal{X}, y \in \mathcal{Y}} \sum_{s=1}^{S} \alpha_s(x, y)}{\max_{x \in \mathcal{X}, y \in \mathcal{Y}} \sum_{s=1}^{S} \alpha_s(x, y)} \tag{46}$$

is $C(1/B)$ using weights for multi-source in (31), while using weights for single-source in (10) such ratio is $C((1/S) \sum_{s=1}^{S} 1/B_s)$, that is smaller than $C(1/B)$ since $B \leq B_s$ for any $s \in [S]$. This means that using weights as in (31) the region of $\mathcal{X} \times \mathcal{Y}$ where the confidence of the classifier is significantly large is bigger than if we use weights as in (10). Using the set of weights $\{\alpha_s(x, y)\}_{s=1}^{S}$ in (31) where each $\alpha_s(x, y)$ depend on all the sources $s \in [S]$, we can improve the trade-off between estimation error and confidence of the classifiers. Using weights as in (31), we do not need significant confidence $\alpha_s(x, y) \approx 1$ for each source $s \in [S]$ but rather that there is enough confidence among all sources, i.e., that $\sum_{s=1}^{S} \alpha_s(x, y) \approx 1$.

### 4.5 Double-weighting for different multi-source distribution shifts

This subsection describes how to apply the unified multi-source double-weighting framework presented in Section 4 to different types of distribution shifts. Specifically, we describe the learning methodology using double-weighting under multi-source covariate and label shift.

#### 4.5.1 Multi-source covariate shift

In a scenario with multi-source covariate shift we have that $\mathrm{p_{te}}(y|x) = \mathrm{p}_s(y|x)$ for $s \in [S]$. Hence, the reference weights in (31) simplify to

$$\alpha_s(x) = \min \left( \frac{\mathrm{p}_s(x)}{\sum_{i=1}^{S} \mathrm{p}_i(x)}, \frac{\mathrm{p}_s(x)}{\mathrm{p_{te}}(x)} C \right), \quad \beta_s(x) = \min \left( \frac{\mathrm{p_{te}}(x)}{\sum_{i=1}^{S} \mathrm{p}_i(x)}, C \right) \tag{47}$$

for any $C > 0$, so that the weights depend only on the covariates $x$.

The usage of weights $\{\alpha_s(x)\}_{s=1}^{S}$ and $\{\beta_s(x)\}_{s=1}^{S}$ tailored to multi-source covariate shift scenarios constitutes a substantial improvement compared to methods based on reweighted approach. It also constitutes a substantial improvement with respect to the double-weighting of a single-source covariate shift, since using weights as in (47) we alleviate the existing trade-off since we do not need to have significant confidence in each source $s \in [S]$. Instead, we need that $\sum_{s=1}^{S} \alpha_s(x, y) \approx 1$, that is achieved since the weights in (47) take into account the rest of the sources. This allows us to consider smaller sets of weights $\{\beta_s(x)\}_{s \in [S]}$ without losing confidence in the classifiers.

We can derive the learning for multi-source covariate shift adaptation using the learning methodology in Section 4.2 by considering that the sets of weight functions $\{\alpha_s(x)\}_{s\in[S]}$ and $\{\beta_s(x)\}_{s\in[S]}$ only depend on the covariates. This simplifies the general distribution shift framework to address multi-source covariate shift scenarios.

The classification rules associated to the MRCs corresponding with the uncertainty set $\mathcal{U}$ in (32) with weight functions $\{\alpha_s(x)\}_{s\in[S]}$ and $\{\beta_s(x)\}_{s\in[S]}$ are given by

$$\mathrm{h}^{\mathcal{U}}(y|x) = \Big(\sum_{s=1}^{S}\alpha_s(x)\Phi(x,y)^{\mathrm{T}}\boldsymbol{\mu}_s^* - \max_{\mathcal{C}\subseteq\mathcal{Y}}\frac{\sum_{s=1}^{S}\sum_{y'\in\mathcal{C}}\alpha_s(x)\Phi(x,y')^{\mathrm{T}}\boldsymbol{\mu}_s^* - 1}{|\mathcal{C}|}\Big)_+ \tag{48}$$

for 0-1-MRC and

$$\mathrm{h}^{\mathcal{U}}(y|x) = \frac{\exp\{\sum_{s=1}^{S}\alpha_s(x)\Phi(x,y)^{\mathrm{T}}\boldsymbol{\mu}^*\}}{\sum_{y'\in\mathcal{Y}}\exp\left\{\sum_{s=1}^{S}\alpha_s(x)\Phi(x,y')^{\mathrm{T}}\boldsymbol{\mu}^*\right\}} \tag{49}$$

for log-loss, where $\{\boldsymbol{\mu}_s^*\}_{s\in[S]}$ is a solution of (34) considering sets of weight functions that only depend on the covariates.

**Predictive per source and per instance confidence.** The classification rules learned for multi-source covariate shift are given in terms of the set of weights $\{\alpha_s(x)\}_{s\in[S]}$. The weights $\alpha_s(x)$ adjust the confidence with which testing instance $x$ is classified using the contribution from the source $s$. Low values of $\alpha_s(x)$ imply that if we rely only on the information from source $s$, the classifier would uniformly assign labels in the set of labels $\mathcal{Y}$. By considering weights $\alpha_s(x)$ for each source, we can classify instance $x$ with high confidence even when some of the $\alpha_s(x)$ take small values, as long as $\sum_{s=1}^{S}\alpha_s(x)\approx 1$.

### 4.5.2 Multi-source label shift

In a scenario with multi-source label shift we have that $\mathrm{p}_{\mathrm{te}}(x|y) = \mathrm{p}_s(x|y)$ for $s \in [S]$. Hence, the reference weights in (31) simplify to

$$\alpha_s(y) = \min\left(\delta_s(y), \frac{\mathrm{p}_s(y)}{\mathrm{p}_{\mathrm{te}}(y)}C\right), \quad \beta_s(y) = \min\left(\delta_s(y)\frac{\mathrm{p}_{\mathrm{te}}(y)}{\mathrm{p}_s(y)}, C\right) \tag{50}$$

for any $C > 0$, where $\delta_s(y) = \mathrm{p}_s(y)/\sum_{i=1}^{S}\mathrm{p}_i(y)$, so that the weights depend only on the labels $y$.

The usage of weights $\{\alpha_s(y)\}_{s=1}^{S}$ and $\{\beta_s(y)\}_{s=1}^{S}$ tailored to multi-source label shift scenarios constitutes a substantial improvement compared to methods based on the reweighted approach. As for multi-source covariate shift, it also constitutes a substantial improvement with respect to the double-weighting of single-source label shift, since using weights as in (50) alleviate the existing trade-off between estimation errors and confidence of the classification rules, as detailed in Section 4.4.

As for multi-source covariate shift, we can simplify the general distribution shift framework in Section 4.2 to address multi-source label shift scenarios by considering that the sets of weight functions $\{\alpha_s(y)\}_{s\in[S]}$ and $\{\beta_s(y)\}_{s\in[S]}$ only depend on the labels.

The classification rules associated to the MRCs corresponding with the uncertainty set $\mathcal{U}$ in (32) with weight functions $\{\alpha_s(y)\}_{s\in[S]}$ and $\{\beta_s(y)\}_{s\in[S]}$ are given by

$$\mathrm{h}^{\mathcal{U}}(y|x) = \Big(\sum_{s=1}^{S}\alpha_s(y)\Phi(x,y)^{\mathrm{T}}\boldsymbol{\mu}_s^* - \max_{\mathcal{C}\subseteq\mathcal{Y}}\frac{\sum_{s=1}^{S}\sum_{y'\in\mathcal{C}}\alpha_s(y)\Phi(x,y')^{\mathrm{T}}\boldsymbol{\mu}_s^* - 1}{|\mathcal{C}|}\Big)_+ \tag{51}$$

for 0-1-MRC and

$$\mathrm{h}^{\mathcal{U}}(y|x) = \frac{\exp\{\sum_{s=1}^{S}\alpha_s(y)\Phi(x,y)^{\mathrm{T}}\boldsymbol{\mu}^*\}}{\sum_{y'\in\mathcal{Y}}\exp\left\{\sum_{s=1}^{S}\alpha_s(y')\Phi(x,y')^{\mathrm{T}}\boldsymbol{\mu}^*\right\}} \tag{52}$$

for log-loss, where $\{\boldsymbol{\mu}_s^*\}_{s\in[S]}$ is a solution of (34) considering sets of weight functions that only depend on the labels.

**Predictive per source and per label confidence.** The classification rules learned for multi-source label shift are given in terms of the set of weights $\{\alpha_s(y)\}_{s \in [S]}$. The weights $\alpha_s(y)$ adjust the confidence with which label $y$ is classified using the contribution from the source $s$. Low values of $\alpha_s(y)$ imply that if we rely only on the information from source $s$, the classifier would never assign label $y$ regardless of the instance we want to classify. By considering weights $\alpha_s(y)$ for each source, we can learn how to classify with label $y$ with high confidence even when some of the $\alpha_s(y)$ take small values, as long as $\sum_{s=1}^{S} \alpha_s(y) \approx 1$.

**Addressing other types of shifts.** The unified framework proposed above can be used to adapt to multiple types of distribution shifts, such as conditional shifts or even a general distribution shift. Even thought the double weighting framework is general enough to deal with any shift, the estimation of weights for conditional shift usually requires specific methods to estimate the conditional relations between instances and labels, that, in practice, would require multiple assumptions regarding the distributions. This additional difficulty is common in the literature. For instance, the methods in (Schweikert et al., 2008; Sun et al., 2011; Chattopadhyay et al., 2012) assume that labeled examples of the test distribution are known, while those in (Zhang et al., 2013; 2015) assume a specific relationship between the conditional distributions.

In this section, we described the unified learning framework for double-weighting for multi-source distribution shift adaptation. In the next section, we will present the algorithms and the implementation of the proposed double-weighting for specific distributions shifts.

# 5 Practical Algorithm and Implementation

This section describes the implementation of the proposed techniques for double-weighting label shift (DW-LS), double-weighting multi-source (MS) covariate shift (DW-MSCS), and double-weighting MS label shift (DW-MSLS) detailed in Algorithm 1, Algorithm 2 and Algorithm 3, respectively.

## 5.1 Practical algorithm for double-weighting label shift

Algorithm 1 shows the implementation of the proposed techniques for DW-LS that is further described in the following. In this section, we assume that $n$ training samples $(x_1, y_1), (x_2, y_2), \ldots, (x_n, y_n)$ and $t$ testing instances $x_{n+1}, x_{n+2}, \ldots, x_{n+t}$ are available at learning.

---

**Algorithm 1** The proposed algorithm for label shift adaptation: DW-LS

---

**Input:**    Training samples $(x_1, y_1), (x_2, y_2) \ldots, (x_n, y_n)$
           Testing instances $x_{n+1}, x_{n+2}, \ldots, x_{n+t}$
           Hyperparameters $\boldsymbol{\lambda}$ and $D$
**Output:** Weights $\hat{\boldsymbol{\beta}}$ and $\hat{\boldsymbol{\alpha}}$
           Classifier parameter $\boldsymbol{\mu}^*$, and Minimax risk $R(\mathcal{U})$
1: $\hat{p}_{te}(y)/\hat{p}_{tr}(y) \leftarrow$ solution of (53)
2: $\hat{\boldsymbol{\beta}}, \hat{\boldsymbol{\alpha}} \leftarrow$ reference solution in (54)
3: $\boldsymbol{\tau} \leftarrow \frac{1}{n} \sum_{i=1}^{n} \hat{\beta}(y_i) \Phi(x_i, y_i)$
4: $\boldsymbol{\mu}^* \leftarrow$ solution of (55)
5: $R(\mathcal{U}) \leftarrow$ optimum value of (55)

---

**Computing weights.** We obtain weights $\boldsymbol{\alpha}$ and $\boldsymbol{\beta}$ considering the reference solution in (27). The weights are obtained using the ratio $\omega(y) := p_{te}(y)/p_{tr}(y)$, for $y \in \mathcal{Y}$, that can be estimated using multiple methods (Lipton et al., 2018; Azizzadenesheli et al., 2019; Garg et al., 2020). In particular the method

in (Zhang et al., 2013) estimates such ratio by solving the optimization problem

$$\min_{\boldsymbol{\omega}} \left\| \frac{1}{t} \sum_{i=1}^{t} K_x(x_{n+i}) - \hat{U}_{\mathcal{X}|\mathcal{Y}} \frac{1}{n} \sum_{i=1}^{n} \boldsymbol{e}_{y_i} \boldsymbol{\omega} K_y(y_i) \right\|_{\mathcal{H}_x}^2$$

$$\text{s.t. } 0 \le \boldsymbol{e}_{y_i} \boldsymbol{\omega} \le B, \text{ for } i = 1, 2, \ldots, n$$

$$\left| \frac{1}{n} \sum_{i=1}^{n} \boldsymbol{e}_{y_i} \boldsymbol{\omega} - 1 \right| \le \epsilon \tag{53}$$

where $K_x : \mathcal{X} \longrightarrow \mathcal{H}$ is a feature map corresponding with a reproducing kernel Hilbert space (RKHS) $\mathcal{H}_x$ with kernel $k_x(x, \bar{x}) = \langle K_x(x), K_x(\bar{x}) \rangle_{\mathcal{H}_x}$, $K_y : \mathcal{Y} \longrightarrow \mathcal{H}_y$ is a feature map corresponding with a RKHS $\mathcal{H}_y$ with kernel $k_y(y, \bar{y}) = \langle K_y(y), K_y(\bar{y}) \rangle_{\mathcal{H}_y}$, and $\hat{U}_{\mathcal{X}|\mathcal{Y}}$ is an estimation of the operator that maps $\mathcal{H}_y$ into $\mathcal{H}_x$, representing the conditional embedding of $p(x|y)$. The optimization problem (53) minimizes the discrepancy between two empirical means of distributions $p_{\text{tr}}$ and $p_{\text{te}}$ considering embeddings of conditional distributions, studied in (Fukumizu et al., 2004; Song et al., 2009; 2010). The minimization problem in (53) can be written as a standard quadratic problem that can be solved by applying standard techniques. In particular, the parameters related to (53) are determined as proposed in (Song et al., 2009; Zhang et al., 2013).

Using the estimated ratios $\hat{p}_{\text{te}}(y)/\hat{p}_{\text{tr}}(y)$, the weights $\boldsymbol{\alpha}$ and $\boldsymbol{\beta}$ are computed using (27) as

$$\beta(y) = \min\left( \frac{\hat{p}_{\text{te}}(y)}{\hat{p}_{\text{tr}}(y)}, \frac{1}{\sqrt{D}} \max_{y' \in \mathcal{Y}} \frac{\hat{p}_{\text{te}}(y')}{\hat{p}_{\text{tr}}(y')} \right), \quad \alpha(y) = \min\left( \frac{\hat{p}_{\text{tr}}(y)}{\hat{p}_{\text{te}}(y)} \frac{1}{\sqrt{D}} \max_{y' \in \mathcal{Y}} \frac{\hat{p}_{\text{te}}(y')}{\hat{p}_{\text{tr}}(y')}, 1 \right) \tag{54}$$

for $y \in \mathcal{Y}$.

**Learning MRCs.** After computing the sets of weights, we solve the optimization problem in (13) by approximating the expectation term in (13) using the $t$ testing instances $x_{n+1}, x_{n+2}, \ldots, x_{n+t}$ and $n$ training samples $(x_1, y_1), (x_2, y_2), \ldots, (x_n, y_n)$, given (7), as

$$\min_{\boldsymbol{\mu}} -\boldsymbol{\tau}^{\text{T}} \boldsymbol{\mu} + \boldsymbol{\lambda}^{\text{T}} |\boldsymbol{\mu}| + \frac{1}{n+t} \left( \sum_{i=1}^{t} \varphi_\ell(\boldsymbol{\mu}, x_{n+i}, \boldsymbol{\alpha}) + \sum_{i=1}^{n} \frac{\beta(y_i)}{\alpha(y_i)} \varphi_\ell(\boldsymbol{\mu}, x_i, \boldsymbol{\alpha}) \right) \tag{55}$$

that is an unconstrained convex optimization problem and can be addressed in practice using conventional optimization methods such as stochastic subgradient methods.

We determine hyperparameters $D$ and $\boldsymbol{\lambda}$ following the same approach as in (Segovia-Martín et al., 2023), detailed in Appendix F.

**Complexity and implementation without testing instances.** The computational complexity of the proposed methods for label shift adaptation is similar to that of existing methods. In particular, the step of obtaining the parameters of the classifiers solving the convex optimization problem in (55) has similar complexity as that for conventional methods. The main difference is that existing reweighted methods estimate the expectation term in (13) utilizing the training samples available at learning. In our approach, if $t$ testing instances are available at learning, we utilize both training and testing instances to estimate the expectation term in (13), making use of the equality in (7).

### 5.2 Practical algorithm for double-weighting multi-source covariate shift

This section describes the implementation of the proposed DW-MSCS, detailed in Algorithm 2. In this section, we assume that $n_s$ training samples $(x_{s,1}, y_{s,1}), \ldots, (x_{s,n_s}, y_{s,n_s})$ from each source $s \in [S]$ and $t$ testing instances $x_{n+1}, x_{n+2}, \ldots, x_{n+t}$ are available at learning.
**Computing weights.** We present an extension of the kernel mean matching (KMM), multi-source KMM (MS-KMM), to determine weights $\{\boldsymbol{\beta}_s \in \mathbb{R}^{n_s}, \boldsymbol{\alpha}_s \in \mathbb{R}^t\}_{s \in [S]}$ for multiple sources by solving the optimization

---

**Algorithm 2** The proposed algorithm for multi-source covariate shift adaptation: DW-MSCS

---

**Input:**   Training samples $\{(x_{s,1}, y_{s,1}), \ldots, (x_{s,n_s}, y_{s,n_s})\}_{s \in [S]}$
  Testing instances $x_{n+1}, x_{n+2}, \ldots, x_{n+t}$
  Hyperparameters $\{\boldsymbol{\lambda}_s\}_{s=1}^{S}$ and $D$

**Output:**   Sets of weights $\{\hat{\boldsymbol{\beta}}_s\}_{s \in [S]}$ and $\{\hat{\boldsymbol{\alpha}}_s\}_{s \in [S]}$
  Classifier parameter $\{\boldsymbol{\mu}_s^*\}_{s=1}^{S}$, Minimax risk $R(\mathcal{U})$

1: $\{\hat{\boldsymbol{\beta}}_s\}_{s \in [S]}, \{\hat{\boldsymbol{\alpha}}_s\}_{s \in [S]} \leftarrow$ solution of (56)
2: $\boldsymbol{\tau}_s \leftarrow \frac{1}{n_s} \sum_{i=1}^{n_s} \hat{\beta}_s(x_{s,i}) \Phi(x_{s,i}, y_{s,i})$ for $s \in [S]$
3: $\{\boldsymbol{\mu}_s^*\}_{s \in [S]} \leftarrow$ solution of (57)
4: $R(\mathcal{U}) \leftarrow$ optimum value of (57)

---

problem

$$\min_{\{\boldsymbol{\beta}_s, \boldsymbol{\alpha}_s\}_{s=1}^{S}} \sum_{s=1}^{S} \left\| \sum_{i=1}^{t} \frac{\alpha_s^{(i)}}{t} K_x(x_{n+i}) - \sum_{i=1}^{n_s} \frac{\beta_s^{(i)}}{n_s} K_x(x_{s,i}) \right\|_{\mathcal{H}_x}^2$$

$$\text{s.t.} \quad 0 \le \beta_s^{(i)} \le B_s/\sqrt{D}, \text{ for } i \in [n_s], s \in [S]$$

$$0 \le \alpha_s^{(i)} \le 1, \text{ for } i \in [t], s \in [S]$$

$$\left| \frac{1}{n_s} \sum_{i=1}^{n_s} \beta_s^{(i)} - \frac{1}{t} \sum_{i=1}^{t} \alpha_s^{(i)} \right| \le \epsilon, \text{ for } s \in [S]$$

$$\left| \sum_{s=1}^{S} \alpha_s^{(i)} - 1 \right| \le \left(1 - \frac{1}{\sqrt{D}}\right), \text{ for } i \in [t] \tag{56}$$

where $K_x : \mathcal{X} \longrightarrow \mathcal{H}_x$ is a feature map corresponding with a RKHS $\mathcal{H}_x$ with kernel $k_x(x, \bar{x}) = \langle K_x(x), K_x(\bar{x}) \rangle_{\mathcal{H}}$. The optimization problem (56) minimizes the sum of the discrepancy between two empirical means of distributions $\mathrm{p}_s$ and $\mathrm{p}_{\text{te}}$, for $s \in [S]$, subject to multiple constraints. The hyperparameter $D \ge 1$ balances the confidence of the classification rules. The minimization problem in (56) can be written as a standard quadratic problem, as detailed in Appendix E, that can be solved by applying standard techniques. The appendix also shows a significant decrease in estimation error, similar to that of Theorem 4.2 for multi-source covariate shift.

**Learning MRCs.** After computing the sets of weights, we learn the parameters of the classifier in (48) for 0-1-loss (resp. (49) for log-loss) solving (34). We use the set of mean vectors $\{\boldsymbol{\tau}_s\}_{s \in [S]}$ defined in (33) and the set of confidence vector $\{\boldsymbol{\lambda}_s\}_{s \in [S]}$. We solve the optimization problem in (34) by approximating the expectation using the $t$ testing instances $x_{n+1}, x_{n+2}, \ldots, x_{n+t}$ as

$$\min_{\{\boldsymbol{\mu}_s\}_{s=1}^{S}} - \sum_{s=1}^{S} \boldsymbol{\tau}_s^{\mathrm{T}} \boldsymbol{\mu}_s + \boldsymbol{\lambda}_s^{\mathrm{T}} |\boldsymbol{\mu}_s| + \frac{1}{t} \sum_{i=1}^{t} \varphi_\ell \left( \{\boldsymbol{\mu}_s\}_{s=1}^{S} x_{n+i}, \{\boldsymbol{\alpha}_s\}_{s=1}^{S} \right) \tag{57}$$

that is an unconstrained convex optimization problem and can be addressed in practice using conventional optimization methods such as stochastic subgradient methods. We determine both hyperparameters $\{\boldsymbol{\lambda}_s\}_{s \in [S]}$ and $D$ following the same approach as in (Segovia-Martín et al., 2023), detailed in Appendix F.

**Complexity.** The computational complexity of the step for MS-KMM that obtains weights is similar to existing methods Sun et al. (2011), that compute KMM $S$ times. The main difference in the proposed methods is that (56) has $tS$ additional variables and $t(S + 1)$ additional constraints corresponding to the weights $\{\boldsymbol{\alpha}_s\}_{s=1}^{S}$. Solving the optimization in (57) to obtain the classifier parameters has similar complexity as that for conventional methods, since we consider $Sm$ variables and conventional methods compute $S$ times a convex optimization problem with $m$ variables.

### 5.3 Practical algorithm for double-weighting multi-source label shift

This section describes the implementation of the proposed DW-MSLS, detailed in Algorithm 3. In this section, we assume that $n_s$ training samples $(x_{s,1}, y_{s,1}), (x_{s,2}, y_{s,2}), \ldots, (x_{s,n_s}, y_{s,n_s})$ from each source $s \in [S]$ and $t$ testing instances $x_{n+1}, x_{n+2}, \ldots, x_{n+t}$ are available at learning.

---

**Algorithm 3** The proposed algorithm for multi-source label shift adaptation: DW-MSCS

---

**Input:**    Training samples $\{(x_{s,1}, y_{s,1}), \ldots, (x_{s,n_s}, y_{s,n_s})\}_{s \in [S]}$
           Testing instances $x_{n+1}, x_{n+2}, \ldots, x_{n+t}$
           Hyperparameters $\{\boldsymbol{\lambda}_s\}_{s=1}^S$ and $D$
**Output:** Sets of weights $\{\hat{\boldsymbol{\beta}}_s\}_{s \in [S]}$ and $\{\hat{\boldsymbol{\alpha}}_s\}_{s \in [S]}$
           Classifier parameter $\{\boldsymbol{\mu}_s^*\}_{s=1}^S$, and Minimax risk $R(\mathcal{U})$
1: $\hat{\mathrm{p}}_{\mathrm{te}}(y)/\hat{\mathrm{p}}_s(y) \leftarrow$ solution of (53) for $s \in [S]$
2: $\hat{\boldsymbol{\beta}}_s, \hat{\boldsymbol{\alpha}}_s \leftarrow$ reference solution in (58)
3: $\boldsymbol{\tau}_s \leftarrow \frac{1}{n_s} \sum_{i=1}^{n_s} \hat{\beta}_s(y_i) \Phi(x_{s,i}, y_{s,i})$ for $s \in [S]$
4: $\{\boldsymbol{\mu}_s^*\}_{s \in [S]} \leftarrow$ solution of (59)
5: $R(\mathcal{U}) \leftarrow$ optimum value of (59)

---

**Computing weights.** We obtain the set of weights $\{\boldsymbol{\alpha}_s\}_{s \in [S]}$ and $\{\boldsymbol{\beta}_s\}_{s \in [S]}$ considering the reference solution in (50). The weights are obtained estimating the ratios $\mathrm{p}_{\mathrm{te}}(y)/\mathrm{p}_s(y)$ for each source $s \in [S]$, and the ratios are estimated solving (53) for each source $s \in [S]$. The set of weights $\{\boldsymbol{\alpha}_s\}_{s \in [S]}$ and $\{\boldsymbol{\beta}_s\}_{s \in [S]}$ are computed as

$$\beta_s(y) = \min \left( \frac{\hat{\mathrm{p}}_{\mathrm{te}}(y)}{\sum_{s'=1}^S \hat{\mathrm{p}}_{s'}(y)}, \frac{1}{\sqrt{D}} \max_{y' \in \mathcal{Y}} \frac{\hat{\mathrm{p}}_{\mathrm{te}}(y')}{\sum_{s'=1}^S \hat{\mathrm{p}}_{s'}(y')} \right)$$

$$\alpha_s(y) = \min \left( \frac{\hat{\mathrm{p}}_s(y)}{\sum_{s'=1}^S \hat{\mathrm{p}}_{s'}(y)}, \frac{\hat{\mathrm{p}}_s(y)}{\sqrt{D}} \max_{y' \in \mathcal{Y}} \frac{1}{\sum_{s'=1}^S \hat{\mathrm{p}}_{s'}(y')} \right) \tag{58}$$

for $y \in \mathcal{Y}$ and $s \in [S]$.

**Learning MRCs.** After computing the sets of weights, we learn the parameters of the classifier in (51) for 0-1-loss (resp. (52) for log-loss). We solve the optimization problem in (34) by approximating the expectation using the $t$ testing instances $x_{n+1}, x_{n+2}, \ldots, x_{n+t}$ as

$$\min_{\{\boldsymbol{\mu}_s\}_{s \in [S]}} -\sum_{s=1}^S \boldsymbol{\tau}_s^{\mathrm{T}} \boldsymbol{\mu}_s + \sum_{s=1}^S \boldsymbol{\lambda}_s^{\mathrm{T}} |\boldsymbol{\mu}_s| + \frac{1}{t} \sum_{i=1}^t \varphi_\ell \left( \{\boldsymbol{\mu}_s\}_{s \in [S]}, x_{n+i}, \{\boldsymbol{\alpha}_s\}_{s \in [S]} \right) \tag{59}$$

that is an unconstrained convex optimization problem and can be addressed in practice using conventional optimization methods such as stochastic subgradient methods. We determine hyperparameters $D$ and $\{\boldsymbol{\lambda}\}_{s=1}^S$ following the same approach as in for Algorithm 2, detailed in Appendix F.

**Complexity and implementation without testing instances.** The computational complexity of the proposed methods for multi-source label shift adaptation is similar to existing methods. In particular, the computational complexity of the step for KMM in (53) that obtains weights is similar to existing methods Zhang et al. (2015), that compute all the weights in a single optimization problem.

Solving the optimization in (57) to obtain the classifier parameters has similar complexity as that for conventional methods, since we consider $Sm$ variables and conventional methods compute $S$ times a convex optimization problem with $m$ variables. The main difference is that existing reweighted methods estimate the expectation term in (34) utilizing the training samples available at learning. In our approach, if $t$ testing instances are available at learning, we utilize the testing instances to estimate the expectation term in (34).

**Remark.** The proposed methodology can be utilized with general types of feature mappings, including those defined using neural networks. For instance, the feature mapping $\Phi(x, y)$ can be given by the output of the penultimate layer $g(\cdot, \theta)$ of a neural network, i.e., $\Phi(x, y) = e_y \otimes g(x, \theta)$, where $\theta$ correspond to the weights

of the network. By feeding the data through the network, the outputs of the penultimate layer provide a high-level representation of the input data, capturing complex patterns and relationships learned during the training of the network. This way, we can deal with shifts in the semantic representation space. We have used pretrained ResNets to map images into feature vectors in the new numerical results, as detailed in the general responses. The results presented in Section 6.2 show how the proposed methodology can accommodate deep learning frameworks, obtaining good performances dealing with multi-source covariate shift.

# 6 Experiments

This section shows experimental results for the proposed approaches in comparison with the state-of-the-art methods. The source code for the methods and the experimental setup presented are publicly available in `https://github.com/MachineLearningBCAM/Unified-Double-Weighting-TMLR-2025`. The results are complemented by those in Appendix F that provide further implementation details.

## 6.1 Experiments for single-source label shift adaptation

This subsection shows experimental results for the proposed approach for label shift adaptation.

**Baseline methods.** As baseline, we employ logistic regression (No Adapt.), which ignore label shift. We also consider four label shift adaptation methods: Reweighted where we perform importance weighting using the true marginal ratios; DW where we perform double-weighting using the true marginal ratios to compute the reference solution in (27); target shift (TarS) (Zhang et al., 2013) that obtain weights solving a KMM problem; black box shift estimation (BBSE) (Lipton et al., 2018) that computes marginal ratios using an invertible confusion matrix; regularized learning under label shift (RLLS) (Azizzadenesheli et al., 2019) that adds a regularization term to the weight computation problem; and Maximum Likelihood Label Shift (MLLS) (Garg et al., 2020) that assumes access to a classifier that outputs the true source distribution conditional probabilities $p_{tr}(y|x)$.

**Experimental details.** The methods TarS (Zhang et al., 2013), BBSE (Lipton et al., 2018), RLLS (Azizzadenesheli et al., 2019), and MLLS (Garg et al., 2020) have been implemented as detailed in their references. For the proposed method, the corresponding hyperparameters are obtained following the approach described in (Segovia-Martín et al., 2023), as detailed in Appendix F.

### 6.1.1 Experiment with synthetic data

In the experiments using synthetic data, we show how the proposed approach can achieve label shift adaptation in a scenario where the existing reweighted approaches are challenged. For such results, the training and testing data are drawn from Gaussian distributions

$$p(x|y=1) = N\left(\mathbf{x}; [0.5, 0]^{T}, 0.1\mathbf{I}\right) \quad p(x|y=2) = N\left(\mathbf{x}; [-0.5, 0]^{T}, 0.1\mathbf{I}\right) \tag{60}$$

and labels are $y = 1$ if $x^{(1)}x^{(2)} \geq 0$, $y = 2$ otherwise. We fix $p_{te}(y = 1) = p_{te}(y = 2) = 0.50$ and we consider training distribution of the form $p_{tr}(y = 1) = \delta$, $p_{tr}(y = 2) = 1 - \delta$. We use values $\delta \in \{0.05, 0.1, 0.15, 0.2, 0.25, 0.3, 0.35, 0.4, 0.45, 0.5\}$ to simulate different relations between the marginals of training and testing labels. We assume that the true marginal probabilities $p_{te}(y)$ and $p_{tr}(y)$ are known, and compute weights as $\beta(y) = p_{te}(y)/p_{tr}(y)$ for the reweighted approach, and as in (27) for the proposed double-weighting approach. In addition, for each type of label shift (value of $\delta$), we carried out 200 random repetitions with 100 training samples and 100 testing samples.

**Results.** Figure 3 shows box-plots corresponding to the classification error of existing and proposed approaches. The results in the figure show how methods that do not take into account label shifts obtain poor performances. In addition, the figure show that reweighted methods obtain poor performances in the cases where $\delta$ takes small values ($p_{tr}(y)$ is small) because the ratios $\beta(y) = p_{te}(y)/p_{tr}(y)$ become large, as discussed in Section 3.1. The proposed methods can adapt to label shifts in situations where the shift between training and testing distribution challenges reweighted approaches. In particular, the double-weighting approach allows us to consider smaller weights $\beta(y)$ in those situations by reducing the corresponding weight $\alpha(y)$.

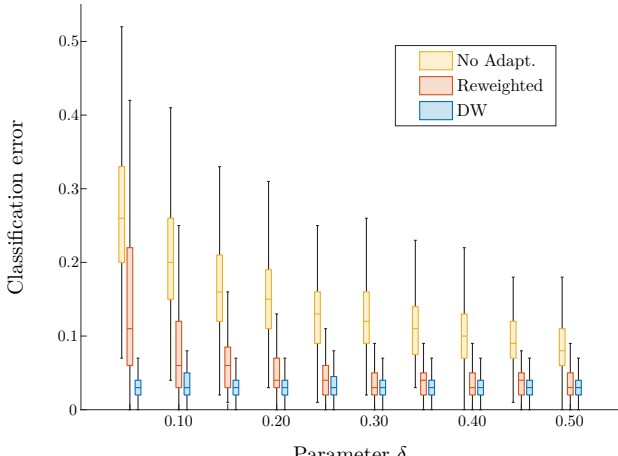

Figure 3: Classification error for the synthetic experiment using different approaches. The boxplot shows the different classification error with different proportions of the labels ratios for the training sets.

### 6.1.2 Experiments with real datasets

In the second set of experiments, we assess the performance of the proposed methods in comparison with existing techniques using real datasets publicly available in the UCI repository Dua & Graff (2017). We generate label shifts in the datasets following (Lipton et al., 2018). In the "tweak-one" shift, the training distribution is uniform over the set of possible labels, $p_{tr}(y) = 1/|\mathcal{Y}|$, while in the testing distribution, we assign probability $p_{te}(y) = \delta$ to half of the classes (rounded up). We set $\delta = 0.05$ for 10 repetitions and $\delta = 0.10$ for another 10 repetitions. In the "knock-out" shift, the testing distribution is uniform over the set of possible labels, while in the training distribution, we remove a proportion $\delta$ of the samples from the selected classes. We set $\delta = 0.9$ and select half of the classes (rounded up) for all 20 repetitions. In the "Dirichlet" shift, the training distribution is uniform over the set of possible labels, while in the testing distribution follow a Dirichlet distribution with parameter $\gamma = 0.1$. Note that we conducted experiments using Dirichlet shift exclusively on multiclass datasets. For the experiments using "redwine", we considered only three of the labels (2, 3, 4), discarding those with a small number of samples (0, 1, 5).

**Results.** Table 1 shows the averaged classification error across various datasets and scenarios of single-source label shift, along with their respective standard deviations. The second column of Table 1 describes the different types of label shifts described above.

Overall, the experimental results show how the proposed methods provide improved adaptation to single-source label shift in binary and multiclass classification setting in situations where the distribution of training is uniform while that of testing is not and vice versa. The label shift adaptation improvement for the proposed DW methods compared to reweighted methods is also observed when the training and testing distributions are assumed to be known. Note that for "knock-out" shift, the ratios $p_{te}(y)/p_{tr}(y)$ become large since $p_{tr}(y)$ is small for certain labels $y \in \mathcal{Y}$. In these cases, reweighted methods achieve poor performance compared with the double-weighting approach that can reduce the large values $\beta(y)$ by reducing the corresponding value $\alpha(y)$. The results agree with the theoretical results in Corollary 3.3 that show how the proposed methodology results in increased effective sample size in comparison with the reweighted approach.

### 6.2 Experiments for multi-source covariate shift adaptation

This subsection shows experimental results for the proposed approach for multi-source covariate shift adaptation in comparison with the state-of-the-art methods.

**Baseline methods.** Three single-source baselines are employed: logistic regression (LR), which ignore multi-source covariate shift, KMM (Huang et al., 2006; Gretton et al., 2008) and double-weighting covariate shift (DW-GCS) (Segovia-Martín et al., 2023), where we compute a classification rule from each of the training sources and evaluate the classification rule that achieves the smallest classification error on the testing

Table 1: Classification errors in 16 scenarios show that the proposed methods can more adequately adapt to single-source label shift. The bold values represent the lowest classification error in each scenario.

| Datasets | No Adapt. | Exact prob. | | Estimated prob. | | | | |
|---|---|---|---|---|---|---|---|---|
| | | Reweighted | DW | TarS | BBSE | RLLS | MLLS | DW-LS |
| **Adult** | | | | | | | | |
| tweak-one | $.43 \pm .01$ | $.08 \pm .02$ | $\mathbf{.07 \pm .02}$ | $\mathbf{.07 \pm .02}$ | $.29 \pm .01$ | $.24 \pm .24$ | $\mathbf{.07 \pm .02}$ | $\mathbf{.07 \pm .02}$ |
| knock-out | $.38 \pm .02$ | $.27 \pm .03$ | $\mathbf{.26 \pm .02}$ | $.30 \pm .04$ | $.42 \pm .05$ | $.43 \pm .06$ | $.39 \pm .05$ | $\mathbf{.28 \pm .03}$ |
| **Diabetes** | | | | | | | | |
| tweak-one | $.45 \pm .02$ | $\mathbf{.08 \pm .03}$ | $.11 \pm .03$ | $.09 \pm .03$ | $.33 \pm .05$ | $.37 \pm .15$ | $\mathbf{.08 \pm .03}$ | $\mathbf{.08 \pm .03}$ |
| knock-out | $.40 \pm .01$ | $.32 \pm .03$ | $\mathbf{.29 \pm .02}$ | $.33 \pm .03$ | $.45 \pm .06$ | $.46 \pm .05$ | $.43 \pm .06$ | $\mathbf{.29 \pm .02}$ |
| **Mammo** | | | | | | | | |
| tweak-one | $.37 \pm .01$ | $\mathbf{.08 \pm .02}$ | $\mathbf{.08 \pm .02}$ | $.12 \pm .02$ | $.11 \pm .02$ | $.18 \pm .25$ | $\mathbf{.08 \pm .03}$ | $.11 \pm .01$ |
| knock-out | $.31 \pm .01$ | $.21 \pm .02$ | $\mathbf{.20 \pm .01}$ | $\mathbf{.21 \pm .02}$ | $.44 \pm .11$ | $.44 \pm .11$ | $.42 \pm .11$ | $.23 \pm .09$ |
| **Usenet2** | | | | | | | | |
| tweak-one | $.50 \pm .02$ | $.33 \pm .01$ | $\mathbf{.30 \pm .02}$ | $.34 \pm .01$ | $.29 \pm .01$ | $.32 \pm .11$ | $\mathbf{.20 \pm .01}$ | $.24 \pm .02$ |
| knock-out | $\mathbf{.38 \pm .03}$ | $.41 \pm .05$ | $\mathbf{.37 \pm .02}$ | $.39 \pm .03$ | $.39 \pm .03$ | $.39 \pm .03$ | $.45 \pm .06$ | $\mathbf{.38 \pm .02}$ |
| **Credit** | | | | | | | | |
| tweak-one | $.48 \pm .02$ | $.14 \pm .01$ | $\mathbf{.11 \pm .01}$ | $.19 \pm .01$ | $.32 \pm .05$ | $.29 \pm .17$ | $\mathbf{.10 \pm .01}$ | $.16 \pm .03$ |
| knock-out | $.27 \pm .03$ | $.23 \pm .06$ | $\mathbf{.21 \pm .05}$ | $.24 \pm .06$ | $.25 \pm .08$ | $.27 \pm .09$ | $.30 \pm .06$ | $\mathbf{.22 \pm .05}$ |
| **20news** | | | | | | | | |
| tweak-one | $.64 \pm .03$ | $.59 \pm .06$ | $\mathbf{.44 \pm .05}$ | $.58 \pm .08$ | $.59 \pm .07$ | $\mathbf{.57 \pm .07}$ | $.70 \pm .12$ | $.58 \pm .09$ |
| knock-out | $\mathbf{.64 \pm .02}$ | $.63 \pm .04$ | $\mathbf{.61 \pm .03}$ | $\mathbf{.64 \pm .04}$ | $\mathbf{.64 \pm .02}$ | $.65 \pm .03$ | $.73 \pm .05$ | $.66 \pm .03$ |
| dirichlet | $.66 \pm .04$ | $.65 \pm .05$ | $\mathbf{.61 \pm .02}$ | $.65 \pm .04$ | $.65 \pm .04$ | $.65 \pm .05$ | $.72 \pm .02$ | $\mathbf{.64 \pm .02}$ |
| **Redwine** | | | | | | | | |
| tweak-one | $.66 \pm .13$ | $.41 \pm .22$ | $\mathbf{.24 \pm .06}$ | $\mathbf{.48 \pm .22}$ | $.55 \pm .26$ | $.59 \pm .25$ | $.57 \pm .25$ | $.52 \pm .25$ |
| knock-out | $.60 \pm .03$ | $.51 \pm .04$ | $\mathbf{.49 \pm .03}$ | $\mathbf{.55 \pm .07}$ | $.58 \pm .07$ | $.58 \pm .07$ | $.59 \pm .08$ | $\mathbf{.55 \pm .09}$ |
| dirichlet | $.65 \pm .07$ | $.56 \pm .13$ | $\mathbf{.51 \pm .07}$ | $.58 \pm .12$ | $.58 \pm .12$ | $.58 \pm .12$ | $.63 \pm .07$ | $\mathbf{.57 \pm .12}$ |

domain. We also compare with the performance obtained by the 2-stage weighting for MS domain adaptation (2SW-MDA) method that extends the reweighted approach for multi-source covariate shift (Sun et al., 2011), and the MS distributionally robust learning (MS-DRL) method that considers the same combination of classifiers learned independently on each of the sources (Wang et al., 2023).

**Experimental details.** The methods KMM (Huang et al., 2006), DW-GCS (Segovia-Martín et al., 2023), 2SW-MDA (Sun et al., 2011), and MS-DRL (Wang et al., 2023) have been implemented as detailed in their references. In addition, we implement a practical version of the theoretical work proposed in (Mansour et al., 2008), called combination-weighted KMM (CW KMM), where the initial set of classifiers $\{h_s\}_{s \in [S]}$ are learned using KMM. Then, we combine the classifiers to obtain a new classification rule of the form $h(y|x) = \sum_{s=1}^{S} \gamma_s(x) h_s(y|x)$, with $\gamma_s(x) = p_s(x) / \sum_{i=1}^{S} p_i(x)$ for $s \in [S]$, where the training probabilities $p_s(x)$ are estimated using kernel density estimation (KDE) (Sugiyama & Kawanabe, 2012). For the existing methods, we consider the default hyperparameter values provided by the authors. For the proposed method, the corresponding hyperparameters are obtained following the approach described in (Segovia-Martín et al., 2023), as detailed in Appendix F.

### 6.2.1 Experiment with synthetic data

In the experiments using synthetic data, we show how the proposed approach can achieve multi-source covariate shift adaptation in a scenario where the existing approaches are challenged. For such results, the training and testing data are drawn from Gaussian distributions

$$p_1(x) = N\left(\mathbf{x}; [-0.5, 0]^{\mathrm{T}}, 0.5^2 \mathbf{I}\right) \quad p_2(x) = N\left(\mathbf{x}; [0.5, 0]^{\mathrm{T}}, 0.5^2 \mathbf{I}\right) \quad p_{\text{te}}(x) = N\left(\mathbf{x}; [0, 0]^{\mathrm{T}}, 0.5^2 \mathbf{I}\right) \quad (61)$$

and labels are

$$y = \begin{cases} \text{sign}(x^{(2)}) & \text{if } x^{(1)} \geq 0 \\ -\text{sign}(x^{(2)}) & \text{if } x^{(1)} < 0. \end{cases} \quad (62)$$

Notice that this synthetic data can be easily classified switching between classifiers $\mathrm{h}_{x^{(1)} \geq 0}$ and $\mathrm{h}_{x^{(1)} < 0}$ depending on the sign of $x^{(1)}$. However the synthetic data is quite difficult to classify without knowing the change of sign corresponding to $x^{(1)}$. We carried out 100 random repetitions with 200 samples from each source and 200 testing samples and considered linear feature mapping. In addition, as a benchmark, we implemented an LR referred to as *LR ideal* that classifies using $\mathrm{h}_{x^{(1)} \geq 0}$ when $x^{(1)} \geq 0$, and using $\mathrm{h}_{x^{(1)} < 0}$ when $x^{(1)} < 0$, i.e., has access to the optimal assignment of the testing instances to their respective classifier.

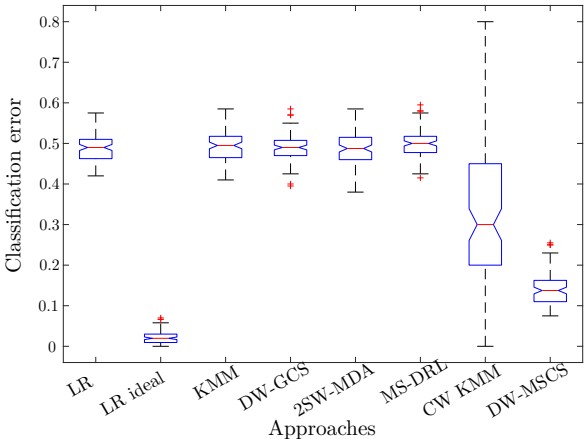

Figure 4: Classification error for the synthetic experiment using different approaches. The boxplot shows how considering classification rules that involve instance-dependent weighted combinations of feature mappings is needed (DW-MSCS). Note that *LR ideal* is a lower bound as it is the ideal case when we know the optimal assignment of testing instances to their respective classifier.

**Results.** Figure 4 shows box-plots corresponding to the classification error of existing and proposed approaches. Existing approaches obtain initial classifiers learned from each source and then combine them regardless of the testing instance $x$ we want to classify, i.e., consider final classification rules of the form $\mathrm{h} = \gamma_1 \mathrm{h}_1 + \gamma_2 \mathrm{h}_2$, with $\gamma_1, \gamma_2 \geq 0$, $\gamma_1 + \gamma_2 = 1$ The figure shows how those approaches obtain poor performances since the assignment of the testing instances to their respective classifier is not properly conducted. The proposed methods can adapt to multi-source covariate shift considering classification rules that involve instance-dependent weighted combinations of feature mappings, i.e., the final classification rules are given in terms of combinations of $\gamma_1(x)$ and $\gamma_2(x)$, with $\gamma_1(x), \gamma_2(x) \geq 0$, $\gamma_1(x) + \gamma_2(x) = 1$ for all $x \in \mathcal{X}$. This way, the classifiers learned assign labels to each testing instance $x$ using the sources that are more likely to classify that instance.

### 6.2.2 Experiments with real datasets

In the second set of experiments, we assess the performance of the proposed methods in comparison with existing techniques using real datasets. We consider Spam Detection, 20 Newsgroups, and Sentiment classification datasets. We also consider two common domain adaptation image datasets, Office-31 and DomainNet. For the experiments using "Spam detection" dataset, we consider a binary classification problem, where we select the labeled email set as the testing set, and the three annotated email sets as the training sources similar to done in (Duan et al., 2012). For the experiments using "20 Newsgroups" dataset, we consider 6 binary problems and 3 multiclass problems. Each of the training sources is generated by selecting the samples from two subcategories selected randomly for each binary task, and the testing set by selecting the samples from all the subcategories. For the experiments using "Sentiment" dataset, we consider 5 binary problems. Each of the training sources is generated by selecting the samples from each of the domains, and the testing set by selecting the samples from all domains. For the experiments using "DomainNet" dataset, we considered 6 multiclass problems. Each of the training sources is generated by selecting the samples from each of the domains, removing one domain (the one corresponding to the experiment's name), and the testing set is obtained by combining the samples from all the domains. For the experiments using "Office-31" dataset, we consider 4 multiclass problems, selecting labels depending of the type of the object. Each of the

two training sources are generated by selecting the samples from domains 'amazon' and 'webcam', and the testing set by combining the samples from all the three domains.

Table 2: Classification errors in 31 scenarios show that the proposed methods can more adequately adapt to multi-source covariate shift. The bold values represent the lowest classification error in each scenario.

| Datasets | LR | KMM | DW-GCS | 2SW-MDA | MS-DRL | CW KMM | DW-MSCS |
|---|---|---|---|---|---|---|---|
| **Spam detection** | | | | | | | |
| 500 features | $.45 \pm .07$ | $.45 \pm .07$ | $\mathbf{.39 \pm .04}$ | $.43 \pm .07$ | $.50 \pm .05$ | $.47 \pm .05$ | $.40 \pm .04$ |
| 1000 features | $.45 \pm .07$ | $.44 \pm .09$ | $.41 \pm .03$ | $.45 \pm .07$ | $.49 \pm .09$ | $.46 \pm .05$ | $\mathbf{.38 \pm .03}$ |
| 1500 features | $.46 \pm .07$ | $.46 \pm .07$ | $.40 \pm .04$ | $.46 \pm .07$ | $.46 \pm .06$ | $.45 \pm .06$ | $\mathbf{.36 \pm .04}$ |
| 2000 features | $.47 \pm .08$ | $.45 \pm .07$ | $.39 \pm .05$ | $.46 \pm .06$ | $.49 \pm .07$ | $.46 \pm .05$ | $\mathbf{.35 \pm .04}$ |
| 2500 features | $.48 \pm .07$ | $.47 \pm .05$ | $.38 \pm .04$ | $.47 \pm .06$ | $.49 \pm .07$ | $.47 \pm .04$ | $\mathbf{.35 \pm .03}$ |
| 3000 features | $.43 \pm .07$ | $.45 \pm .07$ | $.38 \pm .04$ | $.45 \pm .05$ | $.50 \pm .08$ | $.47 \pm .04$ | $\mathbf{.34 \pm .03}$ |
| **20 Newsgroups** | | | | | | | |
| comp vs rec | $.46 \pm .04$ | $.45 \pm .04$ | $.36 \pm .03$ | $.42 \pm .04$ | $.45 \pm .05$ | $.41 \pm .11$ | $\mathbf{.32 \pm .04}$ |
| comp vs sci | $.46 \pm .04$ | $.45 \pm .04$ | $.39 \pm .04$ | $.43 \pm .04$ | $.45 \pm .04$ | $.43 \pm .11$ | $\mathbf{.34 \pm .05}$ |
| comp vs talk | $.36 \pm .06$ | $.34 \pm .05$ | $.27 \pm .04$ | $.44 \pm .03$ | $.36 \pm .05$ | $.32 \pm .08$ | $\mathbf{.22 \pm .03}$ |
| rec vs sci | $.46 \pm .05$ | $.47 \pm .04$ | $.39 \pm .04$ | $.46 \pm .04$ | $.47 \pm .04$ | $.46 \pm .03$ | $\mathbf{.36 \pm .04}$ |
| rec vs talk | $.40 \pm .05$ | $.38 \pm .03$ | $.35 \pm .04$ | $.46 \pm .04$ | $.39 \pm .05$ | $.35 \pm .09$ | $\mathbf{.29 \pm .04}$ |
| sci vs talk | $.43 \pm .04$ | $.42 \pm .04$ | $.36 \pm .05$ | $.45 \pm .06$ | $.39 \pm .05$ | $.35 \pm .09$ | $\mathbf{.32 \pm .03}$ |
| comp vs rec vs sci | $.57 \pm .05$ | $.57 \pm .06$ | $.51 \pm .06$ | $.59 \pm .04$ | $.57 \pm .04$ | $.54 \pm .05$ | $\mathbf{.45 \pm .05}$ |
| comp vs rec vs talk | $.47 \pm .05$ | $.47 \pm .05$ | $.43 \pm .05$ | $.49 \pm .06$ | $.51 \pm .05$ | $.45 \pm .03$ | $\mathbf{.35 \pm .03}$ |
| comp vs sci vs talk | $.50 \pm .05$ | $.51 \pm .05$ | $.44 \pm .04$ | $.52 \pm .05$ | $.57 \pm .05$ | $.49 \pm .04$ | $\mathbf{.41 \pm .04}$ |
| rec vs sci vs talk | $.53 \pm .06$ | $.52 \pm .05$ | $.48 \pm .05$ | $.56 \pm .05$ | $.56 \pm .04$ | $.49 \pm .05$ | $\mathbf{.41 \pm .05}$ |
| **Sentiment** | | | | | | | |
| All Domains | $.48 \pm .06$ | $.43 \pm .04$ | $.32 \pm .01$ | $.34 \pm .02$ | $.49 \pm .04$ | $.46 \pm .03$ | $\mathbf{.23 \pm .01}$ |
| books | $.47 \pm .04$ | $.48 \pm .04$ | $.32 \pm .01$ | $.50 \pm .02$ | $.50 \pm .03$ | $.48 \pm .03$ | $\mathbf{.25 \pm .02}$ |
| dvd | $.48 \pm .05$ | $.44 \pm .04$ | $.32 \pm .01$ | $.48 \pm .01$ | $.50 \pm .02$ | $.48 \pm .02$ | $\mathbf{.24 \pm .01}$ |
| electronics | $.46 \pm .06$ | $.45 \pm .04$ | $.32 \pm .01$ | $.49 \pm .03$ | $.49 \pm .03$ | $.47 \pm .03$ | $\mathbf{.25 \pm .01}$ |
| kitchen | $.46 \pm .05$ | $.45 \pm .04$ | $.33 \pm .03$ | $.48 \pm .02$ | $.49 \pm .02$ | $.47 \pm .03$ | $\mathbf{.25 \pm .02}$ |
| **DomainNet** | | | | | | | |
| clipart | $.41 \pm .05$ | $.44 \pm .05$ | $.46 \pm .03$ | $.42 \pm .06$ | $.51 \pm .03$ | $.33 \pm .03$ | $\mathbf{.31 \pm .03}$ |
| infograph | $.38 \pm .04$ | $.43 \pm .05$ | $.45 \pm .03$ | $.44 \pm .05$ | $.55 \pm .05$ | $.35 \pm .04$ | $\mathbf{.30 \pm .03}$ |
| painting | $.38 \pm .04$ | $.44 \pm .06$ | $.46 \pm .02$ | $.43 \pm .05$ | $.55 \pm .05$ | $.35 \pm .05$ | $\mathbf{.32 \pm .02}$ |
| quickdraw | $.42 \pm .04$ | $.45 \pm .04$ | $.46 \pm .04$ | $.44 \pm .03$ | $.55 \pm .04$ | $.34 \pm .04$ | $\mathbf{.31 \pm .03}$ |
| real | $.40 \pm .05$ | $.45 \pm .05$ | $.47 \pm .02$ | $.43 \pm .04$ | $.54 \pm .04$ | $.35 \pm .03$ | $\mathbf{.31 \pm .03}$ |
| sketch | $.40 \pm .04$ | $.45 \pm .05$ | $.47 \pm .04$ | $.44 \pm .04$ | $.55 \pm .04$ | $\mathbf{.33 \pm .04}$ | $\mathbf{.33 \pm .03}$ |
| **Office-31** | | | | | | | |
| electronics | $.12 \pm .04$ | $.19 \pm .05$ | $.23 \pm .03$ | $\mathbf{.09 \pm .03}$ | $.39 \pm .03$ | $.14 \pm .03$ | $.11 \pm .02$ |
| stationery | $.14 \pm .03$ | $.16 \pm .05$ | $.18 \pm .04$ | $.11 \pm .03$ | $.38 \pm .07$ | $\mathbf{.10 \pm .03}$ | $\mathbf{.10 \pm .03}$ |
| organization | $.13 \pm .03$ | $.17 \pm .04$ | $.21 \pm .05$ | $.13 \pm .04$ | $.39 \pm .05$ | $.13 \pm .03$ | $\mathbf{.12 \pm .03}$ |
| mixed | $.09 \pm .03$ | $.14 \pm .05$ | $.20 \pm .03$ | $\mathbf{.08 \pm .03}$ | $.31 \pm .04$ | $.12 \pm .04$ | $.10 \pm .02$ |

**Results.** Table 2 shows the averaged classification error across various datasets and scenarios of multi-source covariate shift, along with their respective standard deviations. The first column of Table 2 describes the different classification tasks. The experimental results show that the proposed methods provide improved adaptation to multi-source covariate shift in binary and multiclass classification settings. Since the testing set contains samples from an unseen domain for the training sources, in the experiments the support of the testing distribution is larger than the support of the training distribution. Therefore, methods like 2SW-MDA, MS-DRL, and CW KMM, that obtain weights using reweighted techniques, result in poor performance even when compared to LR methods. This is clearly observed in the experiments using the Sentiment dataset. In such dataset, method 2SW-MDA results in a negative transfer in all cases except for the "All domains", where all four domains are used as training sources. In the other cases, one domain is removed (the domain corresponding to the experiment's name). In these experiments, it is clear that considering double-weighting and classification rules that involve instance-dependent weighted combinations of feature mappings substantially improves the model performance.

## 6.3 Experiments for multi-source label shift adaptation

This subsection shows experimental results for the proposed approach for multi-source label shift adaptation in comparison with the state-of-the-art approaches.

Table 3: Classification errors in 20 scenarios using "sentiment analysis" dataset show that the proposed methods can more adequately adapt to multi-source label shift. The bold values represent the lowest classification error in each scenario.

| Datasets | Dirichlet $\gamma$ | LR | KMM | DW-LS | LWC KMM | DW-MSLS |
|---|---|---|---|---|---|---|
| **books** | $\gamma = 0.01$ | $.43 \pm .03$ | $.43 \pm .02$ | $.44 \pm .02$ | $.41 \pm .02$ | $\mathbf{.36 \pm .06}$ |
| | $\gamma = 0.1$ | $.42 \pm .02$ | $.43 \pm .02$ | $.41 \pm .06$ | $.41 \pm .03$ | $\mathbf{.38 \pm .08}$ |
| | $\gamma = 1$ | $.42 \pm .02$ | $.42 \pm .02$ | $.33 \pm .03$ | $.41 \pm .03$ | $\mathbf{.28 \pm .03}$ |
| | $\gamma = 10$ | $.42 \pm .02$ | $.42 \pm .02$ | $.32 \pm .02$ | $.40 \pm .02$ | $\mathbf{.26 \pm .02}$ |
| | $\gamma = 100$ | $.42 \pm .03$ | $.42 \pm .02$ | $.30 \pm .02$ | $.39 \pm .02$ | $\mathbf{.26 \pm .02}$ |
| **dvd** | $\gamma = 0.01$ | $.43 \pm .02$ | $.43 \pm .02$ | $.43 \pm .03$ | $\mathbf{.42 \pm .02}$ | $.43 \pm .07$ |
| | $\gamma = 0.1$ | $.43 \pm .02$ | $.43 \pm .02$ | $.40 \pm .05$ | $.42 \pm .03$ | $\mathbf{.37 \pm .08}$ |
| | $\gamma = 1$ | $.42 \pm .02$ | $.43 \pm .02$ | $.34 \pm .04$ | $.41 \pm .02$ | $\mathbf{.27 \pm .03}$ |
| | $\gamma = 10$ | $.42 \pm .02$ | $.42 \pm .02$ | $.31 \pm .02$ | $.40 \pm .02$ | $\mathbf{.26 \pm .03}$ |
| | $\gamma = 100$ | $.43 \pm .03$ | $.43 \pm .02$ | $.30 \pm .02$ | $.41 \pm .02$ | $\mathbf{.26 \pm .02}$ |
| **electronics** | $\gamma = 0.01$ | $.43 \pm .02$ | $.43 \pm .02$ | $.45 \pm .02$ | $.42 \pm .03$ | $\mathbf{.40 \pm .06}$ |
| | $\gamma = 0.1$ | $.43 \pm .03$ | $.43 \pm .02$ | $\mathbf{.40 \pm .05}$ | $\mathbf{.40 \pm .02}$ | $.40 \pm .08$ |
| | $\gamma = 1$ | $.43 \pm .02$ | $.43 \pm .02$ | $.35 \pm .03$ | $.40 \pm .03$ | $\mathbf{.32 \pm .05}$ |
| | $\gamma = 10$ | $.43 \pm .02$ | $.43 \pm .02$ | $.30 \pm .02$ | $.40 \pm .02$ | $\mathbf{.28 \pm .05}$ |
| | $\gamma = 100$ | $.42 \pm .03$ | $.42 \pm .02$ | $.30 \pm .02$ | $.40 \pm .02$ | $\mathbf{.26 \pm .03}$ |
| **kitchen** | $\gamma = 0.01$ | $.43 \pm .02$ | $.43 \pm .02$ | $.45 \pm .04$ | $\mathbf{.41 \pm .02}$ | $.44 \pm .07$ |
| | $\gamma = 0.1$ | $.43 \pm .02$ | $\mathbf{.42 \pm .02}$ | $.43 \pm .05$ | $\mathbf{.42 \pm .02}$ | $.45 \pm .06$ |
| | $\gamma = 1$ | $.42 \pm .02$ | $.42 \pm .02$ | $.36 \pm .06$ | $.41 \pm .02$ | $\mathbf{.31 \pm .06}$ |
| | $\gamma = 10$ | $.42 \pm .02$ | $.42 \pm .02$ | $.31 \pm .03$ | $.40 \pm .02$ | $\mathbf{.26 \pm .02}$ |
| | $\gamma = 100$ | $.42 \pm .03$ | $.42 \pm .02$ | $.30 \pm .03$ | $.40 \pm .02$ | $\mathbf{.26 \pm .03}$ |

**Baseline methods.** We employ three single-source baselines: LR, which ignore multi-source label shift, KMM (Zhang et al., 2013) and DW-LS. For the baseline methods, we compute a classification rule from each of the training sources and evaluate the classification rule that achieves the smallest classification error on the testing domain. We also compare with the performance obtained by a reweighted method for multi-source label shift label-dependent weighted combination KMM (LWC KMM) (Zhang et al., 2015), where the initial set of classifiers $\{h_s\}_{s \in [S]}$ are learned using KMM. Then, we combine the classifiers to obtain a new classification rule of the form $h(y|x) = \sum_{s=1}^{S} \gamma_s(y) h_s(y|x)$, with $\gamma_s(y) = p_s(y) / \sum_{i=1}^{S} p_i(y)$ for $s \in [S]$, where the training probabilities $p_s(y)$ are estimated as the ratio of samples with label $y$ from source $s$, for $y \in \mathcal{Y}$, $s \in [S]$. The methods KMM (Zhang et al., 2013), and LWC KMM have been implemented as detailed in their references.

**Experimental details.** We assess the performance of the proposed methods in comparison with existing techniques using "sentiment analysis" datasets. We consider 4 binary problems, where each of the training sources is generated by selecting the samples from each of the domains, and the testing set by selecting the samples from all the domains. We generate label shifts in each training source using Dirichlet distributions with concentration parameter $\gamma$ for all classes, while the testing distribution is uniform over the set of labels. We use values $\gamma \in \{0.01, 0.1, 1, 10, 100\}$ to simulate different label shifts in each training source, so that smaller $\gamma$ corresponds to bigger shifts, similar to done in (Azizzadenesheli et al., 2019). For each type of multi-source label shift (value of $\gamma$) we carry out 20 random repetitions with 1000 training samples in each source and 150 testing samples.

**Results.** Table 3 shows the averaged classification error using "sentiment analysis" datasets across different types of multi-source label shifts, along with their respective standard deviations. The first column of the Table 3 describes the training source that we removed. The experimental results show how reweighted approaches struggle to effectively adapt to the label shift between the different sources, so that using a

single-source KMM achieve similar results as that for LWC KMM. In contrast, the proposed methods show significantly improved adaptation to multi-source label shift, leveraging the information between the different training sources.

## 7 Conclusion

In this paper, we presented a unified framework to address general distribution shifts, including label and multi-source shifts. The learning framework for distribution shift adaptation is based on a double-weighting approach, considering weight functions at training and testing that depend on the covariates and the labels. By assigning weights to both training and testing samples, our methods alleviate the problem of existing methods when the weights functions at training take large values. In the multi-source settings, the proposed methods leverage the rich complementary information among sources, considering classification rules that involve sample-dependent weighted combinations of feature mappings. In addition, we present generalization bounds and empirical results for the proposed methods that show an improved adaptation to single-source label and multi-source shifts. The proposed unifying view of double-weighting proposed can enable techniques capable to adapt to a more general range of scenarios affected by distribution shift.

## Acknowledgments

Funding in direct support of this work has been provided by projects PID2022-137063NBI00, CNS2022-135203, and CEX2021-001142-S funded by MCIN/AEI/10.13039/501100011033 and the European Union "NextGenerationEU"/PRTR, BCAM Severo Ochoa accreditation CEX2021-001142-S/MICIN/AEI/10.13039/501100011033 funded by the Spanish Ministry of Science and Innovation, the Amazon Research Award, the Discovery Award of the Johns Hopkins University, and a seed grant from the JHU Institute of Assured Autonomy.

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

# A   Prior Work

In this section, we describe the main existing approaches in the literature that deal with single-source and multi-source covariate and label shift.

## A.1   Covariate shift

**Reweighted approach.** Most of the techniques for covariate shift adaptation are based on the reweighted approach (Sugiyama & Kawanabe, 2012; Shimodaira, 2000; Zadrozny, 2004; Cortes et al., 2008; Dudík et al., 2005; Lin et al., 2002). These methods weight loss functions at training by means of the function $\beta(x) = \mathrm{p_{te}}(x)/\mathrm{p_{tr}}(x)$. Using these weights, reweighted methods can account for the fact that some training instances are unlikely at testing, and assign low relevance to such instances at training. Reweighted methods assume the support of $\mathrm{p_{tr}}$ contains that of $\mathrm{p_{te}}$ (i.e., $\mathrm{p_{te}}(x) > 0 \Rightarrow \mathrm{p_{tr}}(x) > 0$). Even if this condition is satisfied, such methods may achieve poor performances if the ratio $\mathrm{p_{te}}(x)/\mathrm{p_{tr}}(x)$ takes large values at certain training samples. In these cases, the learning process is dominated by few training samples (Cortes & Mohri, 2014).

**Robust approach.** Robust methods for covariate shift adaptation (Liu & Ziebart, 2014; 2017; Chen et al., 2016) are derived from a distributionally robust learning framework. These methods weight functions at testing using the weight function $\alpha(x) = \mathrm{p_{tr}}(x)/\mathrm{p_{te}}(x)$. Using these weights, such methods obtain classification rules that produce less confident predictions for testing instances are unlikely at training. Reweighted methods assume the support of $\mathrm{p_{te}}$ contains that of $\mathrm{p_{tr}}$ (i.e., $\mathrm{p_{tr}}(x) > 0 \Rightarrow \mathrm{p_{te}}(x) > 0$). Even if such condition is satisfied, robust methods may achieve poor performances if the ratio $\mathrm{p_{tr}}(x)/\mathrm{p_{te}}(x)$ take large values at certain testing samples, leading to overconfident classification rules.

## A.2   Label shift

The main existing techniques for label shift adaptation are based on reweighted approach for covariate shift extended to label shift. These methods exploit the fact that, for any function $f$, we have that

$$\mathbb{E}_{\mathrm{p_{te}}} f(x, y) = \mathbb{E}_{\mathrm{p_{tr}}} \beta(y) f(x, y), \text{ for } \beta(y) = \frac{\mathrm{p_{te}}(y)}{\mathrm{p_{tr}}(y)} \tag{63}$$

if $\mathrm{p}_{\mathrm{te}}(y) > 0 \Rightarrow \mathrm{p}_{\mathrm{tr}}(y) > 0$ for all $y \in \mathcal{Y}$.

Using (63), reweighted approaches weight loss functions at training using weight function $\beta(y)$ defined in (63). Using weights $\beta(y)$, reweighted methods assign lower relevance to the training samples whose labels are unlikely to be observed at testing. The following describes how the different approaches estimate the weight function $\beta(y)$.

**Black-box predictor approach.** Some existing techniques for label shift adaptation (Lipton et al., 2018) estimate both $\mathrm{p}_{\mathrm{te}}(y)$ and $\mathrm{p}_{\mathrm{tr}}(y)$ in order to compute weights in (63). $\mathrm{p}_{\mathrm{tr}}(y)$ can be easily estimated, and $\mathrm{p}_{\mathrm{te}}(y)$ is estimated using a confusion matrix. However, the inverse of the confusion matrix may be arbitrarily close to a singular matrix for a reduced number of training samples. The regularized approach (Azizzadenesheli et al., 2019) alleviates the problems regarding the confusion matrix by introducing a regularization term of the form $||\boldsymbol{\beta} - 1||$. The approach proposed in (Garg et al., 2020) applies a calibration heuristic to the classifier. In particular, they present a unified view of both (Lipton et al., 2018) and (Azizzadenesheli et al., 2019), encompassing the methods proposed in (Lipton et al., 2018) for a particular choice of the calibration method.

**Conditional KMM approach.** Certain label shift techniques (Zhang et al., 2013) compute the weight function $\beta(y)$ generalizing the conventional KMM method (Huang et al., 2006; Gretton et al., 2008) for label shift adaptation. In order to obtain the weight function $\beta(y)$, the method in (Zhang et al., 2013) considers kernel mean embedding of marginal and conditional distributions (Song et al., 2009).

Black-box predictor and conditional KMM correct the label shift performing a reweighted approach using the weights $\beta(y)$ estimated. These methods inherit the problems of standard reweighted methods for covariate shift adaptation. If the weights $\beta(y)$ take large values at certain samples with specific labels, the learning process is dominated by samples with those labels.

### A.3 Multi-source covariate shift

**Two-stage weighting approach.** The main existing techniques for multi-source covariate shift adaptation (Sun et al., 2011; Nomura & Saito, 2021) extend the existing reweighted approach for covariate shift to the multi-source case. Two-stage weighting methods weight training samples using the function $\beta_s(x) = \mathrm{p}_{\mathrm{te}}(x)/\mathrm{p}_s(x)$ for each source $s \in [S]$ to learn an initial set of classification rules. Then, they rescale functions $\beta_s(x)$ by a vector $\boldsymbol{\gamma}$ that multiplies all the weights for each source $s \in [S]$ by $\gamma_s$. Using the rescaled weights $\gamma_s \beta_s(x)$, such methods learn a new classification rule over all the training sources. In (Nomura & Saito, 2021), the vector $\boldsymbol{\gamma}$ is computed in order to reduce the variance of the estimators, while in (Sun et al., 2011), it is computed in order to minimize potential mismatches between conditional probability distributions. Two-stage weighting methods inherit the problems of single-source reweighted method. These problems are related to the assumption that the support of $\mathrm{p}_{\mathrm{tr}}$ contains that of $\mathrm{p}_{\mathrm{te}}$ (i.e., $\mathrm{p}_{\mathrm{te}}(x) > 0 \Rightarrow \mathrm{p}_{\mathrm{tr}}(x) > 0$). Even if this condition is satisfied, reweighted methods may exhibit poor performance if the ratio $\mathrm{p}_{\mathrm{te}}(x)/\mathrm{p}_{\mathrm{tr}}(x)$ takes large values at certain training samples. In these cases, the learning process is dominated by a few training samples (Cortes et al., 2008; Cortes & Mohri, 2014; Sugiyama & Kawanabe, 2012). In addition, two-stage weighting approaches only share information between sources to rescale all the weights for each source by the constant vector $\boldsymbol{\gamma}$.

**Approaches based on classification rules combinations.** Certain multi-source domain adaptation approaches (Yang et al., 2007; Schweikert et al., 2008; Tu & Sun, 2012; Sun & Shi, 2013; Xu et al., 2018; Peng et al., 2019; Wang et al., 2023) are based on combining an initial set of pre-learned classifiers $\{\mathrm{h}_s\}_{s \in [S]}$ from each of the sources $s \in [S]$ in a post hoc manner. For example, (Xu et al., 2018; Peng et al., 2019) obtain the pre-learned classifiers minimizing the discrepancy between first moments of the training and testing distributions. These approaches consider the same combination of information from sources for each testing sample to derive a new classification rule $\mathrm{h} = \sum_{s=1}^{S} \gamma_s \mathrm{h}_s$ with $\sum_{s=1}^{S} \gamma_s = 1$. However, the pre-learned classifiers associated with each training source do not take into account the other training sources. In addition, the theoretical work (Blitzer et al., 2007; Mansour et al., 2008; 2009) provide a general theory regarding how to use the samples from the different sources in order to minimize the expected loss, showing that a linear combination rule may perform poorly and using instance-dependent combinations of information from sources can result in improved performance.

### A.4 Multi-source label shift

**Conditional KMM approach.** The techniques presented in (Zhang et al., 2015) extend the KMM approach for label shift correction proposed in (Zhang et al., 2013). This approach considers a set of classifiers learned on each source. These classification rules are combined considering the theoretical work of (Mansour et al., 2008). However, similar to the reweighted methods, KMM methods also inherit the high variance problems of single-source reweighted methods when the weights obtained take large values at certain training samples. Moreover, these methods only share information among sources to rescale all the weights for each source by a constant that represents the overall relevance of such source.

## B Contributions for multi-source distribution shifts

The main contributions of the proposed methodology for multi-source distribution shifts are as follows.

The methods in the paper result in improved classification rules that involve sample-dependent combinations of the most relevant feature mappings. Applying directly the double-weighting approach proposed in (Segovia-Martín et al., 2023) would result in obtaining a set of $S$ classification rules $\{h_s(y|x)\}_{s \in [S]}$ that we would need to combine in order to obtain a classification for the testing instances. However, there is not a direct way to combine the initial set of classification rules. On the other hand, the proposed approach obtains a single classification rule defined in terms of combinations of the most relevant feature mappings. This combination is given by the set of weights $\{\alpha_s(x, y)\}_{s \in [S]}$ and depends on the testing instance we want to classify. In addition, our methodology differs from the state-of-the-art since most of the existing methods obtain a set of classification rules and combine them in a post-hoc manner.

The proposed uncertainty set $\mathcal{U}$ is formed by the $S$ sets of constraints, each of which characterizing weighted feature expectation matching on $\Phi_{\alpha_s}(x, y)$ in each source domain with weights $\alpha_s(x)$. This set is contained in each of the $S$ uncertainty sets $\{\mathcal{U}_s\}_{s=1}^S$ and often significantly smaller. Since $\mathcal{U} \subseteq \mathcal{U}_s$,

$$\min_h \max_{p \in \mathcal{U}} \ell(h, p) \le \min_h \max_{p \in \mathcal{U}_s} \ell(h, p), \tag{64}$$

and $R(\mathcal{U}) \le R(\mathcal{U}_s)$ for $s \in [S]$. This means that the minimax risk of the proposed multi-source method is smaller than the minimax risk of the single-source method. This also differentiates us from directly applying (Segovia-Martín et al., 2023) to each source, since it will not lead to our minimax risk. The classification rule obtained from our method contains instance-dependent combinations of the different feature mappings $\{\Phi_{\alpha_s}(x, y)\}_{s=1}^S$, where the set of weights $\{\alpha_s(x, y)\}_{s=1}^S$ may be interpreted as the ratio of information learned from each of the sources to classify the instance $x$. Besides a smaller minimax risk, the resulting classification rule also has the advantage that we only need one $\alpha_s(x, y)$ from the set $\{\alpha_s(x, y)\}_{s=1}^S$ to be sufficiently large in order to have classification rules with high confidence. By allowing the sources to share information between them, we are able to alleviate the trade-off in (Segovia-Martín et al., 2023), where weights $\alpha_s(x, y)$, for $s \in [S]$, $x \in \mathcal{X}$, need to be sufficiently large in order to have classification rules with high confidence. Then, we can consider more flexible values of $\beta_s(x, y)$, for $s \in [S]$, $x \in \mathcal{X}$, leading to better expectation estimates and better bounds than just applying (Segovia-Martín et al., 2023), where weights are independent for each of the sources.

The techniques presented achieve an improved trade-off between the effective sample size and the confidence of the classification rules. Using the set of weights $\{\beta_s(x, y)\}_{s \in [S]}$, we improve the effective sample size, obtaining better generalization bounds in comparison with existing reweighted techniques and single-source double-weighting. Using the set of weights $\{\alpha_s(x, y)\}_{s \in [S]}$, we obtain classification rules that achieve significant confidence in a larger region in comparison with the double-weighting method in (Segovia-Martín et al., 2023). By choosing carefully the reference weights $\{\alpha_s(x, y)\}_{s \in [S]}$ and $\{\beta_s(x, y)\}_{s \in [S]}$, we can improve the existing trade-off in the double-weighting methodology. In particular, we defined weights $\alpha(x, y)$ in (31) taking into account that we do not need significant confidence $\alpha_s(x, y) \approx 1$ for each source $s \in [S]$ but rather that there is enough confidence among all sources $\sum_{s=1}^S \alpha_s(x, y) \approx 1$. This way, the region where the confidence of the classification rule is large is bigger than using the weights in (Segovia-Martín et al., 2023). The reference weights $\beta_s(x)$ proposed in (31) take into account all the sources, and they are smaller than those in (Segovia-Martín et al., 2023), leading to reduced error due to finite sample sizes.

The generalization bounds presented in Section 4.3 for multi-source distribution shift are non-trivial and different from the ones in Section 3.3 for single-source distribution shift. Specifically, our generalization risk under the true probability distribution $R(\mathrm{h}^{\mathcal{U}})$ is bounded by the minimax risk $R(\mathcal{U})$:

$$R\left(\mathrm{h}^{\mathcal{U}}\right) \leq R(\mathcal{U}) + \sum_{s=1}^{S} \left(|\boldsymbol{\tau}_s - \mathbb{E}_{\mathrm{p}_{\mathrm{te}}} \Phi_{\alpha_s}(x,y)| - \boldsymbol{\lambda}_s\right)^T |\boldsymbol{\mu}_s^*|. \tag{65}$$

On the right-hand-side, we have $R(\mathcal{U}) \leq R(\mathcal{U}_s) \ \forall s \in [S]$, and, since, with probability $1 - \delta$

$$\|\boldsymbol{\tau}_s - \mathbb{E}_{\mathrm{p}_{\mathrm{te}}} \Phi_{\alpha_s}(x,y)\|_{\infty} \leq M \sqrt{\frac{2\|\beta_s(x)\|_{\infty}^2}{n} \log \frac{2}{\delta}} \tag{66}$$

where $M$ is a constant satisfying that $\|\Phi(x,y)\|_{\infty} \leq M$ for all $x \in \mathcal{X}$, $y \in \mathcal{Y}$, smaller weights $\beta_s(x)$ lead to better expectation estimates. Therefore, we obtain much better generalization bounds compared to applying (Segovia-Martín et al., 2023) multiple times. Alternatively, if we use (Segovia-Martín et al., 2023) to solve $S$ optimization problems independently and then take the sum, the resulting generalization bounds are strictly worse. Moreover, as shown in the paper (Mansour et al., 2008), considering rules of the form $\mathrm{h}(y|x) = \sum_{s=1}^{S} \mathrm{h}_s(y|x)$ can lead to poor performance.

The methods introduced effectively leverage information from multiple sources. The methodology presented in Section 4 is specific for multi-source distribution shift. This is not a straightforward extension of (Segovia-Martín et al., 2023), since the proposed methods efficiently leverage information from multiple sources by considering a set of weights $\{\alpha_s(x,y)\}_{s\in[S]}$ and $\{\beta_s(x,y)\}_{s\in[S]}$ that depend on all the training sources, allowing us to obtain improved generalization bounds and allowing us to adapt to more general distribution shift scenarios where state-of-the-art methods are challenged. Finally, the weights considered take into account the number of examples of each source, and may penalize more those where the expectation is worse estimated due to a lower number of training samples available, as we show in Section D.

## C   Proofs for Section 3 and Section 4

In the following, we prove Theorems 4.1 and 4.2, while the proofs of Theorems 3.1 and 3.2 are analogous by setting $S = 1$. The proofs of Theorems 4.1 and 4.2 below are done for the case of finite $\mathcal{X}$. The proofs for infinite $\mathcal{X}$ can be carried out analogously using Fenchel duality instead of Lagrange duality, similarly to as is done in Altun & Smola (2006); Mazuelas et al. (2023).

***Proof of Theorem 4.1.*** Firstly, for each $\mathrm{h} \in \mathrm{T}(\mathcal{X}, \mathcal{Y})$, we have that

$$\begin{aligned} \max_{\mathrm{p}\in\mathcal{U}} \ell(\mathrm{h},\mathrm{p}) = \quad & \max_{\mathbf{p}} \quad \mathbf{l}^{\mathrm{T}}\mathbf{p} - I_{+}(\mathbf{p}) \\ & \text{s.t.} \quad \sum_{y\in\mathcal{Y}} \mathrm{p}(x,y) = \mathrm{p}_{\mathrm{te}}(x), \ \forall x \in \mathcal{X} \\ & \quad \boldsymbol{\tau}_s - \boldsymbol{\lambda}_s \preceq \boldsymbol{\Phi}_{\alpha_s}^{\mathrm{T}}\mathbf{p} \preceq \boldsymbol{\tau}_s + \boldsymbol{\lambda}_s, \ \forall s \in [S] \end{aligned} \tag{67}$$

where $\mathbf{l}$, $\mathbf{p}$, and $\boldsymbol{\Phi}_{\alpha_s}$ denote the vectors and matrix with rows $\ell(\mathrm{h},(x,y))$, $\mathrm{p}(x,y)$, and $\alpha_s(x,y)\Phi(x,y)^{\mathrm{T}}$, respectively, for $x \in \mathcal{X}$, $y \in \mathcal{Y}$, and

$$I_{+}(\mathbf{p}) = \begin{cases} 0 & \text{if } \mathbf{p} \succeq \mathbf{0} \\ \infty & \text{otherwise.} \end{cases}$$

Optimization problem (67) has Lagrange dual

$$\begin{aligned} \min_{\{\boldsymbol{\mu}_{s,1}\}_{s\in[S]},\{\boldsymbol{\mu}_{s,2}\}_{s\in[S]},\nu(x)} \quad & -\sum_{s=1}^{S} \left(\boldsymbol{\tau}_s - \boldsymbol{\lambda}_s\right)^{\mathrm{T}} \boldsymbol{\mu}_{s,1} + \sum_{s=1}^{S} \left(\boldsymbol{\tau}_s + \boldsymbol{\lambda}_s\right)^{\mathrm{T}} \boldsymbol{\mu}_{s,2} \\ & + \mathbb{E}_{\mathrm{p}_{\mathrm{te}}(x)} \nu(x) + f^*\left(\sum_{s=1}^{S} \boldsymbol{\Phi}_{\alpha_s}(\boldsymbol{\mu}_{s,1} - \boldsymbol{\mu}_{s,2}) - \boldsymbol{\nu}\right) \\ \text{s.t.} \quad & \boldsymbol{\mu}_{s,1}, \boldsymbol{\mu}_{s,2} \succeq \mathbf{0}, \ \forall s \in [S] \end{aligned} \tag{68}$$

where $\boldsymbol{\nu}$ is the vector in $\mathbb{R}^{|\mathcal{X}||\mathcal{Y}|}$ with component corresponding with $(x, y)$ for $x \in \mathcal{X}$, $y \in \mathcal{Y}$ given by $\nu(x)$, and $f^*$ is the conjugate function of $f(\mathbf{p}) = -\mathbf{1}^{\mathrm{T}}\mathbf{p} + I_+(\mathbf{p})$ given by

$$f^*(\mathbf{w}) = \sup_{\mathbf{p} \succeq \mathbf{0}} \mathbf{w}^{\mathrm{T}}\mathbf{p} + \mathbf{1}^{\mathrm{T}}\mathbf{p} = \begin{cases} 0 & \text{if } \mathbf{w} \preceq -\mathbf{1} \\ \infty & \text{otherwise.} \end{cases}$$

Therefore, the Lagrange dual above becomes

$$\begin{aligned} \min_{\{\boldsymbol{\mu}_{s,1}\}_{s\in[S]}, \{\boldsymbol{\mu}_{s,2}\}_{s\in[S]}, \nu(x)} \quad & -\sum_{s=1}^{S}\left(\boldsymbol{\tau}_s - \boldsymbol{\lambda}_s\right)^{\mathrm{T}}\boldsymbol{\mu}_{s,1} + \sum_{s=1}^{S}\left(\boldsymbol{\tau}_s + \boldsymbol{\lambda}_s\right)^{\mathrm{T}}\boldsymbol{\mu}_{s,2} + \mathbb{E}_{\mathrm{P_{te}}(x)}\nu(x) \\ \text{s.t.} \quad & \{\boldsymbol{\mu}_{s,1}\}_{s=1}^{S}, \{\boldsymbol{\mu}_{s,2}\}_{s=1}^{S} \succeq \mathbf{0}, \; \forall s \in [S] \\ & \sum_{s=1}^{S} \Phi_{\alpha_s}(x,y)^{\mathrm{T}}(\boldsymbol{\mu}_{s,1} - \boldsymbol{\mu}_{s,2}) - \nu(x) \leq -\ell(\mathrm{h},(x,y)), \; \forall x \in \mathcal{X}, y \in \mathcal{Y}. \end{aligned} \tag{69}$$

It is easy to see that the solution of such optimization problem $\{\bar{\boldsymbol{\mu}}_{s,1}, \bar{\boldsymbol{\mu}}_{s,2}\}_{s\in[S]}$ satisfies that $\bar{\mu}_{s,1}^{(i)}\bar{\mu}_{s,2}^{(i)} = 0$ for $s \in [S]$ and any $i$ such that $\lambda_s^{(i)} > 0$. If there exist a set $\{i_s\}_{s=1}^{S}$ with $\lambda_s^{(i_s)} > 0$ and $\bar{\mu}_{s,1}^{(i_s)}\bar{\mu}_{s,2}^{(i_s)} \neq 0$ for $s \in [S]$, there would exist an $\epsilon > 0$ such that $\bar{\mu}_{s,1}^{(i_s)} > \epsilon$, $\bar{\mu}_{s,2}^{(i_s)} > \epsilon$. Then $\tilde{\mu}_{s,1}, \tilde{\mu}_{s,2}, \nu^*(x)$ with

$$\tilde{\mu}_{s,1}^{(i_s)} = \bar{\mu}_{s,1}^{(i_s)} - \epsilon, \; \tilde{\mu}_{s,1}^{(j_s)} = \bar{\mu}_{s,1}^{(j_s)} \text{ for } j_s \neq i_s$$
$$\tilde{\mu}_{s,2}^{(i_s)} = \bar{\mu}_{s,2}^{(i_s)} - \epsilon, \; \tilde{\mu}_{s,2}^{(j_s)} = \bar{\mu}_{s,2}^{(j_s)} \text{ for } j_s \neq i_s$$

for $s \in [S]$ would satisfy the constrain in (69) and

$$\begin{aligned} & -\sum_{s=1}^{S}\left(\boldsymbol{\tau}_s - \boldsymbol{\lambda}_s\right)^{\mathrm{T}}\tilde{\boldsymbol{\mu}}_{s,1} + \sum_{s=1}^{S}\left(\boldsymbol{\tau}_s + \boldsymbol{\lambda}_s\right)^{\mathrm{T}}\tilde{\boldsymbol{\mu}}_{s,2} + \mathbb{E}_{\mathrm{P_{te}}(x)}\nu^*(x) \\ =& -\sum_{s=1}^{S}\boldsymbol{\tau}_s^{\mathrm{T}}\left(\tilde{\boldsymbol{\mu}}_{s,1} - \tilde{\boldsymbol{\mu}}_{s,2}\right) + \sum_{s=1}^{S}\boldsymbol{\lambda}_s^{\mathrm{T}}\left(\tilde{\boldsymbol{\mu}}_{s,1} + \tilde{\boldsymbol{\mu}}_{s,2}\right) + \mathbb{E}_{\mathrm{P_{te}}(x)}\nu^*(x) \\ =& -\sum_{s=1}^{S}\boldsymbol{\tau}_s^{\mathrm{T}}\left(\bar{\boldsymbol{\mu}}_{s,1} - \bar{\boldsymbol{\mu}}_{s,2}\right) + \sum_{s=1}^{S}\boldsymbol{\lambda}_s^{\mathrm{T}}\left(\bar{\boldsymbol{\mu}}_{s,1} + \bar{\boldsymbol{\mu}}_{s,2}\right) + \mathbb{E}_{\mathrm{P_{te}}(x)}\nu^*(x) - 2\sum_{s=1}^{S}\lambda_s^{(i_s)}\epsilon \\ <& -\sum_{s=1}^{S}\boldsymbol{\tau}_s^{\mathrm{T}}\left(\bar{\boldsymbol{\mu}}_{s,1} - \bar{\boldsymbol{\mu}}_{s,2}\right) + \sum_{s=1}^{S}\boldsymbol{\lambda}_s^{\mathrm{T}}\left(\bar{\boldsymbol{\mu}}_{s,1} + \bar{\boldsymbol{\mu}}_{s,2}\right) + \mathbb{E}_{\mathrm{P_{te}}(x)}\nu^*(x) \\ =& -\sum_{s=1}^{S}\left(\boldsymbol{\tau}_s - \boldsymbol{\lambda}_s\right)^{\mathrm{T}}\bar{\boldsymbol{\mu}}_{s,1} + \sum_{s=1}^{S}\left(\boldsymbol{\tau}_s + \boldsymbol{\lambda}_s\right)^{\mathrm{T}}\bar{\boldsymbol{\mu}}_{s,2} + \mathbb{E}_{\mathrm{P_{te}}(x)}\nu^*(x) \end{aligned}$$

contradicting the fact that $\{\bar{\boldsymbol{\mu}}_{s,1}, \bar{\boldsymbol{\mu}}_{s,2}\}_{s\in[S]}, \nu^*(x)$ forms a solution of (69).

Then $\boldsymbol{\lambda}_s^{\mathrm{T}}(\bar{\boldsymbol{\mu}}_{s,1} + \bar{\boldsymbol{\mu}}_{s,2}) = \boldsymbol{\lambda}_s^{\mathrm{T}}|\bar{\boldsymbol{\mu}}_{s,1} - \bar{\boldsymbol{\mu}}_{s,2}|$ and taking $\boldsymbol{\mu}_s = \boldsymbol{\mu}_{s,1} - \boldsymbol{\mu}_{s,2}$ the Lagrange dual above is equivalent to

$$\begin{aligned} \min_{\{\boldsymbol{\mu}_s\}_{s\in[S]}, \nu(x)} \quad & -\sum_{s=1}^{S}\boldsymbol{\tau}_s^{\mathrm{T}}\boldsymbol{\mu}_s + \sum_{s=1}^{S}\boldsymbol{\lambda}_s^{\mathrm{T}}|\boldsymbol{\mu}_s| + \mathbb{E}_{\mathrm{P_{te}}(x)}\nu(x) \\ & \sum_{s=1}^{S}\Phi_{\alpha_s}(x,y)^{\mathrm{T}}\boldsymbol{\mu}_s - \nu(x) \leq -\ell(\mathrm{h},(x,y)), \; \forall x \in \mathcal{X}, y \in \mathcal{Y} \end{aligned}$$

that has the same value as $\max_{\mathrm{p}\in\mathcal{U}}\ell(\mathrm{h},\mathrm{p})$ since the constraints in (67) are affine and $\mathcal{U}$ is non-empty.

Therefore,

$$\begin{aligned} \min_{\mathrm{h}\in\mathrm{T}(\mathcal{X},\mathcal{Y})} \max_{\mathrm{p}\in\mathcal{U}} \ell(\mathrm{h},\mathrm{p}) = \min_{\mathrm{h},\{\boldsymbol{\mu}_s\}_{s\in[S]}, \nu(x)} \quad & -\sum_{s=1}^{S}\boldsymbol{\tau}_s^{\mathrm{T}}\boldsymbol{\mu}_s + \sum_{s=1}^{S}\boldsymbol{\lambda}_s^{\mathrm{T}}|\boldsymbol{\mu}_s| + \mathbb{E}_{\mathrm{P_{te}}(x)}\nu(x) \\ & \sum_{s=1}^{S}\Phi_{\alpha_s}(x,y)^{\mathrm{T}}\boldsymbol{\mu}_s - \nu(x) \leq -\ell(\mathrm{h},(x,y)), \; \forall x \in \mathcal{X}, y \in \mathcal{Y}. \end{aligned}$$

For 0-1-loss we have that

$$\sum_{s=1}^{S} \Phi_{\alpha_s}(x,y)^{\mathrm{T}} \boldsymbol{\mu}_s - \nu(x) \leq -1 + \mathrm{h}(y|x) \Rightarrow \sum_{s=1}^{S} \Phi_{\alpha_s}(x,y)^{\mathrm{T}} \boldsymbol{\mu}_s - \nu(x) + 1 \leq \mathrm{h}(y|x), \ \forall x \in \mathcal{X}, y \in \mathcal{Y} \quad (70)$$

and, since $\sum_{y \in \mathcal{Y}} \mathrm{h}(y|x) = 1$ implies that $\sum_{y \in \mathcal{C}} \mathrm{h}(y|x) \leq 1$ for all $\mathcal{C} \subseteq \mathcal{Y}$, we have

$$\Rightarrow \sum_{y \in \mathcal{C}} \left( \sum_{s=1}^{S} \Phi_{\alpha_s}(x,y)^{\mathrm{T}} \boldsymbol{\mu}_s - \nu(x) + 1 \right) \leq 1, \ \forall \mathcal{C} \subseteq \mathcal{Y}, x \in \mathcal{X}$$

$$\Rightarrow \nu(x) \geq 1 + \frac{\sum_{y \in \mathcal{C}} \sum_{s=1}^{S} \Phi_{\alpha_s}(x,y)^{\mathrm{T}} \boldsymbol{\mu}_s - 1}{|\mathcal{C}|}, \ \forall \mathcal{C} \subseteq \mathcal{Y}, x \in \mathcal{X}$$

$$\Rightarrow \nu(x) \geq \varphi_{01}(\{\boldsymbol{\mu}_s\}_{s=1}^{S}, x, \{\alpha_s\}_{s=1}^{S}), \ \forall x \in \mathcal{X}. \quad (71)$$

Therefore, for each set $\{\boldsymbol{\mu}_s\}_{s \in [S]}$, we have that any classification rule satisfying

$$\mathrm{h}(y|x) \geq \sum_{s=1}^{S} \Phi_{\alpha_s}(x,y)^{\mathrm{T}} \boldsymbol{\mu}_s - \varphi_{01}(\{\boldsymbol{\mu}_s\}_{s=1}^{S}, x, \{\alpha_s\}_{s=1}^{S}) + 1, \ \forall x \in \mathcal{X}, y \in \mathcal{Y} \quad (72)$$

is solution of

$$\min_{\mathrm{h}, \nu(x)} \mathbb{E}_{\mathrm{P_{te}}(x)} \nu(x) \qquad\qquad\qquad = \mathbb{E}_{\mathrm{P_{te}}(x)} \varphi_{01}(\{\boldsymbol{\mu}_s\}_{s=1}^{S}, x, \{\alpha_s\}_{s=1}^{S})$$

$$\sum_{s=1}^{S} \Phi_{\alpha_s}(x,y)^{\mathrm{T}} \boldsymbol{\mu}_s - \nu(x) + 1 \leq \mathrm{h}(y|x), \ \forall x \in \mathcal{X}, y \in \mathcal{Y}.$$

Specifically, since we are minimizing the function $\mathbb{E}_{\mathrm{P_{te}}(x)} \nu(x)$ and we have that $\nu(x) \geq \varphi_{01}(\{\boldsymbol{\mu}_s\}_{s=1}^{S}, x, \{\alpha_s\}_{s=1}^{S})$ for all $x \in \mathcal{X}$ (71), the best possible $\nu(x)$ would be at $\nu(x) = \varphi_{01}(\{\boldsymbol{\mu}_s\}_{s=1}^{S}, x, \{\alpha_s\}_{s=1}^{S})$ if it satisfies the restrictions of the optimization problem, i.e., if the classification rule $\mathrm{h}(y|x)$ satisfy (72).

To prove that $\sum_{y \in \mathcal{Y}} \mathrm{h}(y|x) = 1$ we will show that if

$$\mathcal{C}' \in \arg\max_{\mathcal{C} \in \mathcal{Y}} \frac{\sum_{s=1}^{S} \sum_{y' \in \mathcal{C}} \Phi_{\alpha_s}(x,y')^{\mathrm{T}} \boldsymbol{\mu}_s^* - 1}{|\mathcal{C}|}$$

and $\varphi_{01}(\{\boldsymbol{\mu}_s\}_{s=1}^{S}, x, \{\alpha_s\}_{s=1}^{S})$ is defined as in (35), then

$$\mathrm{h}^{\mathcal{U}}(y|x) = \sum_{s=1}^{S} \Phi_{\alpha_s}(x,y)^{\mathrm{T}} \boldsymbol{\mu}_s^* - \max_{\mathcal{C} \subseteq \mathcal{Y}} \frac{\sum_{s=1}^{S} \sum_{y' \in \mathcal{C}} \Phi_{\alpha_s}(x,y')^{\mathrm{T}} \boldsymbol{\mu}_s^* - 1}{|\mathcal{C}|} \leq 0$$

if $y \notin \mathcal{C}'$ and

$$\mathrm{h}^{\mathcal{U}}(y|x) = \sum_{s=1}^{S} \Phi_{\alpha_s}(x,y)^{\mathrm{T}} \boldsymbol{\mu}_s^* - \max_{\mathcal{C} \subseteq \mathcal{Y}} \frac{\sum_{s=1}^{S} \sum_{y' \in \mathcal{C}} \Phi_{\alpha_s}(x,y')^{\mathrm{T}} \boldsymbol{\mu}_s^* - 1}{|\mathcal{C}|} \geq 0$$

if $y \in \mathcal{C}'$, so that

$$\sum_{y \in \mathcal{Y}} \mathrm{h}(y|x) = \sum_{y \in \mathcal{C}'} \mathrm{h}(y|x)$$

$$= \sum_{y \in \mathcal{C}'} \left( \sum_{s=1}^{S} \Phi_{\alpha_s}(x,y)^{\mathrm{T}} \boldsymbol{\mu}_s^* - \frac{\sum_{s=1}^{S} \sum_{y' \in \mathcal{C}'} \Phi_{\alpha_s}(x,y')^{\mathrm{T}} \boldsymbol{\mu}_s^* - 1}{|\mathcal{C}'|} \right)$$

$$= \sum_{y \in \mathcal{C}'} \left( \sum_{s=1}^{S} \Phi_{\alpha_s}(x,y)^{\mathrm{T}} \boldsymbol{\mu}_s^* \right) - |\mathcal{C}'| \frac{\sum_{s=1}^{S} \sum_{y' \in \mathcal{C}'} \Phi_{\alpha_s}(x,y')^{\mathrm{T}} \boldsymbol{\mu}_s^*}{|\mathcal{C}'|} + 1 = 1.$$

To prove that if $y \notin \mathcal{C}'$, then $\sum_{s=1}^{S} \Phi_{\alpha_s}(x, y)^{\mathrm{T}} \boldsymbol{\mu}_s^* - \left( \varphi_{01} \left( \{\boldsymbol{\mu}_s\}_{s=1}^{S}, x, \{\alpha_s\}_{s=1}^{S} \right) - 1 \right) \leq 0$, we use

$$\frac{\sum_{s=1}^{S} \sum_{y' \in \mathcal{C}'} \Phi_{\alpha_s}(x, y')^T \boldsymbol{\mu}_s^* - 1}{|\mathcal{C}'|} \geq \frac{\sum_{s=1}^{S} \sum_{y' \in \mathcal{C}' \cup \{y\}} \Phi_{\alpha_s}(x, y')^T \boldsymbol{\mu}_s^* - 1}{|\mathcal{C}'| + 1}$$

$$= \frac{\sum_{s=1}^{S} \sum_{y' \in \mathcal{C}'} \Phi_{\alpha_s}(x, y')^T \boldsymbol{\mu}_s^* - 1}{|\mathcal{C}'| + 1} + \frac{\sum_{s=1}^{S} \Phi_{\alpha_s}(x, y)^T \boldsymbol{\mu}_s^*}{|\mathcal{C}'| + 1}.$$

Then,

$$\sum_{s=1}^{S} \Phi_{\alpha_s}(x, y)^T \boldsymbol{\mu}^* \leq \frac{|\mathcal{C}'| + 1}{|\mathcal{C}'|} \left( \sum_{s=1}^{S} \sum_{y' \in \mathcal{C}'} \Phi_{\alpha_s}(x, y')^T \boldsymbol{\mu}_s^* - 1 \right) - \left( \sum_{s=1}^{S} \sum_{y' \in \mathcal{C}'} \Phi_{\alpha_s}(x, y')^T \boldsymbol{\mu}_s^* - 1 \right)$$

$$= \frac{\sum_{s=1}^{S} \sum_{y' \in \mathcal{C}'} \Phi_{\alpha_s}(x, y')^T \boldsymbol{\mu}_s^* - 1}{|\mathcal{C}'|} = \varphi_{01} \left( \{\boldsymbol{\mu}_s\}_{s=1}^{S}, x, \{\alpha_s\}_{s=1}^{S} \right) - 1.$$

To prove that if $y \in \mathcal{C}'$, then $\sum_{s=1}^{S} \Phi_{\alpha_s}(x, y)^T \boldsymbol{\mu}_s^* - \left( \varphi_{01} \left( \{\boldsymbol{\mu}_s\}_{s=1}^{S}, x, \{\alpha_s\}_{s=1}^{S} \right) - 1 \right) \geq 0$, we use

$$\frac{\sum_{s=1}^{S} \sum_{y' \in \mathcal{C}'} \Phi_{\alpha_s}(x, y')^T \boldsymbol{\mu}_s^* - 1}{|\mathcal{C}'|} \geq \frac{\sum_{s=1}^{S} \sum_{y' \in \mathcal{C}' \setminus \{y\}} \Phi_{\alpha_s}(x, y')^T \boldsymbol{\mu}_s^* - 1}{|\mathcal{C}'| - 1}$$

$$\Rightarrow \frac{\sum_{s=1}^{S} \sum_{y' \in \mathcal{C}'} \Phi_{\alpha_s}(x, y')^T \boldsymbol{\mu}_s^* - 1}{|\mathcal{C}'|} + \frac{\sum_{s=1}^{S} \Phi_{\alpha_s}(x, y)^T \boldsymbol{\mu}_s^*}{|\mathcal{C}'| - 1} \geq \frac{\sum_{s=1}^{S} \sum_{y' \in \mathcal{C}'} \Phi_{\alpha_s}(x, y')^T \boldsymbol{\mu}_s^* - 1}{|\mathcal{C}'| - 1}$$

Then,

$$\sum_{s=1}^{S} \Phi_{\alpha_s}(x, y)^T \boldsymbol{\mu}_s^* \geq \left( \sum_{s=1}^{S} \sum_{y' \in \mathcal{C}'} \Phi_{\alpha_s}(x, y')^T \boldsymbol{\mu}_s^* - 1 \right) - \frac{|\mathcal{C}'| - 1}{|\mathcal{C}'|} \left( \sum_{s=1}^{S} \sum_{y' \in \mathcal{C}'} \Phi_{\alpha_s}(x, y')^T \boldsymbol{\mu}_s^* - 1 \right)$$

$$= \frac{\sum_{s=1}^{S} \sum_{y' \in \mathcal{C}'} \Phi_{\alpha_s}(x, y')^T \boldsymbol{\mu}_s^* - 1}{|\mathcal{C}'|} = \varphi_{01} \left( \{\boldsymbol{\mu}_s\}_{s=1}^{S}, x, \{\alpha_s\}_{s=1}^{S} \right) - 1.$$

The case of log-loss is analogous to the case for 0-1-loss above taking into account that

$$\sum_{s=1}^{S} \Phi_{\alpha_s}(x, y)^{\mathrm{T}} \boldsymbol{\mu}_s - \nu(x) \leq \log(\mathrm{h}(y|x)), \ \forall x \in \mathcal{X}, y \in \mathcal{Y}$$

$$\Rightarrow \sum_{y \in \mathcal{Y}} \exp \left\{ \sum_{s=1}^{S} \Phi_{\alpha_s}(x, y)^{\mathrm{T}} \boldsymbol{\mu}_s - \nu(x) \right\} \leq 1, \ \forall x \in \mathcal{X}$$

$$\Rightarrow \nu(x) \geq \log \left( \sum_{y \in \mathcal{Y}} \exp \left\{ \sum_{s=1}^{S} \Phi_{\alpha_s}(x, y)^{\mathrm{T}} \boldsymbol{\mu}_s \right\} \right), \ \forall x \in \mathcal{X}$$

$$\Rightarrow \nu(x) \geq \varphi_{\log}(\{\boldsymbol{\mu}_s\}_{s=1}^{S}, x, \{\alpha_s\}_{s=1}^{S}), \ \forall x \in \mathcal{X}.$$

$\square$

The lemma below is used in the proof of Theorem 4.2.

**Lemma C.1.** *Let $\mathcal{U}$ be the uncertainty set given by (32) for $\boldsymbol{\tau}_s \in \mathbb{R}^m, \boldsymbol{\lambda}_s \in \mathbb{R}^m$ for $s \in [S]$, and $\mathrm{h}$ be a classification rule. If*

$$\overline{R}(\mathcal{U}, \mathrm{h}) = \min_{\{\boldsymbol{\mu}_s\}_{s=1}^{S}} - \sum_{s=1}^{S} \boldsymbol{\tau}_s^T \boldsymbol{\mu}_s + \mathbb{E}_{\mathrm{p}_{te}(x)} \max_{y \in \mathcal{Y}} \left\{ 1 + \sum_{s=1}^{S} \alpha_s(x, y) \Phi(x, y)^T \boldsymbol{\mu}_s - \mathrm{h}(y|x) \right\} + \sum_{s=1}^{S} \boldsymbol{\lambda}_s^T |\boldsymbol{\mu}_s| \quad (73)$$

$$\overline{R}_{\log}(\mathcal{U}, \mathrm{h}) = \min_{\{\boldsymbol{\mu}_s\}_{s=1}^{S}} - \sum_{s=1}^{S} \boldsymbol{\tau}_s^T \boldsymbol{\mu}_s + \mathbb{E}_{\mathrm{p}_{te}(x)} \max_{y \in \mathcal{Y}} \left\{ \sum_{s=1}^{S} \alpha_s(x, y) \Phi(x, y)^T \boldsymbol{\mu}_s - \log \mathrm{h}(y|x) \right\} + \sum_{s=1}^{S} \boldsymbol{\lambda}_s^T |\boldsymbol{\mu}_s| \quad (74)$$

*then, for any* $p \in \mathcal{U}$

$$\ell_{01}(\mathrm{h}, \mathrm{p}) \leq \overline{R}_{01}(\mathcal{U}, \mathrm{h}) \tag{75}$$

$$\ell_{\log}(\mathrm{h}, \mathrm{p}) \leq \overline{R}_{\log}(\mathcal{U}, \mathrm{h}). \tag{76}$$

***Proof of Lemma C.1.*** The case $\mathcal{U} = \emptyset$ is trivial. For the case where $\mathcal{U} \neq \emptyset$, we will first calculate the Lagrange dual of the optimization problem $\min_{\hat{p} \in \mathcal{U}} \mathbb{E}_{\hat{p}} q$ for a general function $q : \mathcal{X} \times \mathcal{Y} \to \mathbb{R}$. Then we will consider the fact that for any $p \in \mathcal{U}$ and $h \in T(\mathcal{X}, \mathcal{Y})$,

$$\min_{\hat{p} \in \mathcal{U}} \ell(\mathrm{h}, \hat{p}) \leq \ell(\mathrm{h}, \mathrm{p}) \leq \max_{\hat{p} \in \mathcal{U}} \ell(\mathrm{h}, \hat{p})$$

and

$$\max_{\hat{p} \in \mathcal{U}} \ell_{01}(\mathrm{h}, \hat{p}) = -\min_{\hat{p} \in \mathcal{U}} \mathbb{E}_{\hat{p}}\{\mathrm{h}(y|x) - 1\}$$

$$\max_{\hat{p} \in \mathcal{U}} \ell_{\log}(\mathrm{h}, \hat{p}) = -\min_{\hat{p} \in \mathcal{U}} \mathbb{E}_{\hat{p}} \log \mathrm{h}(y|x)$$

for 0-1-loss and log-loss respectively. First, we have that $\min_{\hat{p} \in \mathcal{U}} \mathbb{E}_{\hat{p}} q$ is equal to

$$\min_{\hat{\mathbf{p}}} \quad \mathbf{q}^{\mathrm{T}}\hat{\mathbf{p}} + I_{+}(\hat{\mathbf{p}})$$

$$\text{s.t.} \quad -\sum_{y \in \mathcal{Y}} \hat{p}(x, y) = -p_{\mathrm{te}}(x) \text{ for all } x \in \mathcal{X}$$

$$\boldsymbol{\tau}_s - \boldsymbol{\lambda}_s \preceq \boldsymbol{\Phi}_{\alpha_s}^{\mathrm{T}}\hat{\mathbf{p}} \preceq \boldsymbol{\tau}_s + \boldsymbol{\lambda}_s, \ \forall s \in [S] \tag{77}$$

where $\hat{\mathbf{p}}$, $\mathbf{q}$, $\{\boldsymbol{\Phi}_{\alpha_s}\}_{s=1}^S$ denote the vectors and set of matrices with rows $\hat{p}(x, y)$, $q(x, y)$ and $\alpha_s(x, y)\Phi(x, y)^{\mathrm{T}}$, respectively, for $x \in \mathcal{X}$, $y \in \mathcal{Y}$, and

$$I_{+}(\hat{\mathbf{p}}) = \begin{cases} 0 & \text{if } \hat{\mathbf{p}} \succeq \mathbf{0} \\ \infty & \text{otherwise.} \end{cases}$$

Optimization problem (77) has Lagrange dual

$$\max_{\{\boldsymbol{\mu}_{s,1}\}_{s=1}^S, \{\boldsymbol{\mu}_{s,2}\}_{s=1}^S, \nu(x)} \sum_{s=1}^S (\boldsymbol{\tau}_s - \boldsymbol{\lambda}_s)^{\mathrm{T}}\boldsymbol{\mu}_{s,1} - \sum_{s=1}^S (\boldsymbol{\tau}_s + \boldsymbol{\lambda}_s)^{\mathrm{T}}\boldsymbol{\mu}_{s,2} + \mathbb{E}_{p_{\mathrm{te}}(x)}\nu(x) - f^*\left(\sum_{s=1}^S \boldsymbol{\Phi}_{\alpha_s}(\boldsymbol{\mu}_{s,1} - \boldsymbol{\mu}_{s,2}) + \boldsymbol{\nu}\right)$$

$$\text{s.t.} \quad \{\boldsymbol{\mu}_{s,1}\}_{s=1}^S, \{\boldsymbol{\mu}_{s,2}\}_{s=1}^S \succeq \mathbf{0}$$

where $\boldsymbol{\nu}$ denotes the vector in $\mathbb{R}^{|\mathcal{X}||\mathcal{Y}|}$ with component corresponding with $(x, y)$ for $x \in \mathcal{X}$, $y \in \mathcal{Y}$ given by $\nu(x)$, and $f^*$ is the conjugate function of $f(\hat{\mathbf{p}}) = \mathbf{q}^{\mathrm{T}}\hat{\mathbf{p}} + I_{+}(\hat{\mathbf{p}})$ that becomes

$$f^*(\boldsymbol{w}) = \begin{cases} 0 & \text{if } \mathbf{w} \preceq \mathbf{q} \\ \infty & \text{otherwise.} \end{cases}$$

Therefore, the previous Lagrange dual becomes

$$\max_{\{\boldsymbol{\mu}_{s,1}\}_{s=1}^S, \{\boldsymbol{\mu}_{s,2}\}_{s=1}^S, \nu(x)} \sum_{s=1}^S (\boldsymbol{\tau}_s - \boldsymbol{\lambda}_s)^{\mathrm{T}}\boldsymbol{\mu}_{s,1} - \sum_{s=1}^S (\boldsymbol{\tau}_s + \boldsymbol{\lambda}_s)^{\mathrm{T}}\boldsymbol{\mu}_{s,2} + \mathbb{E}_{p_{\mathrm{te}}(x)}\nu(x)$$

$$\text{s.t.} \quad \{\boldsymbol{\mu}_{s,1}\}_{s=1}^S, \{\boldsymbol{\mu}_{s,2}\}_{s=1}^S \succeq \mathbf{0}$$

$$\sum_{s=1}^S \boldsymbol{\Phi}_{\alpha_s}(\boldsymbol{\mu}_{s,1} - \boldsymbol{\mu}_{s,2}) + \boldsymbol{\nu} \preceq \mathbf{q} \text{ for } s \in [S]$$

which is equivalent to

$$\max_{\{\boldsymbol{\mu}_{s,1}\}_{s=1}^S, \{\boldsymbol{\mu}_{s,2}\}_{s=1}^S} \sum_{s=1}^S (\boldsymbol{\tau}_s - \boldsymbol{\lambda}_s)^{\mathrm{T}}\boldsymbol{\mu}_{s,1} - \sum_{s=1}^S (\boldsymbol{\tau}_s + \boldsymbol{\lambda}_s)^{\mathrm{T}}\boldsymbol{\mu}_{s,2}$$

$$+ \mathbb{E}_{p_{\mathrm{te}}(x)} \min_{y \in \mathcal{Y}} \left\{ q(x, y) - \sum_{s=1}^S \alpha_s(x, y)\Phi(x, y)^{\mathrm{T}}(\boldsymbol{\mu}_{s,1} - \boldsymbol{\mu}_{s,2}) \right\}$$

$$\text{s.t.} \quad \{\boldsymbol{\mu}_{s,1}\}_{s=1}^S, \{\boldsymbol{\mu}_{s,2}\}_{s=1}^S \succeq \mathbf{0}.$$

Taking $\boldsymbol{\mu}_s = \boldsymbol{\mu}_{s,1} - \boldsymbol{\mu}_{s,2}$, for $s = 1, 2, \ldots, S$, the Lagrange dual problem is equivalent to

$$\max_{\{\boldsymbol{\mu}_s\}_{s=1}^S} \sum_{s=1}^S \boldsymbol{\tau}_s^{\mathrm{T}} \boldsymbol{\mu}_s + \mathbb{E}_{\mathrm{p}_{\mathrm{te}}(x)} \min_{y \in \mathcal{Y}} \left\{ q(x,y) - \sum_{s=1}^S \alpha_s(x,y) \Phi(x,y)^{\mathrm{T}} \boldsymbol{\mu}_s \right\} - \sum_{s=1}^S \boldsymbol{\lambda}_s^{\mathrm{T}} |\boldsymbol{\mu}_s|$$

that has the same value as its primal $\min_{\hat{\mathrm{p}} \in \mathcal{U}} \mathbb{E}_{\hat{\mathrm{p}}} q$ since the constraints defining $\mathcal{U}$ are affine and $\mathcal{U} \neq \emptyset$. Then, we have that

$$\max_{\hat{\mathrm{p}} \in \mathcal{U}} \ell_{01}(\mathrm{h}, \hat{\mathrm{p}}) = -\min_{\hat{\mathrm{p}} \in \mathcal{U}} \mathbb{E}_{\hat{\mathrm{p}}} \{\mathrm{h}(y|x) - 1\}$$

$$= \min_{\{\boldsymbol{\mu}_s\}_{s=1}^S} -\sum_{s=1}^S \boldsymbol{\tau}_s^{\mathrm{T}} \boldsymbol{\mu}_s + \mathbb{E}_{\mathrm{p}_{\mathrm{te}}(x)} \max_{y \in \mathcal{Y}} \left\{ 1 + \sum_{s=1}^S \alpha_s(x,y) \Phi(x,y)^{\mathrm{T}} \boldsymbol{\mu}_s - \mathrm{h}(y|x) \right\} + \sum_{s=1}^S \boldsymbol{\lambda}_s^{\mathrm{T}} |\boldsymbol{\mu}_s|$$

$$\max_{\hat{\mathrm{p}} \in \mathcal{U}} \ell_{\log}(\mathrm{h}, \hat{\mathrm{p}}) = -\min_{\hat{\mathrm{p}} \in \mathcal{U}} \mathbb{E}_{\hat{\mathrm{p}}} \log \mathrm{h}(y|x)$$

$$= \min_{\{\boldsymbol{\mu}_s\}_{s=1}^S} -\sum_{s=1}^S \boldsymbol{\tau}_s^{\mathrm{T}} \boldsymbol{\mu}_s + \mathbb{E}_{\mathrm{p}_{\mathrm{te}}(x)} \max_{y \in \mathcal{Y}} \left\{ \sum_{s=1}^S \alpha_s(x,y) \Phi(x,y)^{\mathrm{T}} \boldsymbol{\mu}_s - \log \mathrm{h}(y|x) \right\} + \boldsymbol{\lambda}^{\mathrm{T}} |\boldsymbol{\mu}|$$

$\square$

***Proof of Theorem 4.2.*** Let $\mathcal{U}^\infty$ be the uncertainty set given by the exact mean vectors $\boldsymbol{\tau}_{\infty,s} = \mathbb{E}_{\mathrm{p}_{\mathrm{te}}} \Phi_{\alpha_s}(x, y)$, for $s = 1, 2, \ldots, S$, i.e.,

$$\mathcal{U}^\infty = \{\mathrm{p} \in \Delta\left(\mathcal{X} \times \mathcal{Y}\right) : \mathbb{E}_{\mathrm{p}} \Phi_{\alpha_s}(x, y) = \boldsymbol{\tau}_{\infty,s}, \text{ for } s \in [S], \text{ and } \mathrm{p}(x) = \mathrm{p}_{\mathrm{te}}(x), \forall x \in \mathcal{X}\} \quad (78)$$

It is clear that we have $\mathrm{p}_{\mathrm{te}}(x, y) \in \mathcal{U}^\infty$, then using Lemma C.1 for 0-1-loss and the definition of $\mathrm{h}(y|x)$ in (37), we have that

$$R(\mathrm{h}^{\mathcal{U}}) \leq \overline{R}_{01}(\mathcal{U}^\infty, \mathrm{h}^{\mathcal{U}}) = \min_{\{\boldsymbol{\mu}_s\}_{s=1}^S} -\sum_{s=1}^S \boldsymbol{\tau}_{\infty,s}^{\mathrm{T}} \boldsymbol{\mu}_s + \mathbb{E}_{\mathrm{p}_{\mathrm{te}}(x)} \max_{y \in \mathcal{Y}} \left\{ 1 + \sum_{s=1}^S \alpha_s(x,y) \Phi(x,y)^{\mathrm{T}} \boldsymbol{\mu}_s - \mathrm{h}(y|x) \right\}$$

$$\leq -\sum_{s=1}^S \boldsymbol{\tau}_{\infty,s}^{\mathrm{T}} \boldsymbol{\mu}_s^* + \mathbb{E}_{\mathrm{p}_{\mathrm{te}}(x)} \max_{y \in \mathcal{Y}} \left\{ 1 + \sum_{s=1}^S \alpha_s(x,y) \Phi(x,y)^{\mathrm{T}} \boldsymbol{\mu}_s^* - \mathrm{h}(y|x) \right\} \quad (79)$$

$$\leq -\sum_{s=1}^S \boldsymbol{\tau}_{\infty,s}^{\mathrm{T}} \boldsymbol{\mu}_s^* + \mathbb{E}_{\mathrm{p}_{\mathrm{te}}(x)} \max_{y \in \mathcal{Y}} \left\{ 1 + \max_{\mathcal{C} \subseteq \mathcal{Y}} \frac{\sum_{s=1}^S \sum_{y \in \mathcal{C}} \Phi_{\alpha_s}(x,y)^{\mathrm{T}} \boldsymbol{\mu}_s^* - 1}{|\mathcal{C}|} \right\} \quad (80)$$

$$= -\sum_{s=1}^S \boldsymbol{\tau}_{\infty,s}^{\mathrm{T}} \boldsymbol{\mu}_s^* + \mathbb{E}_{\mathrm{p}_{\mathrm{te}}(x)} \left\{ 1 + \max_{\mathcal{C} \subseteq \mathcal{Y}} \frac{\sum_{s=1}^S \sum_{y \in \mathcal{C}} \Phi_{\alpha_s}(x,y)^{\mathrm{T}} \boldsymbol{\mu}_s^* - 1}{|\mathcal{C}|} \right\} \quad (81)$$

$$\leq -\sum_{s=1}^S \boldsymbol{\tau}_s^{\mathrm{T}} \boldsymbol{\mu}_s^\infty + \mathbb{E}_{\mathrm{p}_{\mathrm{te}}(x)} \left\{ 1 + \max_{\mathcal{C} \subseteq \mathcal{Y}} \frac{\sum_{y \in \mathcal{C}} \sum_{s=1}^S \Phi_{\alpha_s}(x,y)^{\mathrm{T}} \boldsymbol{\mu}_s^\infty - 1}{|\mathcal{C}|} \right\}$$

$$+ \sum_{s=1}^S \boldsymbol{\lambda}_s^{\mathrm{T}} |\boldsymbol{\mu}_s^\infty| + \sum_{s=1}^S (\boldsymbol{\tau}_s - \boldsymbol{\tau}_{\infty,s})^{\mathrm{T}} \boldsymbol{\mu}_s^* - \sum_{s=1}^S \boldsymbol{\lambda}_s^{\mathrm{T}} |\boldsymbol{\mu}_s^*|$$

$$= R^\infty + \sum_{s=1}^S \boldsymbol{\lambda}_s^{\mathrm{T}} (|\boldsymbol{\mu}_s^\infty| - |\boldsymbol{\mu}_s^*|) + \sum_{s=1}^S (\boldsymbol{\tau}_{\infty,s} - \boldsymbol{\tau}_s)^{\mathrm{T}} \boldsymbol{\mu}_s^\infty + \sum_{s=1}^S (\boldsymbol{\tau}_s - \boldsymbol{\tau}_{\infty,s})^{\mathrm{T}} \boldsymbol{\mu}_s^*$$

$$\leq R^\infty + \sum_{s=1}^S \boldsymbol{\lambda}_s^{\mathrm{T}} (|\boldsymbol{\mu}_s^\infty| - |\boldsymbol{\mu}_s^*|) + \sum_{s=1}^S |\boldsymbol{\tau}_s - \boldsymbol{\tau}_{\infty,s}|^{\mathrm{T}} |\boldsymbol{\mu}_s^\infty - \boldsymbol{\mu}_s^*|. \quad (82)$$

where, for inequality (79)-(80), we have used the fact that

$$\mathrm{h}^{\mathcal{U}}(y|x) \geq \sum_{s=1}^S \alpha_s(x,y) \Phi(x,y)^{\mathrm{T}} \boldsymbol{\mu}_s^* - \max_{\mathcal{C} \subseteq \mathcal{Y}} \frac{\sum_{y \in \mathcal{C}} \sum_{s=1}^S \Phi_{\alpha_s}(x,y)^{\mathrm{T}} \boldsymbol{\mu}_s^* - 1}{|\mathcal{C}|}.$$

For log-loss, using Lemma C.1 and the definition of h($y|x$) in (38), we have that

$$R(\mathrm{h}^{\mathcal{U}}) \leq \overline{R}_{\log}(\mathcal{U}^{\infty}, \mathrm{h}^{\mathcal{U}}) = \min_{\{\boldsymbol{\mu}_s\}_{s=1}^S} -\sum_{s=1}^S \boldsymbol{\tau}_{\infty,s}^{\mathrm{T}} \boldsymbol{\mu}_s + \mathbb{E}_{\mathrm{P_{te}}(x)} \max_{y \in \mathcal{Y}} \left\{ \sum_{s=1}^S \alpha_s(x,y) \Phi(x,y)^{\mathrm{T}} \boldsymbol{\mu}_s - \log \mathrm{h}(y|x) \right\}$$

$$\leq -\sum_{s=1}^S \boldsymbol{\tau}_{\infty,s}^{\mathrm{T}} \boldsymbol{\mu}_s^* + \mathbb{E}_{\mathrm{P_{te}}(x)} \max_{y \in \mathcal{Y}} \left\{ \sum_{s=1}^S \alpha_s(x,y) \Phi(x,y)^{\mathrm{T}} \boldsymbol{\mu}_s^* - \log \mathrm{h}(y|x) \right\} \tag{83}$$

$$\leq -\sum_{s=1}^S \boldsymbol{\tau}_{\infty,s}^{\mathrm{T}} \boldsymbol{\mu}_s^* + \mathbb{E}_{\mathrm{P_{te}}(x)} \max_{y \in \mathcal{Y}} \left\{ \log \sum_{y \in \mathcal{Y}} \exp \left\{ \sum_{s=1}^S \Phi_{\alpha_s}(x,y)^{\mathrm{T}} \boldsymbol{\mu}_s^* \right\} \right\} \tag{84}$$

$$= -\sum_{s=1}^S \boldsymbol{\tau}_{\infty,s}^{\mathrm{T}} \boldsymbol{\mu}_s^* + \mathbb{E}_{\mathrm{P_{te}}(x)} \log \sum_{y \in \mathcal{Y}} \exp \left\{ \sum_{s=1}^S \Phi_{\alpha_s}(x,y)^{\mathrm{T}} \boldsymbol{\mu}_s^* \right\} \tag{85}$$

$$\leq -\sum_{s=1}^S \boldsymbol{\tau}_s^{\mathrm{T}} \boldsymbol{\mu}_s^{\infty} + \mathbb{E}_{\mathrm{P_{te}}(x)} \log \sum_{y \in \mathcal{Y}} \exp \left\{ \sum_{s=1}^S \Phi_{\alpha_s}(x,y)^{\mathrm{T}} \boldsymbol{\mu}_s^{\infty} \right\}$$

$$+ \sum_{s=1}^S \boldsymbol{\lambda}_s^{\mathrm{T}} |\boldsymbol{\mu}_s^{\infty}| + \sum_{s=1}^S (\boldsymbol{\tau}_s - \boldsymbol{\tau}_{\infty,s})^{\mathrm{T}} \boldsymbol{\mu}_s^* - \sum_{s=1}^S \boldsymbol{\lambda}_s^{\mathrm{T}} |\boldsymbol{\mu}_s^*|$$

$$= R^{\infty} + \sum_{s=1}^S \boldsymbol{\lambda}_s^{\mathrm{T}} (|\boldsymbol{\mu}_s^{\infty}| - |\boldsymbol{\mu}_s^*|) + \sum_{s=1}^S (\boldsymbol{\tau}_{\infty,s} - \boldsymbol{\tau}_s)^{\mathrm{T}} \boldsymbol{\mu}_s^{\infty} + \sum_{s=1}^S (\boldsymbol{\tau}_s - \boldsymbol{\tau}_{\infty,s})^{\mathrm{T}} \boldsymbol{\mu}_s^*$$

$$\leq R^{\infty} + \sum_{s=1}^S \boldsymbol{\lambda}_s^{\mathrm{T}} (|\boldsymbol{\mu}_s^{\infty}| - |\boldsymbol{\mu}_s^*|) + \sum_{s=1}^S |\boldsymbol{\tau}_s - \boldsymbol{\tau}_{\infty,s}|^{\mathrm{T}} |\boldsymbol{\mu}_s^{\infty} - \boldsymbol{\mu}_s^*|. \tag{86}$$

Using Hoeffding's inequality, we have that with probability at least $1 - \delta$

$$\max_{s \in [S]} \left\| \frac{1}{n} \sum_{i=1}^n \beta_s(x_i, y_i) \Phi(x_i, y_i) - \mathbb{E}_{\mathrm{P_{te}}} \alpha_s(x,y) \Phi(x,y) \right\|_{\infty} \leq \max_{s \in [S]} ||\Phi||_{\infty} \sqrt{2 \frac{||\beta_s(x)||_{\infty}^2}{n} \log \frac{2}{\delta}}$$

$$\leq \max_{s \in [S]} M \sqrt{2 \frac{B^2}{Dn} \log \frac{2m}{\delta}}$$

$$\leq M \sqrt{2 \frac{B^2}{Dn} \log \frac{2m}{\delta}} \tag{87}$$

Then, using (87) and Hölder inequality in (86), we have that

$$R(\mathrm{h}^{\mathcal{U}}) \leq R^{\infty} + \sum_{s=1}^S \boldsymbol{\lambda}_s^{\mathrm{T}} (|\boldsymbol{\mu}_s^{\infty}| - |\boldsymbol{\mu}_s^*|) + M \|\boldsymbol{\mu}_s^{\infty} - \boldsymbol{\mu}_s^*\|_1 \sqrt{2 \frac{B^2}{Dn} \log \frac{2m}{\delta}} \tag{88}$$

with probability at least $1 - \delta$. $\qquad\qquad\square$

## D  Main Results of the Section 4 when each domain has different number of training samples

This section presents the main results of the Section 4 for the case with general number of training samples in each source.

**Reference Solutions.** We consider reference solutions of the form

$$\alpha_s(x,y) = \min \left( \delta_s(x,y), \frac{\mathrm{p}_s(x,y)}{\mathrm{p_{te}}(x,y)} C_s \right), \quad \beta_s(x,y) = \min \left( \delta_s(x,y) \frac{\mathrm{p_{te}}(x,y)}{\mathrm{p}_s(x,y)}, C_s \right) \tag{89}$$

for any $C > 0$, where

$$\delta_s(x,y) = \frac{\mathrm{p}_s(x,y)\sqrt{n_s}}{\sum_{s'=1}^S \mathrm{p}_{s'}(x,y)\sqrt{n_{s'}}}. \tag{90}$$

**Generalization bounds.** Using the reference weight functions defined in (89) and (90), we get the following generalization of Theorem 4.2.

**Theorem D.1.** *Let $\mathcal{U}$ be a non-empty uncertainty set given by (32) and $\mathrm{h}^{\mathcal{U}}$ be an $\ell$-MRC for $\mathcal{U}$. If weights $\{\alpha_s(x,y)\}_{s\in[S]}$ and $\{\beta_s(x,y)\}_{s\in[S]}$ are given by (89) with $C_s = B_s/\sqrt{D}$ for $D \geq 1$ and*

$$B_s = \sqrt{n_s} \sup_{x\in\mathcal{X}} \frac{\mathrm{p}_{te}(x,y)}{\sum_{s'=1}^{S} \sqrt{n_{s'}}\mathrm{p}_{s'}(x,y)}. \tag{91}$$

*Then, with probability at least $1 - \delta$ we have that*

$$R(\mathrm{h}^{\mathcal{U}}) \leq R^{\infty} + \sum_{s=1}^{S} \boldsymbol{\lambda}_s^T (|\boldsymbol{\mu}_s^{\infty}| - |\boldsymbol{\mu}_s^*|) + M\|\boldsymbol{\mu}^{\infty} - \boldsymbol{\mu}^*\|_1 \sup_{x\in\mathcal{X}, y\in\mathcal{Y}} \frac{\mathrm{p}_{te}(x,y)}{\sum_{s'=1}^{S} \sqrt{n_{s'}}\mathrm{p}_{s'}(x,y)} \sqrt{\frac{2}{D} \log \frac{2m}{\delta}}$$

*where*

$$\boldsymbol{\mu}^{\infty} = \begin{bmatrix} \boldsymbol{\mu}_1^{\infty} \\ \boldsymbol{\mu}_2^{\infty} \\ \vdots \\ \boldsymbol{\mu}_S^{\infty} \end{bmatrix}, \quad \boldsymbol{\mu}^* = \begin{bmatrix} \boldsymbol{\mu}_1^* \\ \boldsymbol{\mu}_2^* \\ \vdots \\ \boldsymbol{\mu}_S^* \end{bmatrix}$$

*and $M$ is a constant satisfying $\|\Phi(x,y)\|_{\infty} \leq M$ for all $x \in \mathcal{X}$, $y \in \mathcal{Y}$.*

*Proof.* The proof is analogous to the proof of Theorem 4.2, considering $\{\alpha_s(x,y)\}_{s=1}^{S}$ and $\{\beta_s(x,y)\}_{s=1}^{S}$ as defined in (89). ☐

Using the weights given by (89) and (90) we have that the ratios $B_s/\sqrt{n_s}$ do not depend on the source $s$, so we can avoid that training sources with a smaller number of training samples penalise more in reducing the estimation error.

## E    Performance guarantees of MS-KMM

The MS-KMM approach in (56) is an empirical version of the population problem given the exact expectation as

$$\min_{\{\beta_s(x),\alpha_s(x)\}_{s=1}^{S}} \sum_{s=1}^{S} \left\| \mathbb{E}_{\mathrm{p}_{te}} \alpha_s(x) K_x(x) - \mathbb{E}_{\mathrm{p}_s} \beta_s(x) K_x(x) \right\|_{\mathcal{H}_x}^2$$

$$\begin{aligned} \text{s.t.} \quad & 0 \leq \beta_s(x) \leq B_s/\sqrt{D}, \text{ for } s \in [S] \\ & 0 \leq \alpha_s(x) \leq 1, \text{ for } s \in [S] \\ & \mathbb{E}_{\mathrm{p}_{te}(x)}\alpha(x) = \mathbb{E}_{\mathrm{p}_s(x)}\beta_s(x), \text{ for } s \in [S] \\ & \left| \sum_{s=1}^{S} \alpha_s(x) - 1 \right| \leq \left( 1 - \frac{1}{\sqrt{D}} \right), \text{ for } x \in \mathcal{X}. \end{aligned} \tag{92}$$

Since (89) is a feasible solution of (92), the value at the optimum is zero. Then, the solutions of (92), $\{\hat{\beta}_s(x)\}_{s\in[S]}$, $\{\hat{\alpha}_s(x)\}_{s\in[S]}$, provide consistent estimators of expectations because

$$\mathbb{E}_{\mathrm{p}_{te}(x,y)}\hat{\alpha}_s(x)\Phi(x,y) = \mathbb{E}_{\mathrm{p}_s(x,y)}\hat{\beta}_s(x)\Phi(x,y) \tag{93}$$

is satisfied for $s \in [S]$ if the kernel $k_x$ is characteristic or if $\mathbb{E}_{\mathrm{p}_{te}(y|x)}\Phi(x,y)$ belongs to $\mathcal{H}_x$, analogously as shown in (Yu & Szepesvári, 2012).

The following theorem presents bounds for the discrepancy between empirical means withing the feature space for solutions of (92), when we have a limited amount of samples from each source and testing distribution.

**Theorem E.1.** *For $s \in [S]$, if $\hat{\beta}_s(x)$ and $\hat{\alpha}_s(x)$ are solutions of (92), with probability at least $1 - \delta$ we have that*

$$\left\| \frac{1}{n_s} \sum_{i=1}^{n_s} \hat{\beta}_s(x_{s,i}) K_x(x_{s,i}) - \frac{1}{t} \sum_{j=1}^{t} \hat{\alpha}_s(x_{n+j}) K_x(x_{n+j}) \right\|_{\mathcal{H}_x}$$

$$\leq \left( 1 + \sqrt{2 \log \frac{2}{\delta}} \right) \kappa \sqrt{ \left( \frac{1}{D} \left( \sup_{x \in \mathcal{X}} \frac{\mathrm{p}_{te}(x)}{\sum_{s'=1}^{S} \sqrt{n_{s'}} \mathrm{p}_{s'}(x)} \right)^2 + \frac{1}{t} \right)} \tag{94}$$

*where the constant $\kappa$ satisfies $|k_x(x,x)| \leq \kappa^2$ for all $x \in \mathcal{X}$.*

**Learning jointly achieves better estimation errors than learning independently.** The difference between the empirical means in feature space of solutions of (92) depend on the maximum of the values of the weight functions $\alpha_s(x)$ and $\beta_s(x)$ associated with each source $s \in [S]$. In (Segovia-Martín et al., 2023), $\|\beta_s(x)\|_\infty \leq B_s / \sqrt{D}$, with $B_s = \sup_{x \in \mathcal{X}} \mathrm{p}_{te}(x)/\mathrm{p}_s(x)$, resulting in bounds of the order $\mathcal{O}(\sqrt{B_s / Dn_s + 1/t})$. In methods that obtain weights based on reweighted techniques (Sun et al., 2011; Wang et al., 2023), $\|\beta_s(x)\|_\infty \leq B_s$ with $B_s = \sup_{x \in \mathcal{X}} \mathrm{p}_{te}(x)/\mathrm{p}_s(x)$ resulting in bounds of the order $\mathcal{O}(\sqrt{B_s / n_s + 1/t})$.

Therefore, since the $B_s$ defined in (91) is smaller than in for DW-GCS (Segovia-Martín et al., 2023), and KMM (Huang et al., 2006; Gretton et al., 2008), the proposed MS-KMM significantly decreases the estimation error.

***Proof of Theorem E.1.*** The proof is similar to the proof of Theorem 4.1 in Segovia-Martín et al. (2023). We consider $n_s + t$ independent random variables taking values in the Hilbert space $\mathcal{H}_x$ as follows

$$f_{s,i} = \begin{cases} \frac{1}{n_s} \hat{\beta}(x_{s,i}) K_x(x_{s,i}) & \text{for } i = 1, 2, \dots, n_s \\[2mm] -\frac{1}{t} \hat{\alpha}(x_{n+i}) K_x(x_{n+i}) & \text{if } s = 0, \text{ and } i = 1, 2, \dots, t \end{cases} \tag{95}$$

and we will first bound $\| \sum_{i=1}^{n_s} f_{s,i} + \sum_{i=1}^{t} f_{0,i} \|_{\mathcal{H}_x}$. We have that,

$$\|f_{s,i}\|_{\mathcal{H}_x} \leq \begin{cases} \frac{1}{n} \frac{B_s}{\sqrt{D}} \kappa & \text{for } i = 1, 2, \dots, n_s \\[2mm] \frac{1}{t} \kappa & \text{if } s = 0, \text{ and } i = 1, 2, \dots, t. \end{cases} \tag{96}$$

Taking $v_s = \kappa^2 \left( \frac{B_s^2}{Dn_s} + \frac{1}{t} \right)$ and using the bounded differences inequality, we have that, for all $l \geq \sqrt{v_s}$

$$\mathbb{P} \left\{ \left\| \sum_{i=1}^{n_s} f_{s,i} + \sum_{i=1}^{t} f_{0,i} \right\|_{\mathcal{H}_x} > l \right\} = \mathbb{P} \left\{ \left\| \sum_{i=1}^{n_s} f_{s,i} + \sum_{i=1}^{t} f_{0,i} \right\|_{\mathcal{H}_x} - \mathbb{E} \left\| \sum_{i=1}^{n_s} f_{s,i} + \sum_{i=1}^{t} f_{0,i} \right\|_{\mathcal{H}_x} > l - \mathbb{E} \left\| \sum_{i=1}^{n_s} f_{s,i} + \sum_{i=1}^{t} f_{0,i} \right\|_{\mathcal{H}_x} \right\}$$

$$\leq \exp \left\{ - \frac{ \left( l - \mathbb{E} \left\| \sum_{i=1}^{n_s} f_{s,i} + \sum_{i=1}^{t} f_{0,i} \right\|_{\mathcal{H}_x} \right)^2 }{2 v_s} \right\}. \tag{97}$$

Finally, using Hölder's inequality and by independence, we have that

$$\mathbb{E} \left\| \sum_{i=1}^{n_s} f_{s,i} + \sum_{i=1}^{t} f_{0,i} \right\|_{\mathcal{H}_x} \leq \sqrt{ \mathbb{E} \left\| \sum_{i=1}^{n_s} f_{s,i} + \sum_{i=1}^{t} f_{0,i} \right\|_{\mathcal{H}_x}^2 } = \sqrt{ \sum_{i=1}^{n_s} \mathbb{E} \|f_{s,i}\|_{\mathcal{H}_x}^2 + \sum_{i=1}^{t} \mathbb{E} \|f_{0,i}\|_{\mathcal{H}_x}^2 } \leq \sqrt{v_s}.$$

Therefore,

$$\exp\left\{-\frac{\left(l-\sqrt{v_s}\right)^2}{2v_s}\right\} = \exp\left\{-\frac{\left(l-\sqrt{\kappa^2\left(\frac{B_s^2}{Dn_s}+\frac{1}{t}\right)}\right)^2}{2\kappa^2\left(\frac{B_s^2}{Dn_s}+\frac{1}{t}\right)}\right\}$$

so that,

$$\left\|\frac{1}{n}\sum_{i=1}^{n_s}\hat{\beta}_s(x_{s,i})K_x(x_{s,i}) - \frac{1}{t}\sum_{i=1}^{t}\hat{\alpha}_s(x_{n+i})K_x(x_{n+i})\right\|_{\mathcal{H}_x}$$

$$\leq \left(1+\sqrt{2\log\frac{2}{\delta}}\right)\kappa\sqrt{\left(\frac{1}{D}\left(\sup_{x\in\mathcal{X}}\frac{\mathrm{p_{te}}(x)}{\sum_{s'=1}^{S}\sqrt{n_{s'}}\mathrm{p}_{s'}(x)}\right)^2+\frac{1}{t}\right)} \tag{98}$$

with probability at least $1-\delta$. $\qquad\qquad\qquad\qquad\qquad\qquad\qquad\qquad\qquad\qquad\square$

**Quadratic version of MS-KMM.** The convex optimization in (56) is a quadratic problem since the squared norm in $\mathcal{H}_x$ can be written as

$$\sum_{s=1}^{S}\left\|\frac{1}{t}\sum_{i=1}^{t}\alpha_s^{(i)}K_x(x_{n+i}) - \frac{1}{n_s}\sum_{i=1}^{n_s}\beta_s^{(i)}K_x(x_{s,i})\right\|_{\mathcal{H}_x}^2$$

$$=\sum_{s=1}^{S}\left(\frac{1}{t^2}\sum_{i,j=1}^{t}\alpha_s^{(i)}\alpha_s^{(j)}k_x(x_{n+i},x_{n+j}) + \frac{1}{n_s^2}\sum_{i,j=1}^{n_s}\beta_s^{(i)}\beta_s^{(j)}k_x(x_{s,i},x_{s,j}) - \frac{2}{n_s t}\sum_{i=1}^{t}\sum_{j=1}^{n_s}\alpha_s^{(i)}\beta_s^{(j)}k_x(x_{n+i},x_{s,j})\right)$$

$$=\sum_{s=1}^{S}\frac{\boldsymbol{\alpha}_s^{\mathrm{T}}}{t}\left(\begin{bmatrix}k_x(x_{n+1},x_{n+1}) & \cdots & k_x(x_{n+1},x_{n+t})\\ \vdots & \ddots & \vdots\\ k_x(x_{n+t},x_{n+1}) & \cdots & k_x(x_{n+t},x_{n+t})\end{bmatrix}\frac{\boldsymbol{\alpha}_s}{t} + \frac{\boldsymbol{\beta}_s^{\mathrm{T}}}{n_s}\begin{bmatrix}k_x(x_{s,1},x_{s,1}) & \cdots & k_x(x_{s,1},x_{s,n_s})\\ \vdots & \ddots & \vdots\\ k_x(x_{s,n_s},x_{s,1}) & \cdots & k_x(x_{s,n_s},x_{s,n_s})\end{bmatrix}\frac{\boldsymbol{\beta}_s}{n_s}\right.$$

$$\left.-2\frac{\boldsymbol{\beta}_s^{\mathrm{T}}}{n_s}\begin{bmatrix}k_x(x_{s,1},x_{n+1}) & \cdots & k_x(x_{s,1},x_{n+t})\\ \vdots & \ddots & \vdots\\ k_x(x_{s,n_s},x_{n+1}) & \cdots & k_x(x_{s,n_s},x_{n+t})\end{bmatrix}\frac{\boldsymbol{\alpha}_s}{t}\right)$$

$$=\left[\boldsymbol{\beta}_1^{\mathrm{T}}/n_1, -\boldsymbol{\alpha}_1^{\mathrm{T}}/t, \ldots, \boldsymbol{\beta}_S^{\mathrm{T}}/n_S, -\boldsymbol{\alpha}_S^{\mathrm{T}}/t\right]\mathbf{K}_x\begin{bmatrix}\boldsymbol{\beta}_1/n_1\\ -\boldsymbol{\alpha}_1/t\\ \vdots\\ \boldsymbol{\beta}_S^{\mathrm{T}}/n_S\\ -\boldsymbol{\alpha}_S^{\mathrm{T}}/t\end{bmatrix}$$

where $\mathbf{K}_x$ is the kernel matrix.

Therefore, the optimization problem (56) is equivalent to the quadratic optimization problem

$$\min_{\{\boldsymbol{\beta}_s,\boldsymbol{\alpha}_s\}_{s=1}^{S}}\left[\boldsymbol{\beta}_1^{\mathrm{T}}/n_1, -\boldsymbol{\alpha}_1^{\mathrm{T}}/t, \ldots, \boldsymbol{\beta}_S^{\mathrm{T}}/n_S, -\boldsymbol{\alpha}_S^{\mathrm{T}}/t\right]\mathbf{K}_x\begin{bmatrix}\boldsymbol{\beta}_1/n_1\\ -\boldsymbol{\alpha}_1/t\\ \vdots\\ \boldsymbol{\beta}_S^{\mathrm{T}}/n_S\\ -\boldsymbol{\alpha}_S^{\mathrm{T}}/t\end{bmatrix}$$

$$\text{s.t.}\quad \mathbf{0}\preceq\boldsymbol{\beta}_s\preceq(B_s/\sqrt{D})\mathbf{1}, \quad \mathbf{0}\preceq\boldsymbol{\alpha}_s\preceq\mathbf{1}, \text{ for } s\in[S]$$

$$\left|\boldsymbol{\beta}_s^{\mathrm{T}}\mathbf{1}/n_s - \boldsymbol{\alpha}_s^{\mathrm{T}}\mathbf{1}/t\right|\leq\epsilon, \text{ for } s\in[S] \tag{99}$$

$$\left|\sum_{s=1}^{S}\alpha_s^{(i)}-1\right|\leq\left(1-\frac{1}{\sqrt{D}}\right), \text{ for } i\in[t].$$

## F Implementation details and additional experimental details

This appendix details the datasets and settings used for the experiments in Section 6.

**Hyperparameters.** In principle, cross-validation can be used to determine both hyperparameters $\{\boldsymbol{\lambda_s}\}_{s\in[S]}$ ($\boldsymbol{\lambda}$ for single-source) and $D$. However, it is important to note that standard cross-validation is not applicable when dealing with covariate shift (Sugiyama et al., 2007). We hence avoid cross-validation and determine both parameters as done in (Segovia-Martín et al., 2023). We select the value of $D$ to achieve the lowest minimax risk $R(\mathcal{U})$, defined as the optimal value of (34) ((13) for single-source), over a certain range $D \geq 1$. The second set of hyperparameters $\{\boldsymbol{\lambda}_s\}_{s=1}^{S}$ are determined solving

$$
\begin{aligned}
\min_{\mathrm{p},\boldsymbol{\lambda}_s} \quad & \mathbf{1}^{\mathrm{T}}\boldsymbol{\lambda}_s \\
\text{s.t.} \quad & \boldsymbol{\tau}_s - \boldsymbol{\lambda}_s \preceq \sum_{i=1}^{t}\sum_{y\in\mathcal{Y}}\mathrm{p}(y|x_{n+i})\Phi_{\alpha_s}(x_{n+i},y) \preceq \boldsymbol{\tau}_s + \boldsymbol{\lambda}_s \\
& \boldsymbol{\lambda}_s, \mathbf{p} \succeq \mathbf{0} \\
& \sum_{y\in\mathcal{Y}}\mathrm{p}(y|x_{n+i}) = 1/t \text{ for } i = 1,\dots,t
\end{aligned}
\tag{100}
$$

for $s \in [S]$ ($S = 1$ for single-source), that ensures the uncertainty set considered is non-empty.

**Additional synthetic experimental details.** For the synthetic experiments presented in Section 6.1.1 we utilize linear feature mapping $\Psi(x) = [1, x^{\mathrm{T}}]$ and implement the method referred to as no adapt. using 0-1-loss as shown in (Mazuelas et al., 2023), and reweighted and DW method using true marginal probabilities and 0-1-loss.

For the synthetic experiments presented in Section 6.2.1 we utilize linear feature mapping $\Psi(x) = x$ and implement the existing DW-GCS method estimating weights using 0-1-loss and double-weighting kernel mean matching (DW-KMM) as shown in (Segovia-Martín et al., 2023), 2SW-MDA method estimating weights using KMM as shown in (Huang et al., 2006), and the proposed DW-MSCS method using 0-1-loss and estimating weights using (56).

**Additional real datasets experimental details.** For the experiments in Section 6.1.2, we have considered 7 binary and multiclass classification datasets: "Adult", "Diabetes", "Mammographic", "Usenet2", "Credit", available at the UCI repository (Dua & Graff, 2017), "20 Newsgroups", available at `http://qwone.com/~jason/20Newsgroups/`,and "Redwine", available at (Cortez et al., 2009). For the experiments using "20 Newsgroups" dataset, we utilize the 300 features with highest Pearson's correlation. We considered linear feature mappings $\Psi(x) = [1, x^{\mathrm{T}}]$ and 0-1-loss.

Table 4 details the characteristics of the datasets used in the experiments of Section 6.1.2. The table also shows the number of training and testing samples selected for each of the experiments.

For the experiments in Section 6.2.2, we have considered three multi-source classification datasets: "20 Newsgroups", available at `http://qwone.com/~jason/20Newsgroups/`, "Sentiment Analysis", available at `https://www.cs.jhu.edu/~mdredze/datasets/sentiment/`, and "Spam detection", available at `http://www.ecmlpkdd2006.org/challenge.html`. These datasets have been previously used in single-source and multi-source covariate shift papers (Mansour et al., 2008; Ben-David et al., 2010; Sun et al., 2011; Duan et al., 2012; Sun et al., 2015; Wang et al., 2023). We have also considered two image classification datasets: "DomainNet", available at `https://ai.bu.edu/M3SDA/`, (Peng et al., 2019), and "Office-31", available at `https://github.com/jindongwang/transferlearning/blob/master/data/dataset.md`. For the experiments using "Spam detection" dataset, we utilize multiple number of features with highest Pearson's correlation, and randomly sample 200 training samples from each source and 200 testing samples in each repetition. For the experiments using "Sentiment" dataset, the features were defined as the set of unigrams that appear five times or more in all domains. We used a binary feature vector encoding the presence of those unigrams, as done in (Mansour et al., 2008). We randomly sample 1,000 training samples from each source and 150 testing samples in each repetition. For the experiments using "DomainNet" dataset, we reduce the number of classes to 4, since the original dataset has 256 different classes. The goal is to predict

Table 4: Datasets used in the experiments of Section 6.1.2.

| Dataset | Type of Shift | Covariates | Samples | Samples Selected training | testing |
|---|---|---|---|---|---|
| Adult | tweak-one | 14 | 48842 | 500 | 500 |
| | knock-out | | | 330 | 500 |
| Diabetes | tweak-one | 8 | 768 | 100 | 100 |
| | knock-out | | | 94 | 192 |
| Mammographic | tweak-one | 99 | 1500 | 100 | 100 |
| | knock-out | | | 151 | 276 |
| Usenet2 | tweak-one | 99 | 1500 | 250 | 250 |
| | knock-out | | | 179 | 360 |
| Credit | tweak-one | 15 | 690 | 100 | 100 |
| | knock-out | | | 112 | 198 |
| 20 Newsgroups | tweak-one | 500 | 9016    6017 | 300 | 300 |
| | knock-out | | | 330 | 400 |
| | dirichlet | | | 300 | 300 |
| Redwine | tweak-one | 15 | 690 | 100 | 100 |
| | knock-out | | | 101 | 189 |
| | dirichlet | | | 100 | 100 |

Table 5: Summary of sources using "20Newsgroups" dataset.

| Sources | comp | rec | sci | talk |
|---|---|---|---|---|
| Source 1 | comp.graphics | rec.autos | sci.crypt | talk.politics.guns |
| Source 2 | comp.os.ms-windows.misc | rec.motorcycles | sci.electronics | talk.politics.mideast |
| Source 3 | comp.sys.ibm.pc.hardware | rec.sport.baseball | sci.med | talk.politics.misc |
| Source 4 | comp.sys.mac.hardware | rec.sport.hockey | sci.space | talk.religion.misc |

if the image is an airplane, bus, ambulance or police car. We first use a ResNet-18 to obtain features of dimension 512, and randomly sample 100 training samples from each source and 200 testing samples in each repetition. For the experiments using "Office-31" dataset, we first use a ResNet-50 to obtain features of dimension 2048. We have considered different classification problems, selecting five labels depending of the type of the object. For electronics, the goal is to predict if the image is a laptop, monitor, mouse, keyboard or speaker. For stationery, the goal is to predict if the image is a pen, paper notebook, ruler, scissors or eraser. For organization, the goal is to predict if the image is a file cabinet, ring binder, letter tray, clipboard or trash can. For mixed, the goal is to predict if the image is a backpack, mug, monitor, trash can or scissors. We have utilize the 100 features with highest Pearson's correlation, and randomly sample half of the samples from each domain as training samples, and half of the samples from each domain, ensuring that the same number of samples from each domain, as the testing set in each repetition.

Table 5 details the procedure followed for the creation of the different sources in the "20Newsgroups" dataset. For this experiments, we first generated four sources as shown in Table 5. Then, we generated three training sources subsampling 100 samples from one of the sources and 50 samples from other two sources. For the testing set, we subsampled 40 samples from the three sources from which we sampled 100 samples at training and 30 more samples from the fourth source.

Table 6 details the characteristics of the datasets used in the experiments. The table also shows the parameter $\sigma$ used in the computation of the kernel matrix $\mathbf{K}$ for the KMM, DW-GCS, 2SW-MDA, CW KMM and DW-MSCS methods, which is determined using the common heuristic based on nearest neighbors with $K = 50$. For these experiments, we have considered 0-1-loss for the methods based on double-weighting.

For the experiments in Section 6.3 we also considered "Sentiment analysis" dataset, described in Table 6, defining the features as for multi-source covariate shift. We considered linear feature mappings of the form

Table 6: Datasets used in the experiments of Section 6.2.2.

| Dataset | Sources | Covariates | Samples | | Ratio of majority class | $\sigma$ |
|---|---|---|---|---|---|---|
| **Spam detection** | | | | | | |
| 500 features | 3 | 500 | 200/200/200 | 200 | 0.5000 | 12.6828 |
| 1000 features | 3 | 1000 | 200/200/200 | 200 | 0.5000 | 15.3386 |
| 1500 features | 3 | 1500 | 200/200/200 | 200 | 0.5000 | 16.9269 |
| 2000 features | 3 | 2000 | 200/200/200 | 200 | 0.5000 | 17.7283 |
| 2500 features | 3 | 2500 | 200/200/200 | 200 | 0.5000 | 19.1219 |
| 3000 features | 3 | 3000 | 200/200/200 | 200 | 0.5000 | 21.5424 |
| **20 Newsgroups** | | | | | | |
| comp vs rec | 3 | 1,000 | ≈1,174/1,174/1,174 | 3,143 | 0.4937 | 18.7058 |
| comp vs sci | 3 | 1,000 | ≈1,172/1,172/1,172 | 3,128 | 0.4952 | 18.6501 |
| comp vs talk | 3 | 1,000 | ≈1,066/1,066/1,066 | 2,856 | 0.5435 | 20.4118 |
| rec vs sci | 3 | 1,000 | ≈1,188/1,188/1,188 | 3,161 | 0.5015 | 19.1100 |
| rec vs talk | 3 | 1,000 | ≈1,082/1,082/1,082 | 2,889 | 0.5497 | 21.2683 |
| sci vs talk | 3 | 1,000 | ≈1,080/1,080/1,080 | 2,874 | 0.5483 | 20.8485 |
| comp vs rec vs sci | 3 | 1,000 | ≈1,767/1,767/1,767 | 4,716 | 0.3368 | 37.1962 |
| comp vs rec vs talk | 3 | 1,000 | ≈1,661/1,661/1,661 | 4,444 | 0.3579 | 37.6344 |
| comp vs sci vs talk | 3 | 1,000 | ≈1,659/1,659/1,659 | 4,429 | 0.3565 | 37.5618 |
| rec vs sci vs talk | 3 | 1,000 | ≈1,675/1,675/1,675 | 4,462 | 0.3555 | 38.0282 |
| **Sentiment** | | | | | | |
| All domains | 4 | 3,034 | 2,000/2,000/2,000/2,000 | 8,000 | 0.5000 | 10.3441 |
| books | 3 | 3,034 | 2,000/2,000/2,000 | 8,000 | 0.5000 | 10.3441 |
| dvd | 3 | 3,034 | 2,000/2,000/2,000 | 8,000 | 0.5000 | 10.3441 |
| electronics | 3 | 3,034 | 2,000/2,000/2,000 | 8,000 | 0.5000 | 10.3441 |
| kitchen | 3 | 3,034 | 2,000/2,000/2,000 | 8,000 | 0.5000 | 10.3441 |
| **DomainNet** | | | | | | |
| clipart | 5 | 512 | 385/216/258/698/1900 | 600 | 0.2931 | 21.8061 |
| infograph | 5 | 512 | 2226/216/258/698/1900 | 600 | 0.3028 | 21.5041 |
| painting | 5 | 512 | 2226/385/258/698/1900 | 600 | 0.2891 | 21.5743 |
| quickdraw | 5 | 512 | 2226/385/216/698/1900 | 600 | 0.2984 | 21.5364 |
| real | 5 | 512 | 2226/385/216/258/1900 | 600 | 0.2994 | 21.4893 |
| sketch | 5 | 512 | 2226/385/216/258/698 | 600 | 0.3171 | 24.0266 |
| **Office-31** | | | | | | |
| electronics | 2 | 100 | 249/80 | 141 | 0.2181 | 8.6251 |
| stationery | 2 | 100 | 232/59 | 90 | 0.2234 | 8.7266 |
| organization | 2 | 100 | 214/65 | 102 | 0.2240 | 9.1813 |
| mixed | 2 | 100 | 225/73 | 111 | 0.2451 | 8.5133 |

$\Psi(x) = [1, x^{\mathrm{T}}]$ and implemented existing reweighted methods using log-loss, and DW-MSLS using 0-1-loss. **Additional experiments regarding the hyperparameters.** In the additional experiments we have studied the effectiveness of the proposed selection method for hyperparameters $D$ and $\boldsymbol{\lambda}$.

The hyperparameter $\boldsymbol{\lambda}$ controls the confidence with which the true underlying distribution is contained in the uncertainty set $\mathcal{U}$. Table 7 shows the mean error and the standard deviation of the different experiments performed to see the impact of the hyperparameter $\boldsymbol{\lambda}$ in label shift scenarios. The proposed value of $\boldsymbol{\lambda} = \boldsymbol{\lambda}_p$ is obtained by solving the optimization problem in (100). The rest of the values of the hyperparameter $\boldsymbol{\lambda}$ considered are of the form $\boldsymbol{\lambda} = \lambda_0 \mathbf{1}$ with $\lambda_0 \in \{0, 2^{-8}, 2^{-6}, 2^{-4}, 2^{-2}, 2^{-1}, 1\}$. Table 7 shows how the proposed method for selecting $\boldsymbol{\lambda}$ results in performances near those obtained with the best values of $\boldsymbol{\lambda}$. As we can see from the table, properly tuning the hyperparameter $\boldsymbol{\lambda}$ is crucial, with small values of being preferred in order to achieve small classification errors.

The hyperparameter $D$ controls the trade-off related to the set of weights $\{\beta_s(x, y)\}_{s \in [S]}$, $\{\alpha_s(x, y)\}_{s \in [S]}$. As presented in Section 4.4, $\|\beta_s(x, y)\|_\infty \leq B/\sqrt{D}$, where $B = \sup_{x \in \mathcal{X}, y \in \mathcal{Y}} \mathrm{p_{te}}(x, y) / \sum_{s=1}^{S} \mathrm{p}_s(x, y)$, and

Table 7: Classification error in 6 scenarios using DW-LS methods varying the value of the hyperparameter $\boldsymbol{\lambda}$. The bold values represent the lowest classification error in each scenario.

| Dataset | $\boldsymbol{\lambda}_p$ | $\lambda_0 = 2^{-8}$ | $\lambda_0 = 2^{-6}$ | $\lambda_0 = 2^{-4}$ | $\lambda_0 = 2^{-2}$ | $\lambda_0 = 2^{-1}$ | $\lambda_0 = 1$ |
|---|---|---|---|---|---|---|---|
| **Redwine** | | | | | | | |
| knock-out | $\mathbf{.52 \pm .09}$ | $\mathbf{.52 \pm .09}$ | $.53 \pm .09$ | $.54 \pm .10$ | $.55 \pm .09$ | $.57 \pm .07$ | $.64 \pm .07$ |
| tweak-one | $.49 \pm .24$ | $.47 \pm .24$ | $\mathbf{.44 \pm .24}$ | $.46 \pm .31$ | $.53 \pm .27$ | $.59 \pm .21$ | $.65 \pm .20$ |
| dirichlet | $.53 \pm .11$ | $\mathbf{.52 \pm .11}$ | $.50 \pm .10$ | $.52 \pm .10$ | $.62 \pm .09$ | $.62 \pm .08$ | $.66 \pm .03$ |
| **20news** | | | | | | | |
| knock-out | $.64 \pm .04$ | $.62 \pm .02$ | $.64 \pm .03$ | $.69 \pm .04$ | $.71 \pm .02$ | $.73 \pm .02$ | $.74 \pm .02$ |
| tweak-one | $.61 \pm .08$ | $\mathbf{.58 \pm .04}$ | $.59 \pm .05$ | $.59 \pm .10$ | $.61 \pm .06$ | $.68 \pm .07$ | $.70 \pm .07$ |
| dirichlet | $.65 \pm .02$ | $.65 \pm .02$ | $\mathbf{.63 \pm .02}$ | $.65 \pm .04$ | $.72 \pm .03$ | $.72 \pm .03$ | $.74 \pm .02$ |

$\min_{x \in \mathcal{X}, y \in \mathcal{Y}} \sum_{s=1}^{S} \alpha_s(x,y) / \max_{x \in \mathcal{X}, y \in \mathcal{Y}} \sum_{s=1}^{S} \alpha_s(x,y) = 1/\sqrt{D}$. Considering small values of $D$ results in poor expectation estimate, since the expectation estimate is bounded by $\|\beta_s(x,y)\|_{\infty} \leq B/\sqrt{D}$, as discussed in Section 4.4. On the other hand, considering small values of $D$ results in classification rules with significant confidence in a large region. If $D = 1$, then the weight functions $\alpha_s(x) = 1$ for all $s \in S$. This way, the double-weighting approach is simplified to the standard reweighted approach, and we equally combine the training sources to obtain the classification rules. Figure 5 and Figure 6 show the different values of the sum of the weights alpha, $\sum_{s=1}^{S} \alpha_s(x)$, when we consider different values of the hyperparameter $D$ for the synthetic experiments corresponding to Section 6.1 of the paper. In the figures we can see how the sum $\sum_{s=1}^{S} \alpha_s(x)$ decreases when we increase the value of the hyperparameter $D$. In addition, Figure 7 shows how the value of the weights $\{\beta_s(x)\}_{s \in [S]}$ is also reduced when we increase the value of the hyperparameter $D$. As we can see in the figure, we avoid large weights $\beta_s(x)$ by reducing the corresponding value $\alpha_s(x)$.

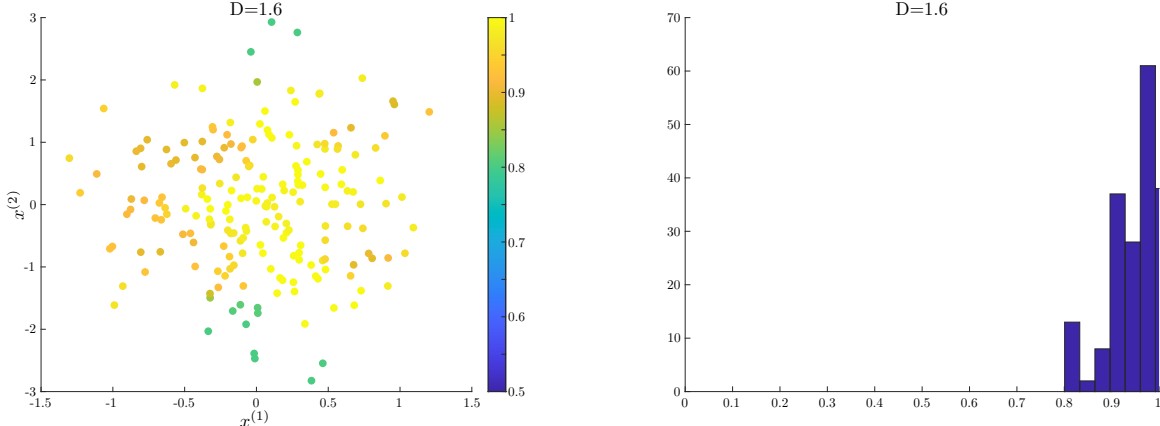

Figure 5: Visualization of $\sum_{s=1}^{S} \alpha_s(x)$ for hyperparameter $D = 1.6$. Left: Plot of the testing instances on $\mathbb{R}^2$ together with the colorbar representing the sum values the values of $\sum_{s=1}^{S} \alpha_s(x)$. Right: Histogram of the values of $\sum_{s=1}^{S} \alpha_s(x)$ for the testing instances.

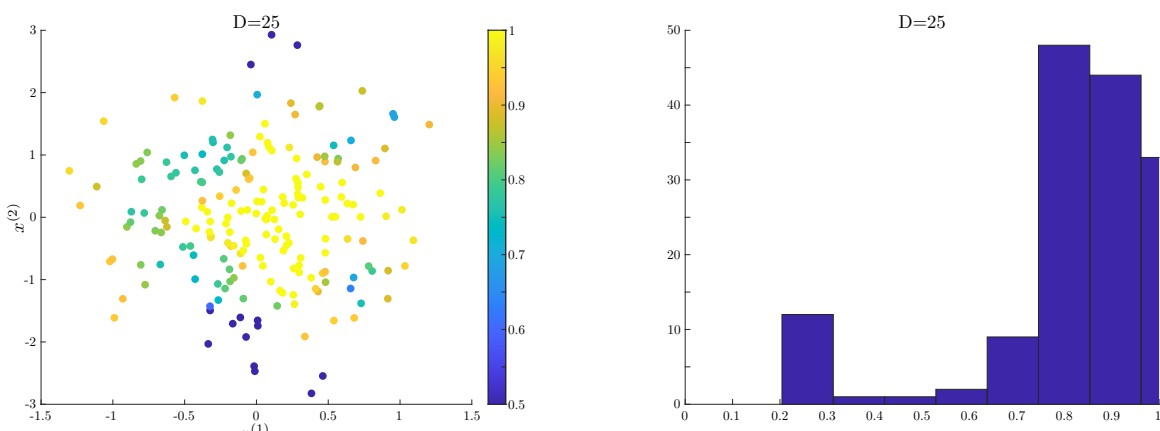

Figure 6: Visualization of $\sum_{s=1}^{S} \alpha_s(x)$ for hyperparameter $D = 25$. Left: Plot of the testing instances on $\mathbb{R}^2$ together with the colorbar representing the sum values the values of $\sum_{s=1}^{S} \alpha_s(x)$. Right: Histogram of the values of $\sum_{s=1}^{S} \alpha_s(x)$ for the testing instances.

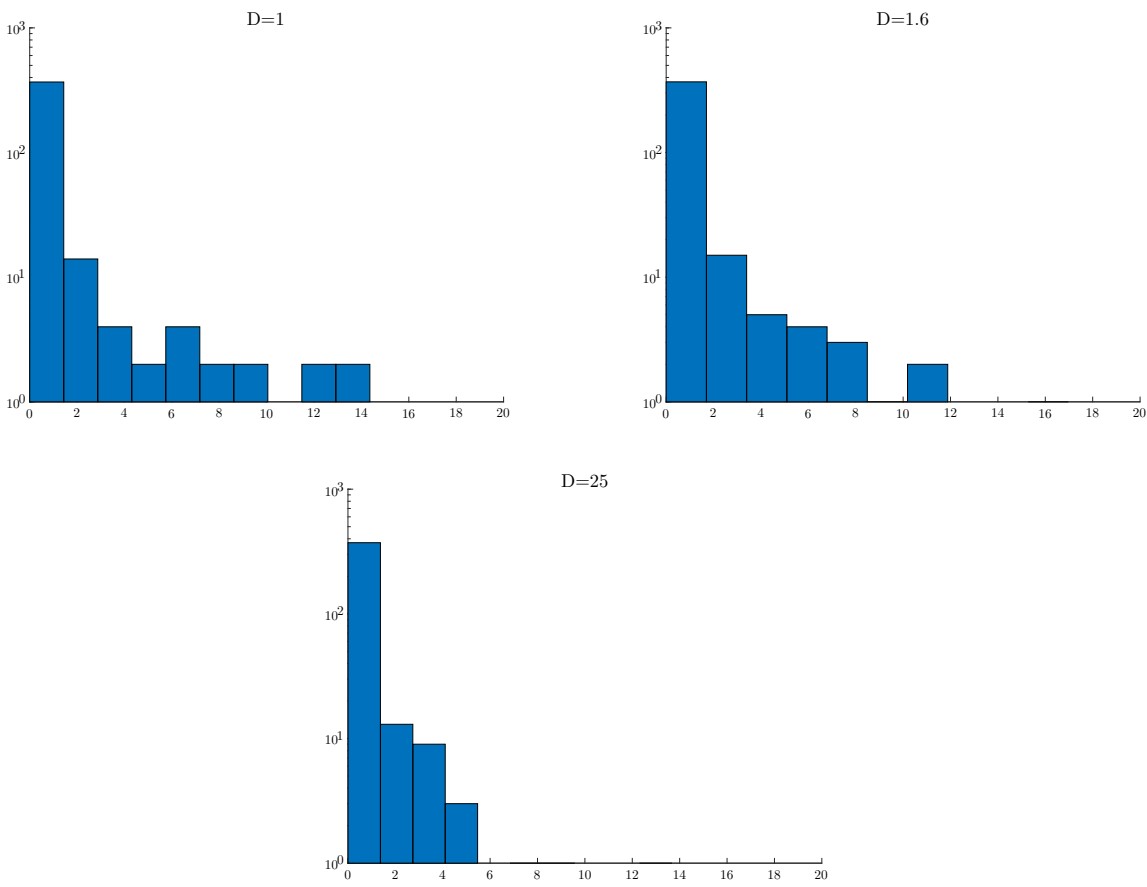

Figure 7: Histogram of $\{\beta_s(x)\}_{s=1}^{S}$ for hyperparameter $D \in \{1, 1.6, 25\}$. Note that for $D = 1$ the weights correspond to those of the reweighted methods.

