# OpenReview forum: "A Unified View of Double-Weighting for Marginal Distribution Shift"
_TMLR — Accepted by TMLR_

### Review · Reviewer_B8SF · 2024-11-06

**Summary Of Contributions:**

This paper provides a weighted general framework for the problem of distribution shift adaptation by applying a dual-weighting technique, which assigns weights to both training and testing samples. This idea is interesting and insightful.

**Audience:**

Yes

**Claims And Evidence:**

No

**Requested Changes:**

See Weaknesses.

**Strengths And Weaknesses:**

**Strengths:**

1. The paper has solid theoretical support.
2. The structure and organization of the paper are well done.
3. The paper considers a relatively comprehensive range of scenarios.

**Weaknesses:**

1. Can this method address the case of conditional shift?
2. The paper seems to be an extension of the ICML 2023 paper; could you further emphasize the contributions and differences? It would be helpful to provide a detailed comparison in the appendix.
3. Domain adaptation methods involve more than just weighting. Could you compare this approach with some recent SOTA methods?
4. Is it possible to adapt this method to deep learning models?
5. Could you conduct experimental comparisons of the proposed method using image datasets, such as DomainNet?
6. Could you include more experimental analyses, such as the impact of hyperparameters, or provide visualization-based analysis?

---

> ### Author Response · Authors · 2024-11-15
>
> We thank the reviewer for the useful suggestions to clarify the technical details presented and improve the paper’s experimental results. We are currently conducting experiments using DomainNet dataset. We will soon provide the revised manuscript together with point to point responses to the Reviewer's suggestions and comments.

---

> ### Author Response · Authors · 2025-01-06
>
> We thank the reviewer for the useful suggestions to clarify the technical details presented and improve the paper’s experimental results. We answer your questions/comments point by point as follows.
>
> **Addressing the case of conditional shift.**
> The proposed general methodology presented for single-source and multi-source distribution shift adaptation can address the case of conditional shift. Notice that from equation (10) in the case of instance conditional shift, where $\mathrm{p}\_{\text{te}}(y)=\mathrm{p}\_{\text{tr}}(y)$  we have that the reference weights become
> $$\beta(x,y)=\min\left(\frac{\mathrm{p}\_{\text{te}}(x|y)}{\mathrm{p}\_{\text{tr}}(x|y)},C\right),\text{ }\alpha(x,y)=\min\left(\frac{\mathrm{p}\_{\text{te}}(x|y)}{\mathrm{p}\_{\text{tr}}(x|y)},C\right).$$
> As described at the end of Section 4 of the paper, obtaining weights is more complicated in cases where the distribution shift involves both labels and covariates. For instance, for conditional shift scenarios, the estimation of ratios of conditional distributions is more complex than marginal distribution shift scenarios, since assuming that $n$ training samples $(x_1,y_1),(x_2,y_2),\ldots,(x_n,y_n)$, and $t$ testing instances $x_{n+1},x_{n+2},\ldots,x_{n+t}$ are available at learning is not enough to estimate both $\alpha(x,y)$ and $\beta(x,y)$ for correcting instance conditional shift.
> If we assume that a set of $t$ testing samples $(x_{n+1},y_{n+1}),(x_{n+2},y_{n+2}),\ldots,(x_{n+t},y_{n+t})$ from distribution $\mathrm{p}_{\text{te}}(x,y)$ is available at learning, we can consider any method for estimate the ratios and use the double-weighting framework considering the reference solutions above.
> Even though the double weighting framework is general enough to deal with any shift, estimating conditional shift weights usually requires a different set of methods in order to estimate the conditional relations between instances and labels, that, in practice, would require multiple assumptions regarding the distributions. So we focus only on marginal shifts in this paper, but we add a discussion at the end of Section 4 in the updated paper.
>
> **Comparison with the ICML 2023 paper.**
> We thank the reviewer for the helpful suggestions about a detailed comparison in the appendix.
> The main difference with respect to the ICML 2023 paper is that 1) such paper addresses the problem of single-source covariate shift, whereas the present work tackles a broader range of distribution shifts, including general shifts and multi-source shifts, using a unified approach; 2) the submitted manuscript introduces new algorithms designed for label shift, multi-source covariate shift, and multi-source label shift, to deal with unique challenges in these cases, which is non-trivial.  We refer to the general response and the updated paper for more detailed discussions.
> The experiments using synthetic and real-world datasets demonstrate that the proposed techniques achieve a significant performance improvement in multiple distribution shift scenarios. Especially, our results show that in cases of significant shifts between training and testing distributions, most existing techniques result in a negative transfer of information among sources, while the proposed approach achieves improved performance.  Following the reviewer's suggestion, we provide additional comparison with the existing double-weighting paper in Section 2.3 and Appendix B.

---

> ### Author Response · Authors · 2025-01-06
>
> **Comparison with domain adaptation methods.**
> We thank the reviewer for pointing out the need to further discuss the domain adaptation methods.
>
> First we want to clarify the relation between domain adaptation and distribution shift. It is true that they are closely related, but the focus of the research may differ. In general, we can say covariate shift or label shift adaptation are special cases of domain adaptation. But usually, when people propose methods for domain adaptation, there is no specific distributional assumptions, instead the research is more on how to align representations between domains or smartly utilizing unlabeled data. It is also due to the fact that there is no specific distributional assumptions, meaningful theoretical guarantees are harder to obtain.
>
> In this paper, we focus on specific distributional assumptions and how they can help us do weighting smartly, which is complementary to many domain adaptation techniques. For example, the proposed method is flexible enough to accommodate deep neural networks (our new experiments) and representation learning frameworks. The weighting process can be carried out not only in the original feature space but also within the learned representation spaces, making it compatible with modern domain adaptation paradigms.
>
> In the updated paper, at the end of the Section 5, we have discussed other domain adaptation methods focusing on representation learning and talked about how we can be compatible.
>
> **Adaptability of the method to deep learning models.**
> The proposed methodology can be adapted to utilize neural networks, for instance, by defining the feature mapping $\Phi(x,y)$ in terms of the output of the penultimate layer $g(\cdot,\theta)$, i.e., $\Phi(x,y)=e_y\otimes g(x,\theta)$, where $\theta$ is the parameter of the network. By feeding the data through the network, the outputs of the penultimate layer provide a high-level representation of the input data, capturing complex patterns and relationships learned during the network training. This way, we can deal with shifts in the semantic representation space. We have used pretrained ResNets to map images into feature vectors in the new numerical results, as detailed in the general responses. The results obtained show how the proposed methodology can accommodate deep learning frameworks, obtaining good performances dealing with multi-source covariate shift.

---

> ### Author Response · Authors · 2025-01-06
>
> **Experiments using image datasets.**
> We provide additional results for the requested experiments as described in the general comments, included in Section 6.2 of the updated paper. We consider a small scale image dataset, Office-31, and a large scale dataset, DomainNet. Table 2 in the paper shows the average classification error across both datasets and scenarios of multi-source covariate shift, along with the standard deviations. Our main experimental takeaway is that the proposed method is more effective than existing methods for adaptation, especially when the testing set contains samples from an unseen domain for the training sources.
>
> **Impact of hyperparameters, and visualization-based analysis**
> The main hyperparameters we need to choose are the confidence vector $\boldsymbol{\lambda}$ and the hyperparameter $D$ related with the values of the set of weights $\lbrace\beta_s(x,y)\rbrace_{s\in[S]}$, $\lbrace\alpha_s(x,y)\rbrace_{s\in[S]}$.
> In the revised version of the paper, in Appendix F, we have included a new experimental discussion, talking about the impact of the hyperparameter $\boldsymbol{\lambda}$ in label shift scenarios. In addition, in Section 6.2.1, we have incorporated a visualization-based analysis to describe the impact of the hyperparameter $D$, that controls the values of the weights $\lbrace\beta_s(x,y)\rbrace_{s\in[S]}$ and $\lbrace\beta_s(x,y)\rbrace_{s\in[S]}$, for multi-source covariate shift adaptation.
>
> The hyperparameter $\boldsymbol{\lambda}$ controls the confidence with which the true underlying distribution is contained in the uncertainty set $\mathcal{U}$.
> The table below shows the mean error and the standard deviation of the different experiments performed to see the impact of the hyperparameter $\boldsymbol{\lambda}$ in label shift scenarios. The proposed value of $\boldsymbol{\lambda}=\boldsymbol{\lambda}_{p}$ is obtained by solving the optimization problem in (96). The rest of the values of the hyperparameter $\boldsymbol{\lambda}$ considered are of the form $\boldsymbol{\lambda}=\lambda_0\boldsymbol{1}$ with $\lambda_0\in\lbrace0,2^{-8},2^{-6},2^{-4},2^{-2},2^{-1},1\rbrace$. In the table, we can see how the
> proposed method for selecting $\boldsymbol{\lambda}$ results in performance near those obtained with the best values of $\boldsymbol{\lambda}$.
> As we can see from the table, proper tuning of the hyperparameter $\boldsymbol{\lambda}$ is crucial, with small values preferred to achieve small classification errors.
>
> | Dataset | $\boldsymbol{\lambda}_p$ | $\lambda_0=2^{-8}$       | $\lambda_0=2^{-6}$       | $\lambda_0=2^{-4}$ | $\lambda_0=2^{-2}$ | $\lambda_0=2^{-1}$ | $\lambda_0=1$ |
> |-----------------|--------------------------|--------------------------|--------------------------|--------------------|--------------------|--------------------|---------------|
> | **Redwine** | | | | | | | |
> | knock-out | $\boldsymbol{.52\pm.09}$ | $\boldsymbol{.52\pm.09}$ | $.53\pm.09$ | $.54\pm.10$ | $.55\pm.09$ | $.57\pm.07$ | $.64\pm.07$ |
> | tweak-one | $.49\pm.24$ | $.47\pm.24$ | $\boldsymbol{.44\pm.24}$ | $.46\pm.31$ | $.53\pm.27$ | $.59\pm.21$ | $.65\pm.20$ |
> | dirichlet | $.53\pm.11$ | $\boldsymbol{.52\pm.11}$ | $.50\pm.10$ | $.52\pm.10$ | $.62\pm.09$ | $.62\pm.08$ | $.66\pm.03$ |
> | **20news** | | | | | | | |
> | knock-out | $.64\pm.04$ | $\boldsymbol{.62\pm.02}$ | $.64\pm.03$ | $.69\pm.04$ | $.71\pm.02$ | $.73\pm.02$ | $.74\pm.02$ |
> | tweak-one | $.61\pm.08$ | $\boldsymbol{.58\pm.04}$ | $.59\pm.05$ | $.59\pm.10$ | $.61\pm.06$ | $.68\pm.07$ | $.70\pm.07$ |
> | dirichlet | $.65\pm.02$ | $.65\pm.02$ | $\boldsymbol{.63\pm.02}$ | $.65\pm.04$ | $.72\pm.03$ | $.72\pm.03$ | $.74\pm.02$ |

---

> ### Author Response · Authors · 2025-01-06
>
> The hyperparameter $D$ controls the trade-off related to the set of weights $\lbrace\beta\_s(x,y)\rbrace\_{s\in[S]}$, $\lbrace\alpha\_s(x,y)\rbrace\_{s\in[S]}$. As presented in Section 4.4, $\|\|\beta_s(x,y)\|\|\_{\infty}\leq B/\sqrt{D}$, where $$B=\sup\_{x\in\mathcal{X},y\in\mathcal{Y}}\frac{\mathrm{p}\_{\text{te}}(x,y)}{\sum\_{s=1}^S\mathrm{p}\_{s}(x,y)},$$ and $$\frac{\min\_{x\in\mathcal{X},y\in\mathcal{Y}}\sum\_{s=1}^S\alpha\_s(x,y)}{\max\_{x\in\mathcal{X},y\in\mathcal{Y}}\sum\_{s=1}^S\alpha\_s(x,y)}=1/\sqrt{D}.$$ Considering small values of $D$ results in a poor expectation estimate, since the expectation estimate is bounded by $\|\|\beta\_s(x,y)\|\|\_{\infty}\leq B/\sqrt{D}$, as discussed in Section 4.4. On the other hand, considering small values of $D$ results in classification rules with significant confidence in a large region. If $D=1$, then the weight functions $\alpha_s(x)=1$ for all $s\in S$. This way, the double-weighting approach is simplified to the standard reweighted approach, and we equally combine the training sources to obtain the classification rules.
> Figures 5 and 6 (Appendix F) of the revised version of the paper show the different values of the sum of the weights alpha, $\sum_{s=1}^S\alpha_s(x)$, when we consider different values of the hyperparameter $D$ for the synthetic experiments corresponding to Section 6.2.1 of the paper. In the figures, we can see how the sum $\sum_{s=1}^S\alpha_s(x)$ decreases when we increase the value of the hyperparameter $D$. In addition, Figure 7 of the paper shows how the value of the weights $\lbrace\beta_s(x)\rbrace_{s\in[S]}$ is also reduced when we increase the value of the hyperparameter $D$. As we can see in the figure, we avoid large weights $\beta_s(x)$ by reducing the corresponding value $\alpha_s(x)$.

---

### Review · Reviewer_SA1f · 2024-12-01

**Summary Of Contributions:**

This paper proposes a unified variant of double-weighting approach to tackle distribution shifts between training and testing domains. The proposed paradigm, which is mostly built and inherited from the previous double-weighting method [1], extends the double-weighting method from specifically considering the covariates shift ($\mathrm{p_{tr}}(x), \mathrm{p_{te}}(x)$) to a more general case involving the shift of joint distributions ($\mathrm{p_{tr}}(x, y), \mathrm{p_{te}}(x, y)$). Furthermore, the unified double-weighting is applied to a multi-source scenario, achieving noticeable improvement compared to existing methods.

**Audience:**

Yes

**Broader Impact Concerns:**

N.A.

**Claims And Evidence:**

Yes

**Requested Changes:**

See Strength and Weaknesses for details.

**Strengths And Weaknesses:**

However, to me the technical contribution of this manuscript seems to be rather trivial. For example, Eq. 10 poses one of the major differences between the previous double-weighting [1], which considering the difference between joint distributions of training and testing domains when computing sample weights. However, this modification does not bring significant differences to the following derivation compared to those in [1]. In fact, most derivations from page 5-8 are nearly identical to [1] (e.g., Eq. 11-18 correspond to Eq. 9-16 in [1]). When coming to the newly discussed label shift scenario, it merely replaces the variance shift $\mathrm{p_{tr}}(x), \mathrm{p_{te}}(x)$ of [1] with $\mathrm{p_{tr}}(y), \mathrm{p_{te}}(y)$.

Multi-source double-weighting approach is the other major contributions claimed in this paper, yet similar observations can be found, where the introduction of multiple sources brings limited differences compared to the existing work. Due to the double-blind reviewing protocol, I can’t tell whether this is an extended journal version of [1]. If not, the current version shows limited difference compared to [1] and should therefore be revised majorly.

[1] José I. Segovia-Martín, Santiago Mazuelas, Anqi Liu. Double-Weighting for Covariate Shift Adaptation. ICML 2023

---

> ### Author Response · Authors · 2025-01-06
>
> **Comparison with the ICML 2023 paper.**
> We thank the reviewer for the helpful suggestions to clarify the technical improvements and differences with respect to the double-weighting ICML 2023 paper. We provide additional comparison with the existing double-weighting paper in the general comments.
>
> In particular, the submitted manuscript is not an extension journal version of the paper "Double-Weighting for Covariate Shift Adaptation, ICML 2023". Our paper introduces a novel and unified approach for adapting to general types of distributional shifts, including label shift, multi-source shifts, and general shifts. The unified double weighting view makes independent contributions to the domain, which is out of the scope of the ICML 2023 paper, focusing only on single-source covariate shift. This work also proposes new algorithms for label shift, multi-source covariate shift, and multi-source label shift to deal with unique challenges existing in their specific cases. Experimental results show improved performance in comparison with existing methods, especially in cases of distribution support mismatch, consistent with the theoretical guarantees and generalization bounds presented.
>
>
> More specifically, we want to emphasize our contribution regarding the double-weighting for multi-source distribution shift compared to the existing work:
> Our proposed uncertainty set $\mathcal{U}$ is formed by the $S$ sets of constraints, each of which characterizing weighted feature expectation matching on $\Phi\_{\alpha_s}(x,y)$ in each source domain with weights $\alpha_s(x)$. This set is contained in each of the $S$ uncertainty sets $\lbrace \mathcal{U}\_s \rbrace_{s=1}^S$ and often significantly smaller. Since $\mathcal{U} \subseteq \mathcal{U}\_s$, $\min\_{h}\max\_{p\in\mathcal{U}}\ell(h,p) \leq\min\_{h}\max\_{p\in\mathcal{U}\_s}\ell(h,p)$, and $R(\mathcal{U})\leq R(\mathcal{U}\_s)$ for $s\in[S]$. This means that the minimax risk of the proposed multi-source method is smaller than the minimax risk of the single-source method. This also differentiates us from directly applying (Segovia-Martín et al., 2023) to each source, since it will not lead to our minimax risk. The classification rule obtained from our method contains instance-dependent combinations of the different feature mappings $\lbrace\Phi\_{\alpha\_s}(x,y)\rbrace\_{s=1}^S$, where the set of weights $\lbrace\alpha\_s(x,y)\rbrace\_{s=1}^S$ may be interpreted as the ratio of information learned from each of the sources to classify the instance $x$.
> Besides a smaller minimax risk, the resulting classification rule also has the advantage that we only need one $\alpha_s(x,y)$ from the set $\lbrace\alpha_s(x,y)\rbrace_{s=1}^S$ to be sufficiently large in order to have classification rules with high confidence. By allowing the sources to share information between them, we are able to alleviate the trade-off in (Segovia-Martín et al., 2023), where weights $\alpha_s(x,y)$, for $s\in[S]$, $x\in\mathcal{X}$, need to be sufficiently large in order to have classification rules with high confidence. Then, we can consider more flexible values of $\beta_s(x,y)$, for $s\in[S]$, $x\in\mathcal{X}$, leading to better expectation estimates and better bounds than just applying (Segovia-Martín et al., 2023), where weights are independent for each of the sources.
>
> The generalization bounds presented in Section 4.3 for multi-source distribution shift are non-trivial and different from the ones in Section 3.3 for single-source distribution shift. Specifically, our generalization risk under the true probability distribution $R(h^{\mathcal{U}})$ is bounded by the minimax risk $R(\mathcal{U})$:
> $$R\left(h^{\mathcal{U}}\right)\leq R(\mathcal{U})+\sum\_{s=1}^S\left(\left|\boldsymbol{\tau}\_s-\mathbb{E}\_{p\_{\text{te}}}\Phi\_{\alpha\_s}(x,y)\right|-\boldsymbol{\lambda}\_s\right)^T\left|\boldsymbol{\mu}\_s^*\right|.$$
> On the right-hand-side, we have $R(\mathcal{U})\leq R(\mathcal{U}\_s)$ $\forall s\in[S]$, and, since, with probability $1-\delta$
> $$\|\|\boldsymbol{\tau}\_{s}-\mathbb{E}\_{p\_{\text{te}}}\Phi\_{\alpha\_s}(x,y)\|\|\_{\infty}\leq M\sqrt{\frac{2\|\|\beta\_s(x)\|\|\_{\infty}^2}{n}\log\frac{2}{\delta}}$$
> where $M$ is a constant satisfying that $\|\|\Phi(x,y)\|\|_{\infty}\leq M$ for all $x\in\mathcal{X}$, $y\in\mathcal{Y}$, smaller weights $\beta_s(x)$ lead to better expectation estimates. Therefore, we obtain much better generalization bounds compared to applying (Segovia-Martín et al., 2023) multiple times.  Alternatively, if we use (Segovia-Martín et al., 2023) to solve $S$ optimization problems independently and then take the sum, the resulting generalization bounds are strictly worse.
>
> We understand the concern about the similarity at first glance.
> However, the differences in the details differentiate us significantly from the previous work. We hope our explanation helps resolve the concern and will revise the writing to emphasize the differences.

---

### Review · Reviewer_jqcK · 2024-12-23

**Summary Of Contributions:**

This paper focuses on domain adaptation, an important problem in machine learning. This paper proposes a weighted-based method for domain adaptation in multiple scenarios, including single-source domain, multi-source domain, and label shift scenarios. The main idea is to reweight both training and test samples. Besides, it also considers whole-domain importance in multi-domain scenarios. This paper makes a lot of theory analysis and contributes many meaningful results. Based on these theories, a novel method is proposed. Extensive experiments are conducted in both synthetic and real dataset, and the results show the effectiveness of the proposed method.

**Audience:**

Yes

**Claims And Evidence:**

Yes

**Requested Changes:**

My biggest concerns are about the experiments. 1) The datasets in the experiments are not the common datasets in domain adaptation. So I suggest conducting the experiments on the most common datasets, such as small-scale datasets, Office-31, middle-scale datasets, Office-Home, and large-scale dataset DomainNet. I think selecting one or two datasets is enough to show the effectiveness of the proposed method. 2) The comparison baseline is outdated, and more recent methods published in the past two years are suggested to compare.

**Strengths And Weaknesses:**

1. This paper focuseson  an important problem.
2. This paper proposes a reweighted based method in both training and test samples, which is novel.
3. This paper makes a lot of theory analysis.
4. This paper proposes a novel method for DA based on the theory.
5. Better results is obtained than baselines.

---

> ### Author Response · Authors · 2025-01-06
>
> We thank the reviewer for the useful suggestions to improve the results presented in the paper.
>
> **Additional experiments on Office-31 and DomainNet datasets.**
> We provide additional results for experiments as described in the general comments, included in Section 6.2 of the revised manuscript. We consider a small scale image dataset, "Office-31", and a large scale dataset, "DomainNet", using pretrained ResNets to map the images into feature vectors. Table 2 in the paper shows the average classification error across both datasets and scenarios of multi-source covariate shift, along with the standard deviations.
>
> Our main experimental takeaway is that  the results obtained show how the proposed methodology can accommodate deep learning frameworks, obtaining good performances dealing with multi-source covariate shift. The proposed method is more effective than existing methods for adaptation, especially when the testing set contains samples from an unseen domain for the training sources.

---

> ### Author Response · Authors · 2025-01-06
>
> **Comparison with the state-of-the-art.**
> Following the suggestions, we implemented the MLLS method for single-source label shift adaptation (Garg et al., 2020) and added the results of the method to the experimental results of the Section 6.1. Our main experimental takeaway is that the proposed method is more effective than existing methods for adaptation, especially when the weights become large (knock-out shift) for label shift adaptation because $\mathrm{p}_{\text{tr}}(y)$ is very small for certain $y\in\mathcal{Y}$.
>
> We would like to clarify that, although domain adaptation and distribution shift are closely related, this paper focuses on the role of weighting in addressing distribution shifts, including label shifts and multi-source label and covariate shifts. In addition, in the multi-source case we focus on the way the information from the multiple sources is utilized jointly in order to obtain classification rules that involve sample-dependent weighted combinations of feature mappings. Note that while domain adaptation methods encompass a variety of strategies beyond weighting, weighting techniques can be complementary with other techniques. Our results show the proposed method is flexible enough to accommodate deep learning and representation learning frameworks.
>
> | Datasets | Type of Shift | No Adapt. | TarS | BBSE | RLLS | MLLS | DW-LS |
> |-------------------|---------------|--------------------------|--------------------------|--------------------------|--------------------------|--------------------------|--------------------------|
> | **Adult** | | | | | | | |
> | | tweak-one | $.43\pm.01$ | $\boldsymbol{.07\pm.02}$ | $.29\pm.01$ | $.24\pm.24$ | $\boldsymbol{.07\pm.02}$ | $\boldsymbol{.07\pm.02}$ |
> | | knock-out | $.38\pm.02$ | $.30\pm.04$ | $.42\pm.05$ | $.43\pm.06$ | $.39\pm.05$ | $\boldsymbol{.28\pm.03}$ |
> | **Diabetes** | | | | | | | |
> | | tweak-one | $.45\pm.02$ | $.09\pm.03$ | $.33\pm.05$ | $.37\pm.15$ | $\boldsymbol{.08\pm.03}$ | $\boldsymbol{.08\pm.03}$ |
> | | knock-out | $.40\pm.01$ | $.33\pm.03$ | $.45\pm.06$ | $.46\pm.05$ | $.43\pm.06$ | $\boldsymbol{.29\pm.02}$ |
> | **Mammo** | | | | | | | |
> | | tweak-one | $.37\pm.01$ | $.12\pm.02$ | $.11\pm.02$ | $.18\pm.25$ | $\boldsymbol{.08\pm.03}$ | $.11\pm.01$ |
> | | knock-out | $.31\pm.01$ | $\boldsymbol{.21\pm.02}$ | $.44\pm.11$ | $.44\pm.11$ | $.42\pm.11$ | $.23\pm.09$ |
> | **Usenet2** | | | | | | | |
> | | tweak-one | $.50\pm.02$ | $.34\pm.01$ | $.29\pm.01$ | $.32\pm.11$ | $\boldsymbol{.20\pm.01}$ | $.24\pm.02$ |
> | | knock-out | $\boldsymbol{.38\pm.03}$ | $.39\pm.03$ | $.39\pm.03$ | $.39\pm.03$ | $.45\pm.06$ | $\boldsymbol{.38\pm.02}$ |
> | **Credit** | | | | | | | |
> | | tweak-one | $.48\pm.02$ | $.19\pm.01$ | $.32\pm.05$ | $.29\pm.17$ | $\boldsymbol{.10\pm.01}$ | $.16\pm.03$ |
> | | knock-out | $.27\pm.03$ | $.24\pm.06$ | $.25\pm.08$ | $.27\pm.09$ | $.30\pm.06$ | $\boldsymbol{.22\pm.05}$ |
> | **20news** | | | | | | | |
> | | tweak-one | $.64\pm.03$ | $.58\pm.08$ | $.59\pm.07$ | $\boldsymbol{.57\pm.07}$ | $.70\pm.12$ | $.58\pm.09$ |
> | | knock-out | $\boldsymbol{.64\pm.02}$ | $\boldsymbol{.64\pm.04}$ | $\boldsymbol{.64\pm.02}$ | $.65\pm.03$ | $.73\pm.05$ | $.66\pm.03$ |
> | | dirichlet | $.66\pm.04$ | $.65\pm.04$ | $.65\pm.04$ | $.65\pm.05$ | $.72\pm.02$  | $\boldsymbol{.64\pm.02}$ |
> | **Redwine** | | | | | | | |
> | | tweak-one | $.66\pm.13$ | $\boldsymbol{.48\pm.22}$ | $.55\pm.26$ | $.59\pm.25$ | $.57\pm .25$ | $.52\pm.25$ |
> | | knock-out | $.60\pm.03$ | $\boldsymbol{.55\pm.07}$ | $.58\pm.07$ | $.58\pm.07$ | $.59\pm.08$ | $\boldsymbol{.55\pm.09}$ |
> | | dirichlet | $.65\pm.07$ | $.58\pm.12$ | $.58\pm.12$ | $.58\pm.12$ | $.63\pm.07$ | $\boldsymbol{.57\pm.12}$ |

---

> ### Comment · Reviewer_jqcK · 2025-01-23
> **Thanks for the extra results**
>
> Thanks for the authors's extra experiments. The newly added results have addressed my concerns. I recommend acceptance.

---

### Author Response · Authors · 2025-01-06

**Changes in the revised paper:**
We uploaded an updated version of the paper, where we use purple texts to highlight the new content added to address the comments and suggestions from the reviewers. The main changes in the manuscript are as follows.

- The novelty and contribution with respect to (Segovia et al., 2023) is described in a new paragraph at the end of Section 2. In addition, the new Appendix B provides a detailed comparison of the proposed approach with the methods presented in (Segovia et al., 2023). Addressing the comment of Reviewer **B8SF** and **SA1f** regarding **Novelty of the proposed approach with respect to (Segovia-Martín et al., 2023)**.
- Addressing the comment of Reviewer **B8SF** and **jqcK** regarding **Experimental results**, we have added new experiments using the "DomainNet" and "Office-31" datasets for image classification. In addition, we have also added an experimental comparison with the MLLS method for label shift adaptation published in NeurIPS 2020.
- At the end of Section 4, we have described how the presented methodology can be used to deal with other cases of distribution shifts. This change addresses the comment of Reviewer **B8SF** regarding **Whether the proposed method can address the case of conditional shift**.
- At the end of Section 5, we have clarified the comparison between the proposed methodology and domain adaptation. In addition, we have further discussed the prior work in Appendix B and added multiple citations on state-of-the-art methods. This change addresses the comment of Reviewers **B8SF** and **jqcK** regarding **Comparison with Domain Adaptation**.
- At the end of Section 5, we have added a remark on the adaptability of the presented methodology to deep learning methods. This change addresses the comment of Reviewer **B8SF** regarding **Adaptability of the method to deep learning models**.
- Addressing the comment of Reviewer **B8SF** regarding **Impact of hyperparameters, and visualization-based analysis**, we have extended the experimental analysis, including Table 7 and Figures 6 and 7 in Appendix F. In addition, in Appendix F we discussed the role of hyperparameters in the presented methodology.

---

### Author Response · Authors · 2025-01-06

**Additional experimental results on image datasets (B8SF: conduct experimental comparisons of the proposed method using image datasets, and jqcK: conducting the experiments on the most common datasets in domain adaptation)**

We thank the reviewers for the suggestions. In the revised experimental results section, we have included results with image datasets, such as the small-scale dataset "Office-31", and the large-scale dataset "DomainNet". For these datasets, we have used ResNets to map the images into feature vectors.

For the experiments using "DomainNet" dataset, we considered 6 multiclass problems. Each of the training sources is generated by selecting the samples from each of the domains and removing one domain (the one corresponding to the experiment's name). The testing set is sampled from the rest of sapmles and from all the domains.
For the experiments using "Office-31" dataset, we consider 4 multiclass problems, selecting labels depending on the type of the object. Each of the two training sources are generated by selecting the samples from domains "amazon" and "webcam", and the testing set by combining the samples from all three domains.
We have included further details of the experimental setup in Appendix F of the updated version of the paper.

The table shown below (Table 2 in the paper) provides the new results obtained with these image datasets. The new results show that the proposed methods can significantly improve the performance obtained by the existing domain adaptation and covariate shift methods.

|Dataset|LR|KMM|DW-GCS|2SW-MDA|MS-DRL|CW KMM|DW-MSCS|
|--------------|-------------|-------------|-------------|--------------------------|-------------|--------------------------|---------------------------|
| **DomainNet** | | | | | | | |
| Clipart | $.41\pm.05$ | $.44\pm.05$ | $.46\pm.03$ | $.42\pm.06$ | $.51\pm.03$ | $.33\pm.03$              | $\boldsymbol{.31\pm.03}$  |
| Infograph | $.38\pm.04$ | $.43\pm.05$ | $.45\pm.03$ | $.44\pm.05$ | $.55\pm.05$ | $.35\pm.04$              | $\boldsymbol{.30\pm.03}$  |
| Painting | $.38\pm.04$ | $.44\pm.06$ | $.46\pm.02$ | $.43\pm.05$ | $.55\pm.05$ | $.35\pm.05$              | $\boldsymbol{.32\pm.02}$  |
| Quickdraw | $.42\pm.04$ | $.45\pm.04$ | $.46\pm.04$ | $.44\pm.03$ | $.55\pm.04$ | $.34\pm.04$              | $\boldsymbol{.31\pm.03}$  |
| Real | $.40\pm.05$ | $.45\pm.05$ | $.47\pm.02$ | $.43\pm.04$ | $.54\pm.04$ | $.35\pm.03$              | $\boldsymbol{.31\pm.03}$  |
| Sketch | $.40\pm.04$ | $.45\pm.05$ | $.47\pm.04$ | $.44\pm.04$ | $.55\pm.04$ | $\boldsymbol{.33\pm.04}$ | $\boldsymbol{.33\pm.03}$  |
| **Office-31** | | | | | | |  |
| Electronics | $.12\pm.04$ | $.19\pm.05$ | $.23\pm.03$ | $\boldsymbol{.09\pm.03}$ | $.39\pm.03$ | $.14\pm.03$ | $.11\pm.02$ |
| Stationery  | $.14\pm.03$ | $.16\pm.05$ | $.18\pm.04$ | $.11\pm.03$ | $.38\pm.07$ | $\boldsymbol{.10\pm.03}$ | $\boldsymbol{.10\pm.03}$ |
| Organization | $.13\pm.03$ | $.17\pm.04$ | $.21\pm.05$ | $.13\pm.04$ | $.39\pm.05$ | $.13\pm.03$ | $\boldsymbol{.12\pm.03}$ |
| Mixed | $.09\pm.03$ | $.14\pm.05$ | $.20\pm.03$ | $\boldsymbol{.08\pm.03}$ | $.31\pm.04$ | $.12\pm.04$ | $.10\pm.02$ |

---

### Author Response · Authors · 2025-01-06

More specifically, regarding 1, applying directly the double-weighting approach proposed in (Segovia-Martín et al., 2023) would result in $S$ classification rules $\lbrace h_s(y|x)\rbrace_{s\in[S]}$ that would need to be combined in order to obtain a classification for the testing instances. However, there is no direct way to combine the initial set of classification rules from the single source double weighting method. On the other hand, the proposed approach obtains a single classification rule defined in terms of combinations of the most relevant feature mappings. This combination is given by the set of weights $\lbrace\alpha_s(x,y)\rbrace_{s\in[S]}$ and depends on the testing instance that we want to classify. In addition, our methodology differs from the state-of-the-art since most of the existing methods obtain a set of classification rules and combine them in a post-hoc manner.

Regarding 2, using the set of weights $\lbrace\beta_s(x,y)\rbrace_{s\in[S]}$, we improve the effective sample size, obtaining better generalization bounds in comparison with existing reweighted techniques and single-source double-weighting. Using the set of weights $\lbrace\alpha_s(x,y)\rbrace_{s\in[S]}$, we obtain classification rules that achieve significant confidence in a larger region in comparison with the double-weighting method in (Segovia-Martín et al., 2023).
By choosing carefully the reference weights $\lbrace\alpha_s(x,y)\rbrace_{s\in[S]}$ and $\lbrace\beta_s(x,y)\rbrace_{s\in[S]}$, we can improve the existing trade-off in the double-weighting methodology. In particular, we defined weights $\alpha(x,y)$ in (31) taking into account that we do not need significant confidence $\alpha_s(x,y)\approx1$ for each source $s\in[S]$ but rather that there is enough confidence among all sources $\sum_{s=1}^S\alpha_s(x,y)\approx1$. This way, the region where the confidence of the classification rule is high is larger than using the weights of (Segovia-Martín et al., 2023). The reference weights $\beta_s(x)$ proposed in (31) take into account all the sources, and they are smaller than those of (Segovia-Martín et al., 2023), leading to reduced error due to finite sample sizes.

Regarding 3, the methodology presented in Section 4 is specific for multi-source distribution shift, which is not a straightforward extension of (Segovia-Martín et al., 2023), since the proposed methods efficiently leverage information from multiple sources by considering a set of weights $\lbrace\alpha_s(x,y)\rbrace_{s\in[S]}$ and $\lbrace\beta_s(x,y)\rbrace_{s\in[S]}$ that depend on all the training sources. This allows improved generalization bounds and adaptation to more general distribution shift scenarios where SOTA methods are challenged. Finally, the weights considered take into account the number of examples of each source, and may penalize more where the expectation is worse estimated due to a lower number of training samples available, as we show in Appendix D.

---

### Author Response · Authors · 2025-01-06

**Contributions and differences with respect to the ICML 2023 paper (Segovia-Martín et al., 2023) (B8SF and SA1f)**

We firmly believe the unified framework is novel and significant, contributing to the domain independently by providing a flexible approach to any (marginal) shift, which is out of the scope of the ICML 2023 paper.
The ICML paper focuses specifically on the problem of single-source covariate shift. In contrast, the present work addresses a broader class of distributional shifts that includes not only covariate shift but also general shifts and multi-source shifts, encompassing the previous paper. This paper presents new algorithms for label shift, multi-source covariate shift, and multi-source label shift. The experimental results demonstrate that the proposed methods obtain better performance than existing covariate and label shift approaches, particularly in scenarios involving distribution support mismatch, aligning with the theoretical performance guarantees and generalization bounds presented in this paper.

In addition, besides a unified approach and richer scenarios, we also want to highlight the challenges in multiple sources adaptation, where we propose new uncertainty sets and optimization methods, leading to new algorithms, new classification rules, and new generalization bounds presented in Section 4. The analysis of how the resulting weights for multiple sources interact with each other in the framework and how they contribute to the sample complexity is non-trivial and completely different from the single source setup. We highlight these differences in the new Section 2.3 and the new Appendix B. Specifically, the main contributions of the proposed methodology for multi-source distribution shifts are as follows.
1) We obtain improved classification rules that involve sample-dependent combinations of the most relevant feature mappings.
2) We achieve improved trade-off between the effective sample size and the confidence of the classification rules.
3) We efficiently leverage information from multiple sources by utilizing weights depending on all sources.

---

### Author Response · Authors · 2025-01-06

We thank the reviewers for their suggestions and feedback. In the following, we provide responses to the reviewers’ shared requests/questions. In the individual responses, we address the specific comments of each reviewer.

---

### Comment · Action_Editor_hWiA · 2025-02-06
**Proof of Theorem 4.1**

Dear Authors,

I'm trying to understand the proof of Theorem 4.1. I found it difficult to follow, especially in the following parts:

1.
> It is easy to see that the solution of such optimization problem $\\{\overline{\boldsymbol{\mu}}\_{s, 1}, \overline{\boldsymbol{\mu}}\_{s, 2}\\}\_{s \in[S]}$  satisfies that $\bar{\mu}\_{s, 1}^{(i)} \bar{\mu}\_{s, 2}^{(i)}=0$ for $s \in [S]$ and any $i$ such that $\lambda\_s^{(i)}$

2. How do we eliminate $h$ when $\mathcal{C}$ is strictly included in $\mathcal{Y}$?:
> $\sum\_{s=1}^S \Phi\_{\alpha\_s}(x, y)^{\mathrm{T}} \boldsymbol{\mu}\_s -\nu(x) \leq-1+\mathrm{h}(y \mid x), \forall x \in \mathcal{X}, y \in \mathcal{Y}$
> $\implies \sum\_{y \in \mathcal{C}}\left(\sum\_{s=1}^S \Phi\_{\alpha\_s}(x, y)^{\mathrm{T}} \boldsymbol{\mu}\_s-\nu(x)+1\right) \leq 1, \forall \mathcal{C} \subseteq \mathcal{Y}, x \in \mathcal{X}$

3. How do I conclude this?
> we have that any classification rule satisfying [..] is solution of [..]

Could you provide more explanations?

---

> ### Author Response · Authors · 2025-02-11
>
> We thank the Action Editor for his/her careful reading of the paper and the useful suggestions to clarify the proof of Theorem 4.1.
>
> * Regarding the first question regarding the **condition for solutions of optimization (69)**, we show that $\lbrace\bar{\boldsymbol{\mu}}\_{s,1},\bar{\boldsymbol{\mu}}\_{s,2}\rbrace\_{s\in[S]}$ satisfies that $\bar{\mu}\_{s,1}^{(i)}\bar{\mu}\_{s,2}^{(i)}=0$ for $s\in[S]$ and any $i$ such that $\lambda\_s^{(i)}>0$ by contradiction. Specifically, in other cases there would exist an $\epsilon>0$ such that the solution $\lbrace\bar{\boldsymbol{\mu}}\_{s,1},\bar{\boldsymbol{\mu}}\_{s,2}\rbrace\_{s\in[S]}$ satisfies that $\bar{\mu}^{(i)}\_{s,1}>\epsilon$, $\bar{\mu}^{(i)}\_{s,2}>\epsilon$ for $s\in[S]$ so that we would reach a contradiction because $\tilde{\mu}^{(i)}\_{s,1}=\bar{\mu}^{(i)}\_{s,1}-\epsilon$, $\tilde{\mu}^{(i)}\_{s,2}=\bar{\mu}^{(i)}\_{s,2}-\epsilon$, $\tilde{\mu}^{(j)}\_{s,1}=\bar{\mu}^{(j)}\_{s,1}$, $\tilde{\mu}^{(j)}\_{s,2}=\bar{\mu}^{(j)}\_{s,2}$ if $j\neq i$ satisfy the constrain of the problem and the value of the optimization problem at
> $\lbrace\tilde{\boldsymbol{\mu}}\_{s,1},\tilde{\boldsymbol{\mu}}\_{s,2}\rbrace\_{s\in[S]}$ is smaller than that at $\lbrace\bar{\boldsymbol{\mu}}\_{s,1},\bar{\boldsymbol{\mu}}\_{s,2}\rbrace\_{s\in[S]}$, contradicting the fact that $\lbrace\bar{\boldsymbol{\mu}}\_{s,1},\bar{\boldsymbol{\mu}}\_{s,2}\rbrace\_{s\in[S]}$ forms a solution. See a detailed derivation in the proof of Theorem 4.1 in page 31 of the updated manuscript.
>
> * Regarding the second question on **how do we eliminate $\mathrm{h}$**, we use the fact that $\sum\_{y\in\mathcal{Y}}\mathrm{h}(y|x)=1\Longrightarrow\sum\_{y\in\mathcal{C}}\mathrm{h}(y|x)\leq1$ for all $\mathcal{C}\subset\mathcal{Y}$ since $\mathrm{h}(y|x)\geq0$ for all $x\in\mathcal{X}$, $y\in\mathcal{Y}$. See a detailed derivation in page 32 of the updated manuscript.
>
> * Regarding the third question regarding the **conclusion of the proof**, we have that $\nu(x)\geq\varphi\_{01}(x)$, and, since we want to minimize
> $$\min\_{\nu(x),\mathrm{h}}\mathbb{E}\_{\mathrm{p}\_{\text{te}}}\nu(x)$$
> we have that the best case would be $\nu(x)=\varphi\_{01}(x)$ if the restriction of the problem is satisfied, i.e., if the classification rule $\mathrm{h}(y|x)$ satisfy that
> $$\mathrm{h}(y|x)\geq \sum\_{s=1}^S\Phi\_{\alpha_s}(x,y)^{\text{T}}\boldsymbol{\mu}\_s-\varphi_{01}(\lbrace\boldsymbol{\mu}\_s\rbrace\_{s=1}^S,x,\lbrace\alpha\_s\rbrace\_{s=1}^S)+1,\  \forall x\in\mathcal{X},y\in\mathcal{Y}.$$
> See a detailed derivation in page 32 of the updated manuscript.
>
> Please do not hesitate to let us know if there are derivations for which we can provide additional details.

---

> > ### Comment · Action_Editor_hWiA · 2025-02-12
> > **Normalization**
> >
> > Thank you for the clarification. That perfectly answers my questions, and now the steps of the proof are much clearer to me. I find it interesting.
> >
> > I have another small question. Does the solution $h^{\mathcal{U}}$ for the 01-loss in Theorem 4.1 (Eq. 37) satisfy the normalization condition $\sum_{y\in\mathcal{Y}} h^{\mathcal{U}}(y|x) = 1$? I can see that this is the case for the log loss, but I'm not sure about the 01-loss.

---

> ### Author Response · Authors · 2025-02-12
>
> Regarding the question regarding the normalization condition of the classification rule $\mathrm{h}$, i.e., $\sum\_{y\in\mathcal{Y}}\mathrm{h}(y|x)=1$,
> the key idea of the proof is to see that if
> \begin{equation*}
>     \mathcal{C'}\in\arg\max\_{\mathcal{C}\in\mathcal{Y}}\frac{\sum\_{s=1}^S\sum\_{y'\in\mathcal{C}}\Phi\_{\alpha_s}(x,y')^\text{T}\boldsymbol{\mu}\_s^*-1}{|\mathcal{C}|}
> \end{equation*}
> $\mathrm{h}(y|x)=0$ if $y\notin\mathcal{C'}$ and $\mathrm{h}(y|x)\geq0$ if $y\in\mathcal{C'}$, so that $\sum\_{y\in\mathcal{Y}}\mathrm{h}(y|x)=\sum\_{y\in\mathcal{C'}}\mathrm{h}(y|x)=1$.
> In the previous version of the manuscript it was demonstrated by contradiction but it is more clear to prove it in a more direct manner.
> See a detailed derivation in pages 32 and 33 of the updated manuscript.

---

> > ### Comment · Action_Editor_hWiA · 2025-02-15
> >
> > Thank you for the update. The proof looks correct and clear to me.

---

### Decision · Action_Editor_hWiA · 2025-02-17

**Recommendation:** Accept with minor revision

**Comment:**

The recommendations from the reviewers were Accept, Leaning Accept, and Leaning Reject. The reviewers are convinced that the paper shows the proposed method has great values through the theory and the experiments. The problem and the extension are highly relevant. It is good work and the acceptance criteria seem to be met.

If the acceptance is approved, I request the authors to prepare the camera ready version based on the latest version incorporating the discussions with the reviewers. Furthermore, here are a few of minor issues, which the authors could check:
- I could not find the clear definition of $\mu^\infty$ in Theorem 4.2. There seems to be a typo $\mu^\infty$ for $\mu^*$.
- I suppose that Theorem 4.2 is not for fully presenting the rate of convergence of the classifier because $\mu^*$ used in the bound is a random quantity and depends on $n$. There could be a remark on that.
- I'm not sure if it's clear which loss was used for the proposed method in the experiments.
- I don't think the standard 01-loss refers to the loss defined in Eq. (2), unless the range of the function is $\\{0, 1\\}$. They are very different and I find it confusing.
- The name "minimax risk classifier" might be also misleading. People might think it's referring to a classifier achieving the rate of the minimax risk in the sense of, e.g., Tsybakov (2009).

**Audience:**

Domain adaptation is an important and commonly encountered problem. This work proposes an extension of the recently proposed method [1] to a wide class of problems, including practically relevant scenarios such as label shift and multi-source domain adaptation. The reviewers seem to agree on these points.

**Claims And Evidence:**

This paper proposes a method that extends the double-weighting method previously proposed by [1] to a more general domain adaptation setup including the cases with multiple source domains and label shift.
The proposed method estimates two sets of weights with which the moment of the given feature will be approximately the same in the training and the test domain. It then minimizes an adversarial loss similar to that of distributionally robust learning, which aims to minimize the risk with respect to the worst test distribution one can think about among those that maintain the moment matching between the two domains, up to a slight deviation.
Thanks to the flexibility of the double weighting approach, the proposed method can avoid estimating the density ratio, which can take large values that can lead to variance large in the estimation. Instead, one can choose to estimates more bounded weights which have a more favorable property (Theorem 4.2).

The minimization of the proposed adversarial loss at first glance may appear difficult to solve, but the authors show that we only need to solve a simpler convex problem (Eq. 34; Theorem 4.1), similarly to [1].

The authors provide many nuemrical experiments using synthetic data and reald data. The results show that the proposed method often gives superior performance compared to previous methods.
In particular, the proposed method outperforms the method of [1] in the multi-source scenarios in most cases, showing that this extension is a great contribution.

[1] José I. Segovia-Martín, Santiago Mazuelas, and Anqi Liu. Double-weighting for covariate shift adaptation. In Proceedings of the 40th International Conference on Machine Learning, pp. 30439--30457, 2023.

---

> ### Author Response · Authors · 2025-03-11
>
> Dear Action Editor,
>
> We would like to extend our sincere gratitude for useful suggestions to improve the final version of the manuscript.
>
> In the latest revision, we have incorporated the reviewers' comments and suggestions addressed during the review process, as well as the final comments provided. The most recent changes to the manuscript are as follows:
> - We have clarified the definition of 0-1-loss in page 4, equation (2).
>
> - We have clarified the context and meaning of the minimax risk classifiers described in the manuscript, adding the reference of the minimax risk given in Tsybakov (2009) in Section 2.2, page 4.
>
> - We have included a note on how can we bound the norm of $||\boldsymbol{\mu}^\infty-\boldsymbol{\mu}^*||_1$ in the Theorem 3.2 for single-source, page 7.
>
> - We have clarified the meaning of $\boldsymbol{\mu}^\infty$ in Theorem 3.2 and Theorem 4.2, pages 7 and 12.
>
> - We have included the loss used for the proposed method in the experiments in the Appendix F, pages 41 and 42.